# Efficient Adaptive Federated Optimization

**Su Hyeong Lee** [1]   **Sidharth Sharma** [2]   **Manzil Zaheer** [3]   **Tian Li** [4]

## Abstract

Adaptive optimization plays a pivotal role in federated learning, where simultaneous server and client-side adaptivity have been shown to be essential for maximizing its performance. However, the scalability of jointly adaptive systems are often constrained by limited resources in communication and memory. In this paper, we introduce a class of efficient adaptive algorithms, named `FedAda`$^2$, designed specifically for large-scale, cross-device federated environments. `FedAda`$^2$ optimizes communication efficiency by avoiding the transfer of preconditioners between the server and clients, while simultaneously utilizing memory-efficient adaptive optimizers on the client-side to reduce extra on-device memory cost. Theoretically, we demonstrate that `FedAda`$^2$ achieves the same convergence rates for general, non-convex objectives as its more resource-intensive counterparts that directly integrate joint adaptivity. Empirically, we showcase the benefits of joint adaptivity and the effectiveness of `FedAda`$^2$ on several image datasets.

## 1. Introduction

Federated learning is a distributed learning paradigm which aims to train statistical models across multiple clients without transmitting raw data (McMahan et al., 2017; Li et al., 2020a; Wang et al., 2021a). In vanilla federated learning, a central server orchestrates the training process by distributing the global model to a subsample of thousands or even millions of clients. These clients collaboratively perform local stochastic gradient descent while drawing from their private data streams. After several epochs have elapsed, each client communicates their aggregate updates to the server, which averages this information to make an informed adjustment to the global model. This algorithm, using non-adaptive weight updates, is called *FedAvg* (McMahan et al., 2017). A recent trend is to investigate utilizing adaptive optimizers to support federated learning (Reddi et al., 2021). Adaptivity can be employed in either the server-side or the client-side, where joint adaptivity (consisting of global *and* local adaptive updates) has been shown to play a pivotal role in accelerating convergence and enhancing accuracy (Wang et al., 2021b).

Nevertheless, efficiency challenges remain for the successful deployment of jointly adaptive algorithms in practice, especially in cross-device federated settings (Kairouz et al., 2021). The server, which collects pseudogradients pushed by participating clients, consolidates a global approximation of the preconditioners for adaptive model updates. Typically, the server sends the preconditioners back to the clients to precondition local adaptive updates. This could lead to significant communication overhead that detracts from the advantages offered by adaptivity (Wang et al., 2022). Furthermore, dynamically varying client resource limitations restrict the reliability of client-side adaptive optimizers to be deployed in practice, especially when additional memory is required to maintain gradient statistics for preconditioning throughout each local training round.

In this work, we propose a class of efficient jointly adaptive distributed training algorithms, called `FedAda`$^2$, to mitigate the aforementioned communication and memory restrictions while retaining the benefits of adaptivity. `FedAda`$^2$ maintains an identical communication complexity as the vanilla FedAvg algorithm. Instead of transmitting global server-side preconditioners from the server to the selected clients, we propose the simple strategy of allowing each client to initialize local preconditioners from constants (such as zero), without any extra communication of preconditioners. In addition, when running local updates, we adopt existing memory-efficient optimizers that factorize the gradient statistics to reduced dimensions to save on-device memory. We prove that for the general, non-convex setting, `FedAda`$^2$ achieves the same convergence rate as prior federated adaptive optimizers (e.g., Reddi et al. 2021).

---

[1]Committee on Computational and Applied Mathematics, University of Chicago, Chicago, IL, USA [2]Department of Electrical and Computer Engineering, Carnegie Mellon University, Pittsburgh, PA, USA [3]Google DeepMind, New York, NY, USA [4]Department of Computer Science, University of Chicago, Chicago, IL, USA. Correspondence to: Su Hyeong Lee <sulee@uchicago.edu>.

Accepted to the Workshop on Advancing Neural Network Training at International Conference on Machine Learning (WANT@ICML 2024).

**Contributions.** Motivated by the importance of client-side adaptivity both empirically and theoretically (Section 3), we propose a simple and effective algorithm `FedAda`$^2$ to avoid extra communication cost and reduce on-device memory while retaining the benefits of joint server- and client-side adaptive optimization (Section 4). We provide convergence analyses for a class of `FedAda`$^2$ algorithms instantiated with different server-side and client-side adaptive methods (Section 5). To the very best of our knowledge, there are no known convergence results on joint adaptive federated optimization in the general convex or non-convex setting. Empirically, we demonstrate that `FedAda`$^2$, without transmitting preconditioners and employing on-device preconditioner compression, matches the performance of its more expensive counterparts and outperforms baselines without client or server adaptivity (Section 6).

## 2. Related Work

We now provide a brief overview of related work in adaptive federated learning and efficient[1] preconditioning.

**Adaptive Federated Optimization.** Adaptive gradient methods are designed to dynamically adjust the learning rate for each model parameter to address update sparsity or scale imbalance, leveraging historical gradient data to enhance optimization efficacy (Duchi et al., 2011). Recent developments in Federated Learning have harnessed adaptivity to augment the selection of server and client model parameter updates. Frameworks such as FedAdam (Reddi et al., 2021) and FederatedAGM (Tong et al., 2020) focus primarily on server-side adaptivity while enforcing a constant learning rate for the clients. Additionally, FedCAMS (Wang et al., 2022) delves into communication-efficient adaptive optimization by implementing error feedback compression to manage client updates while maintaining adaptivity solely on the server-side. Conversely, methodologies such as FedLALR (Sun et al., 2023), Local AdaAlter (Xie et al., 2019), and Local AMSGrad (Chen et al., 2020) have adopted client-side adaptivity exclusively. These approaches involve transmitting both client preconditioners and model parameters for global aggregation in the server. Moreover, some frameworks have embraced joint adaptivity. Local Adaptive FedOPT (Wang et al., 2021b) implements joint adaptivity while incorporating an additional client correction term. These terms, along with transmitted client pseudo-gradients, are aggregated on the server to construct a global

preconditioner used to synthesize the subsequent model update. In contrast with all these approaches, `FedAda`$^2$ avoids the transmission of any local/global preconditioners and optimizer states entirely, maintaining identical communication complexity as vanilla FedAvg despite leveraging joint adaptivity.

**Memory-Efficient Adaptive Optimizers.** The implementation of local adaptive methods substantially increases client memory requirements, as it necessitates the maintenance of local preconditioners. For some language models, it has been noted that the gradients combined with optimizer states consume significantly more memory than the actual model parameters themselves (Raffel et al., 2020). Algorithms such as Adafactor (Shazeer & Stern, 2018) address memory reduction by tracking moving averages of the reduction sums of squared gradients along a singular tensor axis, attaining a low-rank projection of the exponentially smoothed preconditioners. Galore (Zhao et al., 2024) extends this line of work by targeting the low-rank nature of the gradient tensor (possessing an identical shape as the weight tensor) and further projects the *gradients* into a reduced rank. Similarly, Shampoo (Gupta et al., 2018) collapses gradient statistics into separate preconditioning matrices for each tensor dimension, which is extended by Chen et al. (2019) via extreme tensoring. Due to significant empirical enhancement in wall time and convergence speed, we focus on SM3 (Anil et al., 2019) in our implementation and experiments; however, our theoretical framework covers a broad class of memory-efficient optimizers applied on the client-side (Section 5 and Appendix C).

## 3. Utility of Client-Side Adaptivity

In this section, we motivate our work by providing a theoretical description of how leveraging client-side adaptivity improves distributed learning, which is later validated in experiments (Section 6). Our analyses are motivated by prior works that uncover critical conditions under which centralized SGD can diverge, specifically in settings involving heavy-tailed gradient noise (Zhang et al., 2020). We begin by providing a definition of heavy-tailed noise previously reported in the literature for completeness, which is further motivated in Appendix A.2.

**Definition 3.1.** A random variable $\xi \sim \mathcal{D}$ follows a **heavy-tailed** distribution if the $\alpha$-moment is infinite for $\alpha \geq 2$.

We may now present the following proposition.

**Proposition 3.2.** *There exists a federated optimization problem with heavy-tailed client-side gradient noise such that the following arguments hold (where an appropriate learning rate schedule is chosen for (ii)):*

*(i) For vanilla FedAvg, given any client sampling strategy, if*

---

[1]There are various notions of 'efficiency' of adaptive methods in the context of the federated learning, two of them being communication efficiency and client memory efficiency. Our contribution specifically targets at reducing communication and memory costs incurred by *local preconditioners*, complementary with works that reduce communication by repeated local updates or gradient compression.

the probability $p_i^t$ of client $i$ with heavy-tailed gradient noise being sampled at step $t$ is non-zero, then $\mathbb{E}\|\nabla f(x_{t+1})\|^2 = \infty$ for any nontrivial learning rate schedule $\eta_\ell^t > 0$.

***(ii)*** *FedAvg with local adaptivity (i.e., via client-side Ada-Grad) bounds the error in expectation as*

$$\lim_{t\to\infty} \mathbb{E}\|x_t - x^*\| \leq \frac{2\sqrt{3}}{1-\hat{\varepsilon}} \quad \text{for some} \quad \hat{\varepsilon} \approx 0,$$

*where $x^*$ is the global optimum.*

A proof is given by construction in Appendix A, which reveals a vulnerability in non-adaptive federated learning. We show that even a single client with heavy-tailed gradient noise is able to instantaneously propagate their volatility to the global model, which severely destabilizes distributed learning in expectation. Unfortunately, recent works have confirmed that heavy-tailed gradient distributions are empirically observed (Nguyen et al., 2019; Simsekli et al., 2019; 2020), especially within model architectures utilizing attention mechanisms, including transformer-based models (Devlin et al., 2018; Zhang et al., 2020; Brown et al., 2020; Dosovitskiy et al., 2021). Proposition 3.2 suggests that client-side adaptivity has the potential to stabilize local model updates pushed from diverse and large-scale distributed sources, if communication bottlenecks and memory efficiency can be addressed.

The construction of the federated problem in Proposition 3.2 draws gradient noise from the Student $t$-distribution which is heavy-tailed depending on the parameter regime, but whose moments are relatively controlled nevertheless. We may exacerbate the severity of gradient stochasticity by inserting a singular client with Cauchy-distributed noise, while enforcing all other clients to follow non-heavy-tailed Gaussian gradient noise. We further detail this setting in Proposition A.2, Appendix A.

### 3.1. Deep Remorse of FedAvg and SGD

So far, we have examined two problems in which heavy-tailed gradient noise is guaranteed to destabilize distributed training in expectation. We now prove that this is an instantiation of a more general phenomenon in federated learning where a family of online $\mu$-strongly convex global objectives collapses to the identical failure mode. To our knowledge, this provable limitation of distributed training resultant from the heavy-tailed noise of a singular client has not previously been established within the literature. The proofs of all results are given in the appendix.

**Definition 3.3.** A learning algorithm $\mathcal{A}$ is **deeply remorseful** if it incurs infinite or undefined regret in expectation. If $\mathcal{A}$ is guaranteed to instantly incur such regret due to sampling even a single client with a heavy-tailed stochastic gradient distribution, then we say $\mathcal{A}$ is **resentful** of heavy-tailed noise.

We present the following theorem.

**Theorem 3.4.** *Let the global objectives $f_t(x)$ of a distributed training problem satisfy $\mu$-strong convexity for $t = 1, \ldots, T$. Assume that the participation probability of a client with a heavy-tailed stochastic gradient distribution is non-zero. Then, FedAvg becomes a deeply remorseful algorithm and is resentful of heavy-tailed noise. Furthermore, if the probability of the heavy-tailed client being sampled at step $t$ is nontrivial, then the variance of the global objective at $t+1$ satisfies $\mathbb{E}\|f_{t+1}(x_{t+1})\|^2 = \infty$.*

In federated learning, we typically have $f_t(x) \equiv f(x)$ for all $t = 1, \ldots, T$. Proposition 3.2 intuits that inserting local adaptivity successfully breaks the generality of remorse and heavy-tailed resent for FedAvg. A high-level overview is that client-side AdaGrad clips the local updates of each iteration, which mollifies the impact of stochasticity in perturbing the weight updates. This gives Proposition 3.5, which is formulated loosely without utilizing any advantages provided by local adaptivity except for clipping. Given that adaptive methods inherently include a clipping mechanism while also offering the benefits of adaptivity, we consider them to be preferable to clipped SGD for large-scale applications. This preference holds, provided that the memory and computational constraints of the clients can be adequately managed.

**Proposition 3.5.** *Let $f_t \in C(\mathbb{R}^d)$ for $t = 1, \ldots, T$ for $f_t$ not necessarily convex. Introducing client-side adaptivity via AdaGrad for the setting in Theorem 3.4 produces a non-remorseful and a non-resentful algorithm.*

Note that Proposition 3.5 can be straightforwardly extended to jointly adaptive methods. An advantage of federated learning is that when done tactfully, the large supply of clients enable the trainer to draw from a virtually unlimited stream of computational power. The downside is that the global model may be strongly influenced by the various gradient distributions induced by the private client data shards. In this paper, we focus specifically on adaptive optimization as a countermeasure to stabilize learning. In Section 4, we propose `FedAda`$^2$, which utilizes joint adaptivity in an efficient and scalable manner for distributed or federated training.

## 4. `FedAda`$^2$: Efficient Joint Server- and Client-Side Adaptivity

In federated learning, a server-side objective is formed by taking a balanced average of all client objectives $F_i(x)$ for $i \in [N]$,

$$f(x) = \frac{1}{N} \sum_{i=1}^{N} F_i(x).$$

In the case of unbalanced client data sizes, we note that rescaling the local objectives appropriately gives an equivalent formulation to the balanced case. With a slight abuse of notation, we denote $F_i(x) = \mathbb{E}_{z \sim \mathcal{D}_i}[F_i(x, z)]$ where $F_i(x, z)$ is the stochastically realized local objective and $\mathcal{D}_i$ is the data distribution of client $i$. For analytical purposes, we assume that the global objective does not diverge to negative infinity and admits a minimzer $x^*$. To realize joint adaptivity in federated systems, one natural baseline is to estimate (pseudo)gradient statistics on the server (i.e., server-side preconditioners or global preconditioners) before transmitting them to all participating clients at the start of every communication round. And then each selected client performs local adaptive steps with preconditioners starting from the global ones. This approach enables clients to utilize historical gradient information to make informed adjustments to their respective local models. However, transmitting (pseudo)gradient statistics, such as the second moment, at each round significantly increases the communication cost. In addition, running adaptive updates locally based on the local data introduces memory overheads.

**Scalar Preconditioner Initialization.** To enhance the feasibility of joint federated learning in cross-device settings, we first address extra major communication bottleneck brought by server-side preconditioners. We propose a simple strategy of uniformly initializing local preconditioners to zero (or some constant, as discussed later) at the beginning of each training round, thus eliminating the need for preconditioner transmission between server and clients. In addition to communication reduction, this approach enables the use of different optimizers on the server and clients, as the server and client can maintain independent gradient statistics estimates.

Assuming Adagrad is selected as the server-side optimizer (Reddi et al., 2021) for expository purposes, we have the following server update rule (**SU**) for $-\Delta_i^t$ the accumulated pseudogradient from client $i$ at step $t$,

$$
\begin{aligned}
\Delta_t &= \tfrac{1}{|\mathcal{S}^t|} \sum_{i \in \mathcal{S}^t} \Delta_i^t, \quad \widetilde{m}_t = \widetilde{\beta}_1 \widetilde{m}_{t-1} + (1 - \widetilde{\beta}_1)\Delta_t, \\
\widetilde{v}_t &= \widetilde{v}_{t-1} + \Delta_t^2, \quad x_t = x_{t-1} + \eta \tfrac{\widetilde{m}_t}{\sqrt{\widetilde{v}_t} + \tau}.
\end{aligned}
$$

(**SU**)

Here, $\widetilde{v}_t$ acts as the second moment statistic for the server-side pseudogradient $-\Delta_t$. An extension to the case when Adam is selected as the server optimizer is given in Appendix B.2. At each communication round, the server does not communicate $\widetilde{v}_t$ to the participating clients; instead, each client only receives $x_t$ and initializes the local preconditioners from zero. The variant of $\texttt{FedAda}^2$ where the client and server utilizes differing optimizers may also be realized as a special case of blended optimization (Section 5.1, Appendix C).

**Addressing Client-Side Resource Constraints.** To accommodate local memory restrictions, we employ memory-efficient optimizers for all clients. Our framework allows any such optimizer to be used, including a heterogeneous mixture within each communication round, and we provide a convergence guarantee for a very broad class of optimizer strategies in Theorem C.1. In this paper, we specifically focus on SM3 (Anil et al., 2019) adaptations of Adam and Adagrad.

Intuitively, SM3 exploits natural activation patterns observed in model gradients to accumulate approximate parameter-wise statistics for preconditioning. More precisely, the gradient information in each coordinate element $\{1, \ldots, d\}$ is blanketed by a cover $\{S_1, \ldots, S_q\}$ satisfying $\bigcup_{b=1}^q S_b = \{1, \ldots, d\}$ for which an auxiliary $\mu_k(b)$ is assigned for each $b \in [q]$. The $\mu_k(b)$ then act to form $v_k$ as a coordinate ascent upper bound to the squared gradient sum $\sum_{\ell=1}^k (g_{i,\ell}^t)^2$ as SM3 iterates over each $j \in [d]$. A more in-depth explanation is given in Appendix B.

As an add-on, utilizing the staleness of gradients to construct preconditioners has previously been suggested as a strategy to accelerate adaptive optimization without hurting the performance (Gupta et al., 2018; Li et al., 2023). Therefore, we may optionally further mollify the burden of client-side adaptive optimizers by enforcing delayed preconditioner updates. This is given by the following client update rule (**DCU**) which incorporates delay step $z$,

$$
\begin{aligned}
v_k(j) &\leftarrow \min_{b:S_b \ni j} \mu_{k-1}(b) + \left(g_{i,k}^t(j)\right)^2 \\
\mu_k(b) &\leftarrow \max\{\mu_k(b), v_k(j)\}, \text{for } \forall b : S_b \ni j
\end{aligned}
$$

(**DCU**)

for $(k-1)/z \in \mathbb{Z}$ and $v_k(j) \leftarrow v_{k-1}(j)$ otherwise, where $k$ is the local iteration.

These methodologies are consolidated into Algorithm 1, which we call $\texttt{FedAda}^2$. For simplicity, we describe the variant in which both the client and server employ AdaGrad. However, we present other instantiations of $\texttt{FedAda}^2$ in Appendix C.

## 5. Convergence Analyses

One of the challenges in proving the convergence bound for jointly adaptive systems lies in handling gradient mixing on the client-side. In typical convergence proofs in non-convex settings (e.g., Zhang et al. 2020; Reddi et al. 2021), an upper bound on the global objective is formed using L-smoothness, on which expectation is taken. When local SGD is used, the expectation independently acts on each localized stochastic gradient due to linearity. However, in the case of local adaptive methods that maintain second-order stochastic gradient histories, the individual gradients may not be isolated due to dependencies between client gradients in the model update. Furthermore, server adaptivity actively interferes with the

**Algorithm 1** `FedAda`$^2$: SM3-ADAGRAD Variant

**Require:**

  (**SM3**) A full cover $\{S_1, \ldots, S_q\} \subset \mathcal{P}([d])$.

  (**Main**) Initializations $x_0, \widetilde{v}_0 \geq \tau^2$ and $\widetilde{m}_0 = 0$. Smoothing terms $\varepsilon_s, \varepsilon, \tau > 0$. Global decay parameter $\widetilde{\beta}_1 \in [0, 1)$.

  (Optional) Update delay step size $z \in \mathbb{Z}_{\geq 1}$.

1: **for** $t = 1, \ldots, T$ **do**
2:   Sample subset $\mathcal{S}^t \subset [N]$ of clients
3:   **for** each client $i \in \mathcal{S}^t$ (in parallel) **do**
4:     Initialize $v_0 \geq 0$ (default $v_0 \leftarrow 0$), $x_{i,0}^t \leftarrow x_{t-1}$
5:     **for** $k = 1, \ldots, K$ **do**
6:       Draw unbiased gradient $g_{i,k}^t \sim \mathcal{D}(x_{i,k-1}^t)$
7:       $m_k \leftarrow g_{i,k}^t$, $\mu_k(b) \leftarrow 0$ for $\forall b \in \{1, \ldots, q\}$
8:       **for** $j = 1, \ldots, d$ **do**
9:         **Delayed Client Update (DCU)**
10:       **end for**
11:       **if** $0 < \|m_k/(\sqrt{v_k} + \varepsilon)\| < \varepsilon_s$, **do** $m_k \leftarrow 0$
12:       $x_{i,k}^t \leftarrow x_{i,k-1}^t - \eta_\ell \cdot m_k/(\sqrt{v_k} + \varepsilon)$
13:     **end for**
14:     $\Delta_i^t = x_{i,K}^t - x_{t-1}$
15:   **end for**
16:   **Server Update (SU)**
17: **end for**

---

application of techniques introduced in singularly client-side adaptive works such as Xie et al. 2019. To address these issues, we transition to the gradient descent setting and employ gradient clipping as a key technique, detailed below.

We encounter additional challenges with subsampling, which we manage by uniformly bounding the relevant terms (Appendix B.1). In typical gradient descent proofs, subsampling or random initialization introduces randomness, necessitating the formation of upper bounds via expectations. However, we provide a stronger result by proving a uniform upper bound for any arbitrary initial weight $x_0$ and client sampling scheme, thereby eliminating the need for expectation bounds related to initialization or subsampling randomness. To proceed with the convergence analysis, we make the following assumptions where the $\ell_2$ norm is taken by default.

**Assumption 1 (L-smoothness).** The local objectives are $L$-smooth and satisfy $\|\nabla F_i(x) - \nabla F_i(y)\| \leq L\|x - y\|$ for all $x, y \in \mathcal{X}$ and $i \in [N]$.

**Assumption 2 (Bounded Gradients).** The local objective gradient is bounded by $\left|[\nabla F_i(x, z)]_j\right| \leq G$ for $j \in [d]$.

These assumptions are standard within the literature and have been used in previous works (e.g. Xie et al. 2020;

Wang et al. 2020; Li et al. 2020b; Reddi et al. 2021). We note that Assumption 2 implies $|\nabla F_i(x)| \leq G$ for $x \in \mathcal{X}$ via Jensen and integrating over $z \sim \mathcal{D}_i$. In particular, this delineates an $\widetilde{L}$-Lipschitz family of client objectives given that the arguments are $\eta_\ell \varepsilon_s$-bounded away from each other,

$$\|\nabla F_i(x) - \nabla F_j(y)\| \leq \widetilde{L}\|x - y\| := \frac{2\sqrt{d}G}{\eta_\ell \varepsilon_s}\|x - y\|$$

for $i, j \in [N]$ and $\|x - y\| \geq \eta_\ell \varepsilon_s$. Here, $\varepsilon_s$ is an epsilon smoothing term that activates on the client-side. This quantity is used in a gradient clipping step in `FedAda`$^2$ (Algorithm 1), where if the local gradient update is negligibly small in magnitude, the gradient is autonomously clipped to 0. $\eta_\ell > 0$ is the local learning rate, and in particular, we note that $\widetilde{L} = \Theta(\eta_\ell^{-1})$. By taking $\varepsilon_s \to 0$, our algorithm recovers federated algorithms that do not utilize local gradient clipping. Therefore to remain consistent with most federated learning implementations, we take $\varepsilon_s$ to be a negligible value during the experiment section.

We now provide a convergence bound for the general, non-convex case under gradient descent which holds for both full and partial client participation. The full theorem statement as well as the generalization to the case where Adam is selected for the choice of adaptive optimizer is provided in Appendices B.1, B.2. We note that the convergence bound for the variant of `FedAda`$^2$ with SM3 inactive trivially follows from the analysis.

**Theorem 5.1.** *Let Assumptions 1, 2 hold. Then given any choice of initialization $x_0$, Algorithm 1 deterministically satisfies*

$$\min_{t\in[T]}\|\nabla f(x_{t-1})\|^2 \leq \frac{\Psi_1 + \Psi_2 + \Psi_3 + \Psi_4 + \Psi_5}{\Psi_6}$$

*where asymptotically,*

$$\psi_1 = \Theta(1), \; \psi_2 = \eta^2\eta_\ell^2 T, \; \psi_3 = \eta\eta_\ell^2 T, \; \psi_4 = \eta\eta_\ell \log(1 + T\eta_\ell^2)$$

*and*

$$\psi_5 = \begin{cases} \eta^3\eta_\ell^3 T & if\, \mathcal{O}(\eta_\ell) \leq \mathcal{O}(1) \\ \eta^3\eta_\ell T & if\, \Theta(\eta_\ell) > \Omega(1) \end{cases}, \; \psi_6 = \begin{cases} \eta\eta_\ell T & if\, \mathcal{O}(T\eta_\ell^2) \leq \mathcal{O}(1) \\ \eta\sqrt{T} & if\, \Theta(T\eta_\ell^2) > \Omega(1) \end{cases}.$$

In particular, we make no other assumptions on local or global learning rates to extract the most general use of Theorem 5.1. We have the following two corollaries:

**Corollary 5.2.** *Any of the following conditions are sufficient to ensure convergence of Algorithm 1:*

$(A): \quad \eta_\ell \leq \mathcal{O}(T^{-\frac{1}{2}}) \quad for \quad \Omega(T^{-1}) < \eta\eta_\ell < \mathcal{O}(1),$

$(B): \quad \eta_\ell = \Theta(T^{-\frac{49}{100}}) \quad for \quad \Omega(T^{-\frac{1}{2}}) < \eta < \mathcal{O}(T^{\frac{12}{25}}).$

**Corollary 5.3.** *Algorithm 1 converges at rate $\mathcal{O}(T^{-1/2})$.*

In particular, $\eta_\ell$ must necessarily decay to establish convergence in Theorem 5.1. However, striking a balance between local and global learning rates provably allows for greater than $\Omega(T^{1/3})$ divergence in the server learning rate without nullifying the desirable convergence property. This theoretically demonstrates the enhanced resilience of adaptive client-side federated learning algorithms to mitigate suboptimal choices of server learning rates.

### 5.1. Discussion of Convergence Bound

There have been several recent works exploring adaptivity and communication efficiency in federated learning. The convergence rate in Corollary 5.3 matches the state of the art for federated non-convex optimization methods (Xie et al., 2019; Chen et al., 2020; Tong et al., 2020; Reddi et al., 2021; Wang et al., 2022; Sun et al., 2023). However, to the best of our knowledge, there are no known convergence results of jointly adaptive federated systems, in either the GD nor SGD setting. Previous work most related to ours is given in Wang et al. 2021b, Theorem 1, which presents a decaying bound on the distance between the realized weights $x^t$ of their federated algorithm to the fixed point $\tilde{x}$ of a contractive operator for strongly convex objectives. However, the authors are unable to derive a closed form expression for the $h_i$ term that appears on the right hand side for client-side Adam and Adagrad, opting for a numerical approximation instead[2]. Our analysis holds for both these optimizers, while additionally incorporating the elements of delayed updates and memory efficiency.

**Generality of FedAda$^2$: Blended Optimization.** The gradient descent setting used in the analysis of Theorem 5.1 is conceptually equivalent to accessing oracle client workers capable of drawing their entire localized empirical data stream. While this constraint is a limitation of our theory (we refer to Section 7), it enables us to derive stronger results and induce additional adaptive frameworks for which our analysis generalizes. For instance, our bound deterministically guarantees asymptotic stabilization of the minimum gradient, regardless of initialization or client subsampling procedure. Furthermore, in Appendix C, we prove that our analysis can be extended to form a flexible framework for federated learning which we call Federated Blended Optimization (Algorithm 4).

Blended optimization distributes local optimizer strategies during the subsampling process, which are formalized as functions that take as input the availability of client resources and outputs hyperparameters such as delay step size $z$ or choice of optimizer (Adam, AdaGrad, SGD, etc). These may be chosen to streamline model training based

on a variety of factors, such as straggler mitigation or low availability of local resources. Under certain non-restrictive conditions on optimizer choices contained in the strategies, this dynamic hyperparameter allocation scheme allows for guaranteed convergence of the global gradient objective (Theorem C.1). In particular, this framework permits the deployment of different adaptive optimizers for each round per device, enhancing the utility of communication-efficient frameworks that do not retain preconditioners between clients or between the server and client. This flexibility is especially beneficial in scenarios where there is a mismatch between adaptive optimizer choices.

## 6. Experiments and Discussion

In this section, we conduct experiments to empirically validate the benefits of joint adaptivity motivated by communication and memory efficiency, which accumulates to an empirical derivation of FedAda$^2$. Our study aims to contrast FedAda$^2$ with frameworks that lack adaptivity on either the server or client-sides. In the pre-trained transformer setting below, we have discovered that adaptivity induces qualitatively varying model dynamics, leading to significant performance improvements after hyperparameter tuning.

**Evaluation Setup and Dataset Splits.** We explore the impact of joint adaptivity on a small vision transformer (ViT-S), introduced in Sharir et al. 2021 and pre-trained on the ImageNet-21K dataset (Ridnik et al., 2021). FedAda$^2$, along with its partially non-adaptive counterparts, are evaluated by fine-tuning the pre-trained model on the GLD-23K subset of the Google Landmarks dataset (Weyand et al., 2020), which represents a domain shift onto natural user-split pictorial data. Analogous experiments are carried out using CIFAR-10 (Krizhevsky, 2009), involving 1000 clients with a subsampling rate of 0.01. Data partitioning was achieved using LDA (Blei et al., 2003) with $\alpha = 0.5$. In ViT experiments, images are resized to $224 \times 224$ pixels, and 30 communication rounds are conducted for CIFAR-10 and 200 rounds for GLD-23K. For the latter dataset, the client optimizer employed a learning rate schedule that began with a 15-step linear warm-up phase, followed by a step-function decay where the learning rate was reduced by a factor of 0.1 every 15 backpropagation steps. We set the local training batch size to 32 for both datasets. Details for hyperparameter selection along with compute resources are provided in Appendix G.

**Benefits of Joint Adaptivity.** We briefly verify that a natural, expensive implementation of joint client- and server-side Adam with transmitted global preconditioners, supersedes the performance of FedAvg (McMahan et al., 2017) and singularly server-side adaptivity (FedAdam Reddi et al. 2021), as shown in Figure 1. Intuitively, these results are expected

---

[2]Moreover, we have discovered unrecoverable issues with their Lemma 2 via a counterexample, which forms a central backbone in developing their theory.

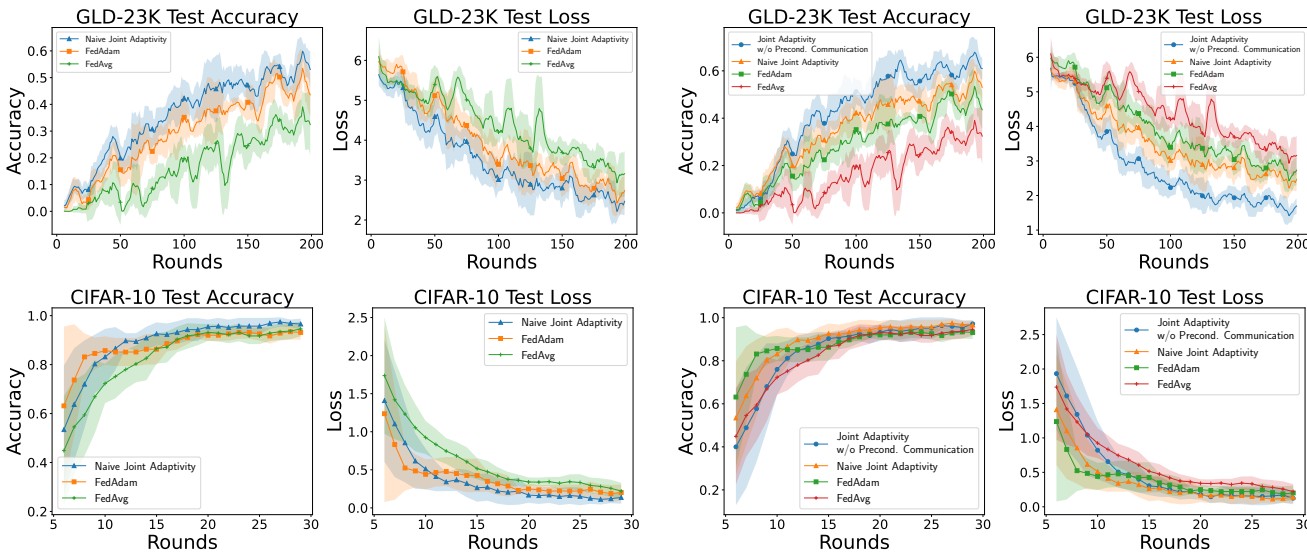

Figure 1. Naive joint adaptivity performs better than FedAdam and FedAvg on both GLD-23K and CIFAR-10 datasets. The shadows represent the moving averages of the accuracy.

Figure 2. Effects of joint adaptivity without preconditioner communication compared with the more expensive baseline of naive joint adaptivity with server-client preconditioner synchronization. We see that initializing from zero (as opposed to a global preconditioner as in naive joint adaptivity) does not meaningfully degrade performance and sometimes can even achieve higher test accuracies in certain applications.

as Section 3 confirms that adaptivity aids the stabilization of gradient updates. However, note that full gradient transmission incurs significant communication cost, as noted in Section 1. Moreover, the adaptive optimizer substantially increases the memory demand on the client, due to the maintenance of auxiliary second order statistics used to synthesize model updates in every local iteration. This motivates the communication-efficient and low-memory preconditioning design of $\texttt{FedAda}^2$, and we report the results below.

**Communication-Efficiency of $\texttt{FedAda}^2$.** To address the issue of additional communication overhead introduced by joint adaptivity, we propose to simply initialize the local preconditioners from zero (or a constant) at each round. In Figure 2, we compare the performance with and without preconditioner communication. We find that initializing from zero does not underperform the more complicated algorithm (naive joint adaptivity). To our surprise, on some datasets (GLD-23k), such a compromise for the purpose of reducing communication can even achieve better test performance than the more expensive baseline.

**Memory-Efficiency of $\texttt{FedAda}^2$.** Here, we investigate the performance of using approximated, memory-efficient local adaptive optimizers in $\texttt{FedAda}^2$. We implement SM3 on the client-side, which is a sublinear method to reduce memory consumption. In Figure 3 (top), the difference between $\texttt{FedAda}^2$ and joint adaptivity without preconditioner communication is the usage of SM3 as an approximation of the full client-side preconditioners during local updates. We see that $\texttt{FedAda}^2$ still retains the competitive perfor-

mance of naive joint adaptivity, while being communication- and memory-efficient. As an optional add-on, we provide a sample result for the delayed preconditioner update strategy detailed in Section 4, which may further reduce local computation. As intuitively expected in cases where preconditioners remain generally stable across local iterations, only updating them periodically on the client-side does not substantively affect the performance (Figure 3, bottom).

In addition to convergence plots, we provide accuracy numbers of $\texttt{FedAda}^2$ and the baselines in Table 1. We see that naive joint adaptivity (NJA) is superior to side-side adaptivity (FedAdam) as well as FedAvg. Empirically, avoiding preconditioner transmission and leveraging client-side preconditioner approximations (i.e., $\texttt{FedAda}^2$) does not harm the performance of its more expensive variants.

Table 1. Test accuracies of different methods. NJA stands for Naive Joint Adaptivity, JAPC for naive Joint Adaptivity without Preconditioner Communication, and DU for Delayed Updates.

| Datasets | FedAvg | FedAdam | NJA | JAPC | $\texttt{FedAda}^2$ | $\texttt{FedAda}^2$ + DU |
|----------|--------|---------|-------|-------|---------|-------------|
| CIFAR-10 | 95.59 | 92.80 | 96.4 | **97.2** | 96.4 | **97.2** |
| GLD-23K | 29.94 | 41.22 | 51.02 | **59.79** | 52.75 | 49.338 |

## 7. Extensions and Future Work

Our convergence analysis studies full gradient descent, whereas experiments are conducted using stochastic gradient descent for better industrial scalability. Although our

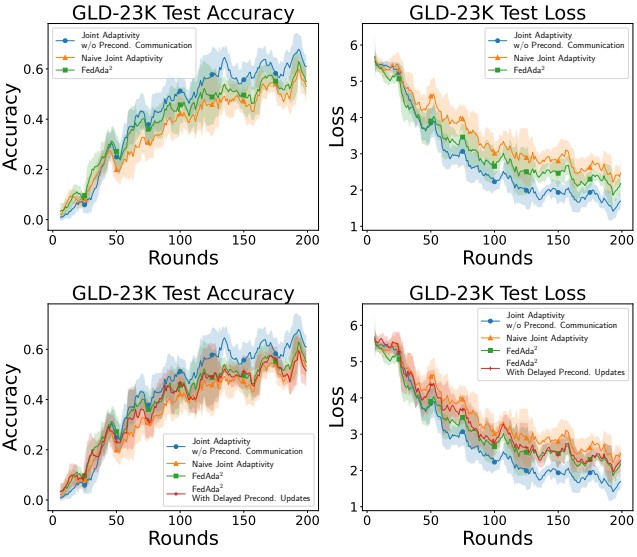

*Figure 3.* (Top) Performance of joint adaptivity with memory-efficient client preconditioners. We compare our communication- and memory-efficient adaptive algorithm (FedAda$^2$) with baselines of naive joint adaptivity (i.e., using server-side preconditioners to initialize client-side preconditioners) and only communication-efficient joint adaptivity (i.e., without transmitting server-preconditioners but using full preconditioners for local updates). We observe that using memory-efficient approximated local preconditioners does not materially harm performance. (Bottom) To further reduce computation costs on the client-side, we optionally add delayed preconditioner updates on top of FedAda$^2$ (FedAda$^2$ with Delayed Updates). We observe that the overall accuracy remains competitive with FedAda$^2$. Similar results for CIFAR-10 are given in Appendix F.2, Figure 6.

work provides a first convergence guarantee for a jointly adaptive system matching the state of the art (e.g. Xie et al. 2019; Li et al. 2020b; Reddi et al. 2021; Wang et al. 2022), moving to the stochastic gradient setting motivates additional challenges left for future research.

Another extension is to study the performance of Federated Blended Optimization (Section 5.1, Appendix C), a naturally induced framework discovered by generalizing our nonconvex convergence analysis. Blended optimizaton allows the trainer to utilize the unique strengths of each individual optimizer, balancing compute limitations and client noise. Generally, drawing from noisy local data streams will benefit more from adaptive methods in return for higher computational cost. Furthermore, each client has the option to run different optimizer strategies as the training rounds progress, adapting to individual resource constraints and distribution shifts in the data stream. We note that this approach faithfully mirrors real-world settings where the availability of local resources are actively dynamic. Future work will provide empirical results on the performance of blended optimization, including identifying the settings in which

mixing optimizer strategies are advantageous for distributed learning.

## 8. Conclusion

In this work, we introduce FedAda$^2$, a class of jointly adaptive algorithms designed to enhance scalability and performance in large-scale, cross-device federated environments. FedAda$^2$ is conceptually simple and straightforward to implement. By optimizing communication efficiency and employing localized memory-efficient adaptive optimizers, FedAda$^2$ significantly reduces the overhead associated with transferring preconditioners and extra on-device memory cost without degrading model performance. Our empirical results demonstrate the practical benefits of FedAda$^2$ in real-world federated learning scenarios. Future research could explore extensions of FedAda$^2$ to accommodate more diverse and large-scale federated learning tasks, such as by blending varied client optimizer strategies.

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

# A. Utility of Client-Side Apdaptivity

**Overview of Student's *t*-distribution**  For the convenience of the reader, we provide a brief summary of basic properties of the Student's $t$-distribution. Intuitively, the $t$-distribution can be understood as an approximation of the Gaussian with heavier tails. The density is given by

$$f_\nu(t) = \frac{\Gamma\left(\frac{\nu+1}{2}\right)}{\sqrt{\pi\nu}\Gamma\left(\frac{\nu}{2}\right)} \left(1 + \frac{t^2}{\nu}\right)^{-(\nu+1)/2}$$

where $\nu \in \mathbb{R}_{>0}$ is the degree of freedom (or normality parameter), and $\Gamma$ is the gamma function. We recover the normalized Gaussian as the degree of freedom tends to infinity. The first moment is 0 for $\nu > 1$, and the second moment satisfies $\nu/(\nu - 2)$ for $\nu > 2$ while being infinite for $1 < \nu \le 2$, where the heavy-tails are most pronounced. Following the convention of Zhang et al. 2020, we refer to a distribution as being heavy-tailed if the second moment is infinite.

The following proposition showcases the utility of local adaptivity in federated learning.

**Proposition A.1.** *There exists a federated optimization problem with heavy-tailed client noise which satisfies the following under FedAvg (where appropriate learning rate schedules are chosen for (ii-iv)):*

*(i) Given any client sampling strategy, if the probability $p_i^t$ of client $i$ with heavy-tailed gradient noise being sampled at step $t$ is non-zero, then $\mathbb{E}\|\nabla f(x_{t+1})\|^2 = \infty$ for any nontrivial learning rate schedule $\eta_\ell^t > 0$.*

*(ii) Local adaptivity via client-side AdaGrad bounds the error in expectation as*

$$\lim_{t\to\infty} \mathbb{E}\|x_t - x^*\| \le \frac{2\sqrt{3}}{1 - \hat{\varepsilon}} \quad \text{for some} \quad \hat{\varepsilon} \approx 0,$$

*where $x^*$ is the global optimum.*

*(iii) Furthermore, local adaptivity implicitly constructs a critical Lyapunov stable region which stabilizes the gradient variance via the following inequality which holds once any learned weight enters the region:*

$$\min_{t\in\{1,\dots,T\}} \mathbb{E}\|\nabla f(x_t)\|^2 \le \mathcal{O}\left(\frac{1}{T}\right).$$

*(iv) The global gradient variance of the federated problem with heavy-tailed client noise is fully stabilized via*

$$\mathbb{E}[\|\nabla f(x_t)\|^2] \le 2\|x_0\|^2 + 2\left(\int_1^\infty \frac{1}{x^2}\,\mathrm{d}x\right)^2 \quad \text{for} \quad \forall t \in \{1,\dots,T\}.$$

This proposition demonstrates that even a single client with heavy-tailed gradient noise is able to instantaneously propagate their volatility to the global model, which destabilizes federated training in expectation. However, recent work (Zhang et al., 2020) has shown that heavy-tailed gradient distributions appear frequently in language model applications, and more generally within model architectures utilizing any kind of attention mechanism, including transformers. To our knowledge, this provable failure mode of distributed training resultant from the unbiased, yet heavy-tailed noise of a singular client has not previously been reported within the literature.

**Proof of (i).**  Let the local stochastic objectives be given by $F_i(x, \xi_i) = x^2/2 + \xi_i x$ where gradient noise follows a $t$-distribution with $i + 1$ degrees of freedom, $\xi_i \sim t_{i+1}$ for $\forall i \in \{1,\dots,N\}$. Minibatches are sampled with replacement, which ensures that gradient noise in each client epoch are independent amongst and in between any two (possibly identical) clients, and further identically distributed conditional on the client ID $i$. Clearly, the global objective is

$$f(x) = \frac{1}{N}\sum_{i=1}^N \mathbb{E}_{\xi_i}[f_i(x, \xi_i)] = \frac{1}{N}\mathbb{E}\left[\frac{N}{2}x^2 + \sum_{i=1}^N \xi_i x\right] = \frac{1}{2}x^2.$$

For global step $t$, we subsample clients $\mathcal{S}^t$ following any sampling strategy, where $\mathcal{C}^t$ is the collection of all possible multisets $\mathcal{S}_r^t$ whose elements indicate (possibly repeated) client selection, with associated probabilities $p_C^t(r) > 0$ of realization for $r \in [|\mathcal{C}^t|]$. Assume that $1 \in \mathcal{S}_m^t$ for some $m$.

Then, FedAvg updates may be written

$$x_{t+1} = x_t - \frac{\eta_\ell}{|\mathcal{S}^t|} \sum_{i \in \mathcal{S}^t} \sum_{\ell=1}^{K} g_{i,\ell}^t$$

which gives the squared length of the global gradient under expectation as

$$\mathbb{E}_t \|\nabla f(x_{t+1})\|^2 = \mathbb{E}_t \left\| x_t - \frac{\eta_\ell}{|\mathcal{S}^t|} \sum_{i \in \mathcal{S}^t} \sum_{\ell=1}^{K} \left( \nabla f(x_{i,\ell-1}^t) + \xi_{i,\ell-1}^t \right) \right\|^2$$

$$= \mathbb{E}_{\xi|t} \mathbb{E}_{\mathcal{S}^t|\xi,t} \left\| x_t - \frac{\eta_\ell}{|\mathcal{S}^t|} \sum_{i \in \mathcal{S}^t} \sum_{\ell=1}^{K} \left( \nabla f(x_{i,\ell-1}^t) + \xi_{i,\ell-1}^t \right) \right\|^2$$

$$= \sum_{r=1}^{|\mathcal{C}^t|} \mathbb{E}_{\xi|t} p_C^t(r) \left\| x_t - \frac{\eta_\ell}{|\mathcal{S}_r^t|} \sum_{i \in \mathcal{S}_r^t} \sum_{\ell=1}^{K} \left( \nabla f(x_{i,\ell-1}^t) + \xi_{i,\ell-1}^t \right) \right\|^2$$

$$\geq p_C^t(m) \mathbb{E}_{\xi|t} \left\| x_t - \frac{\eta_\ell}{|\mathcal{S}_m^t|} \sum_{i \in \mathcal{S}_m^t} \sum_{\ell=1}^{K} \left( x_{i,\ell-1}^t + \xi_{i,\ell-1}^t \right) \right\|^2$$

where in the second equality we have conditioned on local gradient noise $\xi$ and stochastic realizations up to timestep $t$, using the law of iterated expectations. Recursively unravelling $x_{i,\ell-1}^t$ in terms of sampled noise and $x_{i,0}^t = x_t$ gives

$$x_{i,\ell-1}^t = x_{i,\ell-2}^t - \eta_\ell g_{i,\ell-2}^t = x_{i,0}^t - \eta_\ell \sum_{p=0}^{\ell-2} g_{i,p}^t$$

$$= x_{i,0}^t - \eta_\ell \left( \sum_{p=0}^{\ell-2} \nabla f(x_{i,p}^t) + \xi_{i,p}^t \right)$$

$$= x_{i,0}^t - \eta_\ell \left( \sum_{p=0}^{\ell-2} x_{i,p}^t + \xi_{i,p}^t \right)$$

$$= a_t x_t - \sum_{p=0}^{\ell-2} a_{i,p}^t \xi_{i,p}^t$$

where $a_t, a_{i,p}^t \in \mathbb{Q}[\eta_\ell]$ are polynomial functions of the learning rate with rational coefficients. Therefore, we have for $b_{i,p}^t \in \mathbb{Q}[\eta_\ell]$

$$p_C^t(m) \mathbb{E}_{\xi|t} \left\| x_t - \frac{\eta_\ell}{|\mathcal{S}_m^t|} \sum_{i \in \mathcal{S}_m^t} \sum_{\ell=1}^{K} \left( a_t x_t - \sum_{p=0}^{\ell-2} a_{i,p}^t \xi_{i,p}^t + \xi_{i,\ell-1}^t \right) \right\|^2$$

$$= p_C^t(m) \mathbb{E}_{\xi|t} \left\| \left( 1 - \frac{\eta_\ell}{|\mathcal{S}_m^t|} \sum_{i \in \mathcal{S}_m^t} \sum_{\ell=1}^{K} a_t \right) x_t + \frac{\eta_\ell}{|\mathcal{S}_m^t|} \sum_{i \in \mathcal{S}_m^t} \sum_{\ell=1}^{K} \left( \sum_{p=0}^{\ell-2} a_{i,p}^t \xi_{i,p}^t + \xi_{i,\ell-1}^t \right) \right\|^2$$

$$= p_C^t(m) \mathbb{E}_{\xi|t} \left\| \left( 1 - \frac{\eta_\ell}{|\mathcal{S}_m^t|} \sum_{i \in \mathcal{S}_m^t} \sum_{\ell=1}^{K} a_t \right) x_t \right\|^2 + \frac{\eta_\ell^2 p_C^t(m)}{|\mathcal{S}_m^t|^2} \mathbb{E}_{\xi|t} \left\| \sum_{i \in \mathcal{S}_m^t} \left( \sum_{p=0}^{K-2} b_{i,p}^t \xi_{i,p}^t + \xi_{i,K-1}^t \right) \right\|^2$$

$$\geq \frac{\eta_\ell^2 p_C^t(m) \mathbb{E} \left\| \xi_{1,K-1}^t \right\|^2}{|\mathcal{S}_m^t|^2} = \infty,$$

where we have used that $\xi_{i,\ell}^t \sim t_{i+1}$ independently with mean 0, for all permissible $i, \ell,$ and $t$.

**Proof of (ii).** We specialize to the setting with client-side AdaGrad with $K = 1$. Assume that clients $S^t$ have been selected to participate in the round, which gives the update as

$$
\begin{aligned}
x_{t+1} &= x_t - \frac{\eta_\ell}{|\mathcal{S}^t|} \sum_{i \in \mathcal{S}^t} \sum_{\ell=1}^{K} \frac{g_{i,\ell}^t}{\|g_{i,\ell}^t\| + \varepsilon} \\
&= x_t - \frac{\eta_\ell}{|\mathcal{S}^t|} \sum_{i \in \mathcal{S}^t} \frac{\nabla f(x_{i,0}^t) + \xi_{i,1}^t}{\|\nabla f(x_{i,0}^t) + \xi_{i,1}^t\| + \varepsilon} \\
&= x_t \left( 1 - \frac{\eta_\ell}{|\mathcal{S}^t|} \sum_{i \in \mathcal{S}^t} \frac{1}{\|x_t + \xi_i\| + \varepsilon} \right) - \frac{\eta_\ell}{|\mathcal{S}^t|} \sum_{i \in \mathcal{S}^t} \frac{\xi_i}{\|x_t + \xi_i\| + \varepsilon}
\end{aligned}
\tag{1}
$$

where we have gradually simplified notation. Noting that

$$
\int \frac{1}{\|x_t + \xi_i\| + \varepsilon} \, p(\xi_i) \, \mathrm{d}\xi_i \leq \frac{1}{\varepsilon},
$$

setting $\eta_\ell \leq \varepsilon$ gives

$$
\|\nabla f(x_{t+1})\| = \|x_{t+1}\| \leq \|x_t\| \cdot \left( 1 - \frac{\eta_\ell}{|\mathcal{S}^t|} \sum_{i \in \mathcal{S}^t} \frac{1}{\|x_t + \xi_i\| + \varepsilon} \right) + \frac{\eta_\ell}{|\mathcal{S}^t|} \sum_{i \in \mathcal{S}^t} \frac{\|\xi_i\|}{\|x_t + \xi_i\| + \varepsilon}.
\tag{2}
$$

Using $\mathbb{E}_t$ to denote expectation conditional over realizations up to step $t$, we have

$$
\mathbb{E}_t \|x_{t+1}\| \leq \|x_t\| \cdot \left( 1 - \frac{\eta_\ell}{|\mathcal{S}^t|} \mathbb{E}_t \left[ \sum_{i \in \mathcal{S}^t} \frac{1}{\|x_t + \xi_i\| + \varepsilon} \right] \right) + \frac{\eta_\ell}{|\mathcal{S}^t|} \sum_{i \in \mathcal{S}^t} \mathbb{E}_t \left[ \frac{\|\xi_i\|}{\|x_t + \xi_i\| + \varepsilon} \right].
$$

To further bound the right hand side, consider the functional

$$
I_i(\varepsilon) := \int \frac{1}{\|x_t + \xi_i\| + \varepsilon} \, p_{i+1}(\xi_i) \, \mathrm{d}\xi_i,
$$

where clearly

$$
I_i(0) \geq \int_{-x_t^-}^{-x_t^+} \frac{1}{\|x_t + \xi_i\|} \, p_{i+1}(\xi_i) \, \mathrm{d}\xi_i \approx \int_{0^-}^{0^+} \frac{p_{i+1}(-x_t)}{|x|} \, \mathrm{d}x = \infty
$$

and $I_i(1) < 1$. By continuity and strict decay of $I_i(\varepsilon)$, there exists $1 \gg \hat{\varepsilon}_i > 0$ and $\varepsilon_i \in (0, 1]$ such that for all $i \in [N]$, we have $1 > I_i(\varepsilon) \geq 1 - \hat{\varepsilon}_i$ for $\varepsilon \in [\varepsilon_i, 1]$. Taking $\varepsilon \in [\max_{i \in [N]} \varepsilon_i, 1]$ and $\hat{\varepsilon} := \max_{i \in [N]} \hat{\varepsilon}_i$, we thus obtain

$$
\mathbb{E}_t \|x_{t+1}\| \leq \|x_t\| \cdot (1 - \eta_\ell(1 - \hat{\varepsilon})) + \frac{\eta_\ell}{|\mathcal{S}^t|} \sum_{i \in \mathcal{S}^t} \mathbb{E}_t \left[ \frac{\|\xi_i\|}{\|x_t + \xi_i\| + \varepsilon} \right].
\tag{3}
$$

To bound the remaining term, it is easy to show that $\|\xi_i\| p_{i+1}(\xi_i)$ is symmetric around the origin $O$, and strictly increases from 0 to $(3/2 + 2/(i + 1))^{-1/2}$ while strictly decreasing afterwards. Defining the even extension of

$$
h_{i+1}(\xi_i) = \begin{cases} -\dfrac{x}{(3/2 + 2/(i+1))^{-1/2}} + \sup_{\xi_i \in \mathbb{R}} \|\xi_i\| p_{i+1}(\xi_i) + \epsilon & \text{for } 0 \leq \xi_i \leq \left( \dfrac{3}{2} + \dfrac{2}{i+1} \right)^{-\frac{1}{2}}, \\ \|\xi_i\| p_{i+1}(\xi_i) & \text{for } \xi_i > \left( \dfrac{3}{2} + \dfrac{2}{i+1} \right)^{-\frac{1}{2}} \end{cases}
$$

to be $h_{i+1}(\xi_i)$ for small $1 \gg \epsilon > 0$, we note that $1/(\|x_t + \xi_i\| + \varepsilon)$ analogously is symmetric around $\xi_i = -x_t$ while decaying with respect to the argument $\|x_t + \xi_i\|$. As $h_{i+1}(\xi_i)$ is symmetric around $O$ and decays moving to the left and right of $O$, by matching monotonicity and maxima with $1/(\|x_t + \xi_i\| + \varepsilon)$, we conclude that the left hand side of (4) is maximized for $x_t = 0$:

$$
\mathbb{E}_t \left[ \frac{\|\xi_i\|}{\|x_t + \xi_i\| + \varepsilon} \right] \leq \int \frac{h_{i+1}(\xi_i)}{\|\xi_i\| + \varepsilon} \, \mathrm{d}\xi_i = B_i.
\tag{4}
$$

Asymptotically as $\xi_i \to \infty$, we have

$$\frac{h_{i+1}(\xi_i)}{\|\xi_i\| + \varepsilon} \lesssim p_{i+1}(\xi_i),$$

which gives that $B_i < \infty$. Letting $B := \max_{i \in [N]} B_i$ and scheduling the learning rate $\eta_\ell^t = 1/((t + t_0)(1 - \hat{\varepsilon}))$ where $t_0$ is the smallest positive integer satisfying $\eta_\ell^t < \varepsilon$ for all $t$, we thus conclude

$$
\begin{aligned}
\mathbb{E}\|x_{t+1}\| &\leq \frac{t + t_0 - 1}{t + t_0} \mathbb{E}\|x_t\| + \frac{B}{(t + t_0)(1 - \hat{\varepsilon})} \\
&\leq \frac{t + t_0 - 2}{t + t_0} \mathbb{E}\|x_{t-1}\| + \frac{2B}{(t + t_0)(1 - \hat{\varepsilon})} \\
&\leq \cdots \leq \frac{t_0 - 1}{t + t_0} \mathbb{E}\|x_0\| + \frac{(t+1)B}{(t + t_0)(1 - \hat{\varepsilon})} \\
&\leq \mathcal{O}\left(\frac{1}{t}\right) + \frac{B}{1 - \hat{\varepsilon}}.
\end{aligned}
$$

As this bound holds for any choice of client subsample $S^t$, we are done. It is easy to show by straightforward integration that $B < 2\sqrt{3}$.

**Proof of (iii).** Our strategy is to locate a 1-shot stabilization regime of the gradient norm that is formed via local client adaptivity, which may be viewed as a Lyapunov stable region of the optimum $x^*$. From (2) and Jensen,

$$
\begin{aligned}
\|x_{t+1}\|^2 &\leq 2\|x_t\|^2 \cdot \left(1 - \frac{\eta_\ell}{|\mathcal{S}^t|} \sum_{i \in \mathcal{S}^t} \frac{1}{\|x_t + \xi_i\| + \varepsilon}\right)^2 + \frac{2\eta_\ell^2}{|\mathcal{S}^t|^2} \left(\sum_{i \in \mathcal{S}^t} \frac{\|\xi_i\|}{\|x_t + \xi_i\| + \varepsilon}\right)^2 \\
&\leq 2\|x_t\|^2 \cdot \left(1 - \frac{\eta_\ell}{|\mathcal{S}^t|} \sum_{i \in \mathcal{S}^t} \frac{1}{\|x_t + \xi_i\| + \varepsilon}\right)^2 + \frac{2\eta_\ell^2}{|\mathcal{S}^t|} \sum_{i \in \mathcal{S}^t} \left(\frac{\|\xi_i\|}{\|x_t + \xi_i\| + \varepsilon}\right)^2.
\end{aligned}
$$

We now impose $\eta_\ell \leq 2\varepsilon$, while letting $\|x_t\| < \delta$ for some $\delta \in \mathbb{R}_{>0}$. Taking expectations gives

$$\mathbb{E}_t\|x_{t+1}\|^2 \leq 2\|x_t\|^2 + \frac{2\eta_\ell^2}{|\mathcal{S}^t|} \sum_{i \in \mathcal{S}^t} \mathbb{E}_t \left(\frac{\|\xi_i\|}{\|x_t + \xi_i\| + \varepsilon}\right)^2,$$

and by similar arguments to the proof of **(ii)**, the summands of the second term are bounded uniformly by $\widetilde{B}$ which yields

$$\mathbb{E}\|x_{t+1}\|^2 \leq 2\delta^2 + 2\eta_\ell^2 \widetilde{B}.$$

Setting $\delta, \eta_\ell^t \leq \mathcal{O}(1/\sqrt{T})$ immediately gives the desired inequality.

**Proof of (iv).** An advantage of client-side adaptive optimization is the autonomous normalization and clipping of the stochastic gradients. Let $\eta_\ell^t := 1/t^2$. Telescoping (1) gives

$$x_{T+1} = x_0 - \sum_{t=1}^{T} \frac{\eta_\ell^t}{|\mathcal{S}^t|} \sum_{i \in \mathcal{S}^t} \sum_{\ell=1}^{K} \frac{g_{i,\ell}^t}{\|g_{i,\ell}^t\| + \varepsilon},$$

which implies

$$\|x_{T+1} - x_0\| = \left\| \sum_{t=1}^{T} \frac{\eta_\ell^t}{|\mathcal{S}^t|} \sum_{i \in \mathcal{S}^t} \sum_{\ell=1}^{K} \frac{g_{i,\ell}^t}{\|g_{i,\ell}^t\| + \varepsilon} \right\|$$

$$\implies \left| \|x_{T+1}\| - \|x_0\| \right| \leq \left\| \sum_{t=1}^{T} \frac{\eta_\ell^t}{|\mathcal{S}^t|} \sum_{i \in \mathcal{S}^t} \sum_{\ell=1}^{K} \frac{g_{i,\ell}^t}{\|g_{i,\ell}^t\| + \varepsilon} \right\|$$

$$\implies \|x_{T+1}\| \leq \|x_0\| + \left\| \sum_{t=1}^{T} \frac{\eta_\ell^t}{|\mathcal{S}^t|} \sum_{i \in \mathcal{S}^t} \sum_{\ell=1}^{K} \frac{g_{i,\ell}^t}{\|g_{i,\ell}^t\| + \varepsilon} \right\|$$

$$\implies \mathbb{E}\|x_{T+1}\|^2 \leq 2\|x_0\|^2 + 2\mathbb{E} \left\| \sum_{t=1}^{T} \frac{\eta_\ell^t}{|\mathcal{S}^t|} \sum_{i \in \mathcal{S}^t} \sum_{\ell=1}^{K} \frac{g_{i,\ell}^t}{\|g_{i,\ell}^t\| + \varepsilon} \right\|^2.$$

Substituting the learning rate schedule gives

$$\mathbb{E} \left\| \sum_{t=1}^{T} \frac{\eta_\ell^t}{|\mathcal{S}^t|} \sum_{i \in \mathcal{S}^t} \sum_{\ell=1}^{K} \frac{g_{i,\ell}^t}{\|g_{i,\ell}^t\| + \varepsilon} \right\|^2 \leq \mathbb{E} \left\| \sum_{t=1}^{T} K \eta_\ell^t \right\|^2$$

$$\leq \mathbb{E} \left\| K \int_1^\infty \frac{1}{x^2} \, \mathrm{d}x \right\|^2.$$

Therefore, we conclude that for any $t$,

$$\mathbb{E}\|x_t\|^2 \leq 2\|x_0\|^2 + 2K^2 \left( \int_1^\infty \frac{1}{x^2} \, \mathrm{d}x \right)^2.$$

### A.1. Exacerbation of singular client noise

**Overview of Cauchy–Lorentz distribution**  For the convenience of the reader, we provide a brief description of the Cauchy distribution $\mathcal{CL}(x_0, \gamma)$. The density is given by

$$f(x; x_0, \gamma) = \frac{1}{\pi \gamma \left[ 1 + \left( \frac{x - x_0}{\gamma} \right)^2 \right]} = \frac{1}{\pi} \left[ \frac{\gamma}{(x - x_0)^2 + \gamma^2} \right],$$

where $x_0$ is the location parameter and $\gamma > 0$ the scale parameter. Note that the Cauchy distribution is an example of "worst case gradient noise" that a federated problem may encounter in its clients. That is, the tails are so heavy that the distribution, despite being symmetric around the origin $O$, does not admit a mean due to being non-(Lebesgue) integrable. In particular, this indicates that the law of large numbers cannot be applied due to uncontrolled stochasticity, which lethally destabilizes pure stochastic gradient descent. Despite this limitation, we provide an example demonstrating that local adaptivity can be utilized to successfully mollify extreme client noise even in this "worst case" setting.

**Proposition A.2.** *There exists a generalized federated optimization problem which satisfies the following under FedAvg:*

*(i) Given any client sampling strategy without replacement, if the probability $p_i^t$ of client $i$ with heavy-tailed gradient noise being sampled at each step $t$ is non-zero, then $\mathbb{E}\|\nabla f(x_{t+1})\| = \infty$ or $\mathbb{E}\|\nabla f(x_t)\| = \infty$ for any $t \in \mathbb{Z}_{\geq 1}$ and nontrivial learning rate $\eta_\ell^t > 0$.*

*(ii) Under local adaptivity via client-side AdaGrad, we have bounded gradient length as*

$$\lim_{t \to \infty} \mathbb{E}\|\nabla f(x_t)\| \leq \frac{2}{1 - \hat{\varepsilon}} \quad \text{for some} \quad \hat{\varepsilon} \approx 0.$$

**Proof of (i).**  We provide a similar construction as in the proof of Theorem A.1. Let all local stochastic objectives be given by $F_i(x, \xi_i) = x^2/2 + \xi_i x$ where client gradient noise mostly models a Gaussian, $\xi_i \sim \mathcal{N}(0, \sigma_i^2)$ for $\forall i \in \{2, \ldots, N\}$ and

$\sigma_i \in \mathbb{R}$. For the first client, we let $\xi_1 \sim \mathcal{CL}(0, \gamma)$ for any $\gamma \in (0, 1/3)$. We sample minibatches with replacement, but clients are selected without replacement. In this case, we must consider a generalized version of the federated objective as strictly speaking, the deterministic local objective

$$\mathbb{E}_{\xi_1}[F_1(x, \xi_1)] = \frac{1}{2}x^2 + x \int \xi_1 \, \mathrm{d}\xi_1$$

does not exist due to extreme stochasticity. That is, even though $\mathcal{CL}(0, \gamma)$ is symmetric around $O$, $\mathbb{E}_{\xi_1}[\xi_1]$ is not Lebesgue integrable. Most importantly, this implies that the law of large numbers cannot be applied. Note that such a construction dislocates this example from the vast majority of convergence results, as most assume bounded variance or controlled gradient noise which sidesteps the consideration of the kind of stochasticity that we explore here entirely. To proceed with the analysis, we use symmetry to define the reasonable objective

$$\mathbb{E}[F_1(x, \xi_1)] = \frac{1}{2}x^2$$

which is consistent with the desired population objective that is distributed across all other clients, though with less noise. As before, we have the convex global objective $f(x) = x^2/2$. Note that it can be shown that the empirical mean of the Cauchy distribution follows the Cauchy distribution, that is, the CL-distribution is stable.

As the general case has been handled in Theorem A.1 (i), we specialize to $K = 1$. To simplify notation, assume that participating clients have been selected as $\mathcal{S}^t$, where client 1 participates. Then, the FedAvg update may be written

$$x_{t+1} = x_t - \frac{\eta_\ell}{|\mathcal{S}^t|} \sum_{i \in \mathcal{S}^t} g_{i,1}^t$$

which gives the length of the global gradient under expectation as

$$
\begin{aligned}
\mathbb{E}\|\nabla f(x_{t+1})\| = \mathbb{E} & \left\| x_t - \frac{\eta_\ell}{|\mathcal{S}^t|} \sum_{i \in \mathcal{S}^t} \left( \nabla f(x_{i,0}^t) + \xi_{i,1}^t \right) \right\| \\
\geq \mathbb{E} & \left\| \frac{\eta_\ell}{|\mathcal{S}^t|} \xi_{1,1}^t \right\| - \mathbb{E} \left\| \left( 1 - \frac{\eta_\ell}{|\mathcal{S}^t|} \right) x_t - \frac{\eta_\ell}{|\mathcal{S}^t|} \sum_{i \in \mathcal{S}^t \setminus \{1\}} \left( \nabla f(x_{i,0}^t) + \xi_{i,1}^t \right) \right\| \\
\geq \mathbb{E} & \left\| \frac{\eta_\ell}{|\mathcal{S}^t|} \xi_{1,1}^t \right\| - \mathbb{E} \left\| (1 - \eta_\ell) x_t - \frac{\eta_\ell}{|\mathcal{S}^t|} \sum_{i \in \mathcal{S}^t \setminus \{1\}} \xi_{i,1}^t \right\| \\
\geq \mathbb{E} & \left\| \frac{\eta_\ell}{|\mathcal{S}^t|} \xi_{1,1}^t \right\| - \mathbb{E} \left\| (1 - \eta_\ell) x_t \right\| - \frac{\eta_\ell}{|\mathcal{S}^t|} \sum_{i \in \mathcal{S}^t \setminus \{1\}} \mathbb{E} \left\| \xi_{i,1}^t \right\|
\end{aligned}
$$

Note that we allow $\eta_\ell = 1$. As $\mathbb{E} \left\| \xi_{i,1}^t \right\| < \infty$ for $i \in \{2, \ldots, N\}$, we thus have

$$\mathbb{E}\|\nabla f(x_{t+1})\| + |1 - \eta_\ell| \, \mathbb{E} \|\nabla f(x_t)\| \geq \infty$$

which gives the desired result.

**Proof of (ii).** As we intervened only on gradient noise while preserving client objectives, an analogous proof strategy used in Theorem A.1 (ii) carries through. The only difference is the value of $B$, which may be computed as being upper bounded by 2 for $\gamma < 1/3$.

### A.2. FedAvg and Stochastic Gradient Descent are deeply remorseful

In Appendix A, we have provided two localized examples of how heavy-tailed gradient noise can destabilize distributed training. In this subsection, we prove that this is an instantiation of a more general phenomenon in which federated learning with a $\mu$-strongly convex global objective collapses to an analogous failure mode. We begin by motivating a precise definition of heavy-tailed noise previously reported in the literature (Zhang et al., 2020) for completeness.

**Definition A.3.** A random variable $\xi \sim \mathcal{D}$ follows a **heavy-tailed** distribution if the $\alpha$-moment is infinite for $\alpha \geq 2$.

Intuitively, this expresses that the $\alpha$-moment is not sparsely supported outside a compact interval. That is, $\int_{\|\xi\|>R} \|\xi\|^\alpha p(\xi)\, d\xi < \infty$ indicates a dense support integrating to infinity in the closed ball $\mathcal{B}_0(R)$, and a light tail for $\mathcal{B}_0(R)^c$. Definition 3.1 enforces that the noise must not decay rapidly outside said compact ball, i.e. that light tails must be excluded. This follows from the observation that $\int_{\|\xi\|>R} \|\xi\|^\alpha p(\xi)\, d\xi = \infty$ for all $\alpha \geq 2$ and any $R \geq 0$ because $\int_{\|\xi\|\leq R} \|\xi\|^\alpha p(\xi)\, d\xi \leq R^\alpha < \infty$ via continuity and the extremal value theorem. By equivalence of norms on $\mathbb{R}^d$ and hence their preserved continuity, we analogously have for $\|\cdot\|_\infty$ the supremum norm,

$$\int_{\|\xi\|_\infty > R} c^\alpha \|\xi\|_2^\alpha\, p(\xi)\, d\xi \geq \int_{\|\xi\|_\infty > R} \|\xi\|_\infty^\alpha\, p(\xi)\, d\xi = \infty$$

for some $c > 0$. To proceed with the analysis, we impose an integrability condition on the mean, which gives $\mathbb{E}[\xi] = \mu \in \mathbb{R}^d$.

**Problem Setup.** The local objectives are determined by $F_i(x) = \mathbb{E}_z[F_i(x,z)]$, where $z$ integrates over the randomness in the stochastic objective. The gradient noise $\xi$ is additively modeled via a possibly uncentered random variable with $\mathbb{E}(\xi) = \mu$. Minibatches are sampled with replacement, implying that gradient noise in each client epoch are independent amongst and in between any two possibly identical clients. We analyze the case where noise is identically distributed conditional on client ID $i$. The global objective is given as the expected client objective under the uniform sampling prior, $f(x) = \sum_{i\in[N]} F_i(x)/N$.

We now present the following definition.

**Definition A.4.** A learning algorithm $\mathcal{A}$ is **deeply remorseful** if it incurs infinite or undefined regret in expectation. If $\mathcal{A}$ is guaranteed to instantly incur such regret due to sampling even a single client with a heavy-tailed stochastic gradient distribution, then we say $\mathcal{A}$ is **resentful** of heavy-tailed noise.

We are now ready to prove the following theorem.

**Theorem A.5.** *Let the global objectives $f_t(x)$ of a distributed training problem satisfy $\mu$-strong convexity for $t = 1,\ldots,T$. Assume that the participation probability of a client with a heavy-tailed stochastic gradient distribution is non-zero. Then, FedAvg becomes a deeply remorseful algorithm and is resentful of heavy-tailed noise. Furthermore, if the probability of the heavy tailed client being sampled at step $t$ is nontrivial, then the variance of the global objective at $t+1$ satisfies $\mathbb{E}\|f_{t+1}(x_{t+1})\|^2 = \infty$.*

*Proof.* Assuming that a heavy-tailed client may be subsampled at step $t$ with non-zero probability, let us show that the regret

$$R(T) := \sum_{t=1}^T f_t(x_t) - \sum_{t=1}^T f_t(x^*)$$

is infinite under expectation, assuming it is well-defined. Here, $x^*$ is taken to be the argument uniformly minimizing the materialized global objectives up to step $T$, $x^* := \arg\min_x \sum_{t=1}^T f_t(x)$. For notational simplicity, we carry out the analysis conditioned on the event that the heavy-tailed client has been subsampled. We aim to show that $\mathbb{E}[f_{t+1}(x_{t+1}) - f_{t+1}(x^*)] = \infty$ where $x^*$ is arbitrarily fixed and $f_{t+1}$ satisfies $\mu$-strong convexity. Clearly,

$$f_{t+1}(x_{t+1}) \geq f_{t+1}(x_t) - \left\langle \nabla f_{t+1}(x_t), \frac{\eta_\ell}{|\mathcal{S}^t|} \sum_{i\in\mathcal{S}^t} \sum_{\ell=1}^K g_{i,\ell}^t \right\rangle + \frac{\mu\eta_\ell^2}{2|\mathcal{S}^t|^2} \left\| \sum_{i\in\mathcal{S}^t} \sum_{\ell=1}^K g_{i,\ell}^t \right\|^2$$

$$\geq f_{t+1}(x_t) - \left\langle \nabla f_{t+1}(x_t), \frac{\eta_\ell}{|\mathcal{S}^t|} \sum_{i\in\mathcal{S}^t} \sum_{\ell=1}^K \left(\nabla f(x_{i,\ell-1}^t) + \xi_{i,\ell-1}^t\right) \right\rangle$$

$$+ \frac{\mu\eta_\ell^2}{2|\mathcal{S}^t|^2} \left\| \sum_{i\in\mathcal{S}^t} \sum_{\ell=1}^K \left(\nabla f(x_{i,\ell-1}^t) + \xi_{i,\ell-1}^t\right) \right\|^2.$$

Denoting $\mathbb{E}_{t+}[\cdot]$ to be the expectation conditional over all stochastic realizations up to step $t$ and $\ell = K - 1$, we have

$$\mathbb{E}_{t+}[f_{t+1}(x_{t+1})] \geq f_{t+1}(x_t) - \left\langle \nabla f_{t+1}(x_t), \frac{\eta_\ell}{|\mathcal{S}^t|} \sum_{i \in \mathcal{S}^t} \left( \left( \sum_{\ell=1}^{K-1} \nabla f(x_{i,\ell-1}^t) + \xi_{i,\ell-1}^t \right) + \nabla f(x_{i,K-1}^t) \right) \right\rangle$$

$$- \left\langle \nabla f_{t+1}(x_t), \frac{\eta_\ell}{|\mathcal{S}^t|} \sum_{i \in \mathcal{S}^t} \mathbb{E}_{t+} \left[ \xi_{i,K-1}^t \right] \right\rangle + \frac{\mu \eta_\ell^2}{2|\mathcal{S}^t|^2} \mathbb{E}_{t+} \left\| \sum_{i \in \mathcal{S}^t} \sum_{\ell=1}^{K} \left( \nabla f(x_{i,\ell-1}^t) + \xi_{i,\ell-1}^t \right) \right\|^2. \tag{5}$$

As the means of all gradient noise are finite (typically centered at 0), it suffices to show that

$$\mathbb{E}_{t+} \left\| \sum_{i \in \mathcal{S}^t} \sum_{\ell=1}^{K} \left( \nabla f(x_{i,\ell-1}^t) + \xi_{i,\ell-1}^t \right) \right\|^2 = \infty.$$

However, this is clear as expanding the norm gives

$$\mathbb{E}_{t+} \left\| \sum_{i \in \mathcal{S}^t} \sum_{\ell=1}^{K} \left( \nabla f(x_{i,\ell-1}^t) + \xi_{i,\ell-1}^t \right) \right\|^2 = \left\| \sum_{i \in \mathcal{S}^t} \sum_{\ell=1}^{K-1} \left( \nabla f(x_{i,\ell-1}^t) + \xi_{i,\ell-1}^t \right) + \sum_{i \in \mathcal{S}^t} \nabla f(x_{i,K-1}^t) \right\|^2$$

$$+ 2 \left\langle \sum_{i \in \mathcal{S}^t} \sum_{\ell=1}^{K-1} \left( \nabla f(x_{i,\ell-1}^t) + \xi_{i,\ell-1}^t \right) + \sum_{i \in \mathcal{S}^t} \nabla f(x_{i,K-1}^t), \sum_{i \in \mathcal{S}^t} \mathbb{E}_{t+}[\xi_{i,K-1}^t] \right\rangle + \sum_{i \in \mathcal{S}^t} \mathbb{E}\|\xi_{i,K-1}^t\|^2,$$

where in the final line we used the independence of the noise random variables. As there exists $i \in \mathcal{S}^t$ that satisfies heavy-tailed noise, we obtain

$$\mathbb{E}_{t+}[f_{t+1}(x_{t+1})] \geq \infty.$$

Taking expectations on both sides gives that $\mathbb{E}[f_{t+1}(x_{t+1})] \geq \infty$ under the law of iterated expectations, assuming that the expectation is well-defined. Thus, FedAvg is deeply resentful of the influence of heavy-tailed noise.

Now, we change perspectives and write the general form of (5) as

$$f_{t+1}(y) \geq f_{t+1}(x) + \langle \nabla f_{t+1}(x), y - x \rangle + \frac{\mu}{2} \|y - x\|^2$$

$$= f_{t+1}(x) + \sum_{j=1}^{d} (\nabla f_{t+1}(x))_j (y_j - x_j) + \frac{\mu}{2} \sum_{j=1}^{d} (y_j - x_j)^2.$$

For any arbitrarily fixed $x$, there exists $\tilde{a}_{t+1,j} > 0$, $R_j > 0$, and $\tilde{b}_{t+1,j} < 0$ such that

$$\tilde{f}_{t+1,j}(y_j) = \begin{cases} \tilde{a}_{t+1,j}(y_j - R_j) & \text{for} \quad y_j > R_j, \\ 0 & \text{for} \quad |y_j| \leq R_j, \\ \tilde{b}_{t+1,j}(y_j + R_j) & \text{for} \quad y_j < -R_j, \end{cases} \tag{6}$$

and

$$0 \leq \tilde{f}_{t+1,j}(y_j) \leq \frac{f_{t+1}(x)}{d} + (\nabla f_{t+1}(x))_j (y_j - x_j) + \frac{\mu}{2} (y_j - x_j)^2$$

for $|y_j| > R_j$. Without loss of generality, we may substitute $\tilde{a}_{t+1,j} \leftarrow \tilde{a} = \min_j \tilde{a}_{t+1,j}$, $\tilde{b}_{t+1,j} \leftarrow \tilde{b} = \max_j \tilde{b}_{t+1,j}$, and $R_j \leftarrow R := \max_{j \in [d]} R_j$. We thus have

$$\mathbb{E}_{t+}[\|f_{t+1}(x_{t+1})\|^2] \geq \mathbb{E}_{t+} \left[ \chi\{x_{t+1} \in B_R^\infty(0)^c\} \|f_{t+1}(x_{t+1})\|^2 \right]$$

where $\chi$ is the indicator and $B_R^\infty(0)$ is the closed ball in $\mathbb{R}^d$ under the infinity norm centered at 0. As $f_{t+1}(y) \geq \sum_{j=1}^{d} \tilde{f}_{t+1,j}(y_j)$ for $y \in B_R^\infty(0)^c$,

$$\mathbb{E}_{t+}[\|f_{t+1}(x_{t+1})\|^2] \geq \mathbb{E}_{t+}[\chi\{x_{t+1} \in B_R^\infty(0)^c\} \| \sum_{j=1}^{d} \tilde{f}_{t+1,j}(x_{t+1})\|^2]$$

$$\geq \mathbb{E}_{t+}[\chi\{x_{t+1} \in B_R^\infty(0)^c\} \| \sum_{j=1}^{d} \tilde{f}_{t+1,j} \left( \frac{\eta_\ell}{|\mathcal{S}^t|} \sum_{i \in \mathcal{S}^t} \sum_{\ell=1}^{K} \left( \nabla f(x_{i,\ell-1}^t)_j + (\xi_{i,\ell-1}^t)_j \right) \right) \|^2].$$

The integrand on the final line is non-negatively lower bounded given $x_{t+1} \in B_R^\infty(0)^c$ by

$$
\left( c \sum_{j=1}^d \left| \frac{\eta_\ell}{|\mathcal{S}^t|} \sum_{i \in \mathcal{S}^t} \left( \left( \sum_{\ell=1}^{K-1} \nabla f(x_{i,\ell-1}^t) + \xi_{i,\ell-1}^t \right) + \nabla f(x_{i,K-1}^t) \right)_j + \frac{\eta_\ell}{|\mathcal{S}^t|} \sum_{i \in \mathcal{S}^t} (\xi_{i,K-1}^t)_j \pm R_j \right| \right)^2
$$

$$
\geq \sum_{j=1}^d c^2 \left| \frac{\eta_\ell}{|\mathcal{S}^t|} \sum_{i \in \mathcal{S}^t} \left( \left( \sum_{\ell=1}^{K-1} \nabla f(x_{i,\ell-1}^t) + \xi_{i,\ell-1}^t \right) + \nabla f(x_{i,K-1}^t) \right)_j + \frac{\eta_\ell}{|\mathcal{S}^t|} \sum_{i \in \mathcal{S}^t} (\xi_{i,K-1}^t)_j \pm R_j \right|^2
$$

$$
\geq \sum_{j=1}^d c^2 \left| \frac{\eta_\ell}{|\mathcal{S}^t|} \sum_{i \in \mathcal{S}^t} \left( \left( \sum_{\ell=1}^{K-1} \nabla f(x_{i,\ell-1}^t) + \xi_{i,\ell-1}^t \right) + \nabla f(x_{i,K-1}^t) \right)_j \pm R_j \right|^2
$$

$$
+ 2 \sum_{j=1}^d c^2 \left\langle \frac{\eta_\ell}{|\mathcal{S}^t|} \sum_{i \in \mathcal{S}^t} \left( \left( \sum_{\ell=1}^{K-1} \nabla f(x_{i,\ell-1}^t) + \xi_{i,\ell-1}^t \right) + \nabla f(x_{i,K-1}^t) \right)_j \pm R_j, \frac{\eta_\ell}{|\mathcal{S}^t|} \sum_{i \in \mathcal{S}^t} (\xi_{i,K-1}^t)_j \right\rangle
$$

$$
+ \sum_{j=1}^d \frac{c^2 \eta_\ell^2}{|\mathcal{S}^t|^2} \left( \sum_{i \in \mathcal{S}^t} (\xi_{i,K-1}^t)_j \right)^2
$$

where $c = \min\{|\tilde{a}|, |\tilde{b}|\}$. The sign on $R_j$ is determined by the sign of the value $(x_{t+1})_j$ and equation (6).

Clearly, there exists compact intervals $[\bar{a}_{i,j}, \bar{b}_{i,j}]$ such that with non-zero probability, $(\xi_{i,K-1}^t)_j \in [\bar{a}_{i,j}, \bar{b}_{i,j}]$. For the setminus operation subtracting only one selection of client $i$ from the multiset $\mathcal{S}^t$ and $1 \in S^t$ being the heavy-tailed client, let $\hat{R}$ be equal to

$$
\frac{|\mathcal{S}^t|}{\eta_\ell} \left( |R| + \max_{i,j} \left( \frac{\eta_\ell \max\{|\bar{a}_{i,j}|, |\bar{b}_{i,j}|\}}{|\mathcal{S}^t|} + \left| \frac{\eta_\ell}{|\mathcal{S}^t|} \sum_{\tilde{i} \in \mathcal{S}^t} \left( \left( \sum_{\ell=1}^{K-1} \nabla f(x_{\tilde{i},\ell-1}^t) + \xi_{\tilde{i},\ell-1}^t \right) + \nabla f(x_{\tilde{i},K-1}^t) \right)_j \right| \right) \right).
$$

Then as

$$
\chi\{x_{t+1} \in B_R^\infty(0)^c\} \geq \chi\{x_{t+1} \in B_R^\infty(0)^c\} \Pi_{i \in S^t \setminus \{1\}} \chi\{(\xi_{i,K-1}^t)_j \in [\bar{a}_{i,j}, \bar{b}_{i,j}]\}
$$
$$
\geq \chi_j^+ := \chi\{|(\xi_{1,K-1}^t)_j| > \hat{R}\} \Pi_{i \in S^t \setminus \{1\}} \chi\{(\xi_{i,K-1}^t)_j \in [\bar{a}_{i,j}, \bar{b}_{i,j}]\},
$$

we may conclude

$$
\mathbb{E}_{t+}[\|f_{t+1}(x_{t+1})\|^2] \geq \mathbb{E}_{t+} \left[ \chi_j^+ \| \sum_{j=1}^d \tilde{f}_{t+1,j} \left( \frac{\eta_\ell}{|\mathcal{S}^t|} \sum_{i \in \mathcal{S}^t} \sum_{\ell=1}^K \left( \nabla f(x_{i,\ell-1}^t)_j + (\xi_{i,\ell-1}^t)_j \right) \right) \|^2 \right]
$$

$$
\geq \sum_{j=1}^d c^2 \mathbb{E}_{t+}[\chi_j^+] \left| \frac{\eta_\ell}{|\mathcal{S}^t|} \sum_{i \in \mathcal{S}^t} \left( \left( \sum_{\ell=1}^{K-1} \nabla f(x_{i,\ell-1}^t) + \xi_{i,\ell-1}^t \right) + \nabla f(x_{i,K-1}^t) \right)_j \pm R_j \right|^2
$$

$$
+ 2 \sum_{j=1}^d c^2 \left\langle \frac{\eta_\ell}{|\mathcal{S}^t|} \sum_{i \in \mathcal{S}^t} \left( \left( \sum_{\ell=1}^{K-1} \nabla f(x_{i,\ell-1}^t) + \xi_{i,\ell-1}^t \right) + \nabla f(x_{i,K-1}^t) \right)_j \pm R_j, \frac{\eta_\ell}{|\mathcal{S}^t|} \sum_{i \in \mathcal{S}^t} \mathbb{E}_{t+}[\chi_j^+ (\xi_{i,K-1}^t)_j] \right\rangle
$$

$$
+ \sum_{j=1}^d \frac{c^2 \eta_\ell^2}{|\mathcal{S}^t|^2} \mathbb{E}_{t+} \left[ \left( \sum_{i \in \mathcal{S}^t} (\xi_{i,K-1}^t)_j \right)^2 \right]
$$

$$
\geq C_1(t^+) + \sum_{j=1}^d \frac{c^2 \eta_\ell^2}{|\mathcal{S}^t|^2} \mathbb{E}_{t+} \left[ \chi_j^+ \left( \sum_{i \in \mathcal{S}^t} (\xi_{i,K-1}^t)_j \right)^2 \right]
$$

Noting that

$$
\mathbb{E}_{t+}[\chi_j^+ (\xi_{i,K-1}^t)_j] = \int_{\bar{a}_{i,j}}^{\bar{b}_{i,j}} (\xi_{i,K-1}^t)_j \, dp(\xi_{i,K-1}^t),
$$

we deduce that the existence of $\mathbb{E}(\xi^t_{i,K-1})_j \in \mathbb{R}$ (from all noise having finite mean) enforces that $\mathbb{E}_{t+}[\chi^+_j(\xi^t_{i,K-1})_j]$ must also exist and be finite. Thus, $C_1(t^+)$ is finite and well-defined given $t^+$. It remains to analyze the final term

$$\sum_{j=1}^d \mathbb{E}_{t+}\left[\chi^+_j\left(\sum_{i\in\mathcal{S}^t}(\xi^t_{i,K-1})_j\right)^2\right] = \sum_{j=1}^d \mathbb{E}_{t+}\left[\chi^+_j \sum_{i\in\mathcal{S}^t}(\xi^t_{i,K-1})_j^2\right] + 2\mathbb{E}_{t+}\left[\chi^+_j\sum_{i_1<i_2}(\xi^t_{i_1,K-1})_j(\xi^t_{i_2,K-1})_j\right]$$

$$= \sum_{j=1}^d\sum_{i\in\mathcal{S}^t}\mathbb{E}_{t+}\left[\chi^+_j(\xi^t_{i,K-1})_j^2\right] + 2\sum_{i_1<i_2}\mathbb{E}_{t+}\left[\chi^+_j(\xi^t_{i_1,K-1})_j\right]\mathbb{E}_{t+}\left[\chi^+_j(\xi^t_{i_2,K-1})_j\right]$$

where we used the independence of $\xi^t_{i,\ell}$ which is preserved across coordinate projections. Finally, note that for $C_2 := \min_{j\in[d]}\Pi_{i\in S^t\setminus\{1\}}\mathbb{P}((\xi^t_{i,K-1})_j \in [\bar{a}_{i,j}, \bar{b}_{i,j}]) \neq 0$, we have

$$\sum_{j=1}^d\sum_{i\in\mathcal{S}^t}\mathbb{E}_{t+}\left[\chi^+_j(\xi^t_{i,K-1})_j^2\right] \geq C_2\sum_{j=1}^d\int_{|(\xi^t_{1,K-1})_j|>\hat{R}}\|(\xi^t_{1,K-1})_j\|^2\,\mathrm{d}p(\xi^t_{1,K-1})$$

$$\geq C_2\int_{\|(\xi^t_{1,K-1})\|_\infty>\hat{R}}\|\xi^t_{1,K-1}\|^2\,\mathrm{d}p(\xi^t_{1,K-1}) = \infty.$$

Thus, we have as before

$$\mathbb{E}_{t+}[\|f_{t+1}(x_{t+1})\|^2] \geq \infty.$$

As the variance is well-defined, we conclude that $\mathbb{E}[\|f_{t+1}(x_{t+1})\|^2] = \infty$ under the tower law of expectation. $\qquad\square$

For federated learning, we typically have $f_t(x) \equiv f(x)$ for all $t = 1,\ldots,T$. We saw from Proposition A.1 that inserting local adaptivity successfully breaks the generality of remorse and heavy-tailed resent for FedAvg. A high-level, intuitive overview is that client-side AdaGrad clips the local updates of each iteration, which mollifies the impact of stochasticity in perturbing the weight updates. We present the following proposition, formulated loosely without utilizing any advantages provided via local adaptivity except for clipping which leaves room for far sharper generalization. For this reason, we view local adaptive methods to be more desirable than clipped SGD in large-scale applications, if memory and computation constraints of the clients can be addressed.

**Proposition A.6.** *Let $f_t \in C(\mathbb{R}^d)$ for $t = 1,\ldots,T$ for $f_t$ not necessarily convex. Introducing client-side adaptivity via AdaGrad into the setting in Theorem 3.4 produces a non-remorseful and a non-resentful algorithm.*

*Proof.* By Jensen, we have that $\|\mathbb{E}f(x_t)\| \leq \mathbb{E}\|f(x_t)\|$. Thus, it is enough to show $\mathbb{E}\|f(x_t)\| < \infty$ which guarantees that the $t$-th regret update $\mathbb{E}[f_t(x_t)] - f_t(x^*)$ is finite for any $x^*$ arbitrarily fixed. However, this is immediate as $x_t \in B_{Kt}(x_0)$, where $K$ is the number of local SGD iterations prior to server synchronization. Thus, by the extremal value theorem, there exists an $M \in \mathbb{R}_{\geq 0}$ such that

$$0 \leq \mathbb{E}\|f(x_t)\| \leq \mathbb{E}[M] < \infty.$$

Similarly, we may also show that the variance $\mathbb{E}\|f(x_t)\|^2 < \infty$. $\qquad\square$

# B. SM3 with Delayed Preconditioner Updates

We now present a description of SM3-I/II with delayed preconditioner updates as Algorithms 2 and 3. SM3-II capitalizes on a tighter approximation of the second moment, and empirically demonstrates better results. We have opted to implement a smoothing term $\varepsilon$ instead of treating any zero denominator as zero as done in the original work. We provide the analysis for SM3-II which generalizes the analysis for SM3-I.

To enhance clarity, we present several lemmas before giving the proof of Theorem 5.1. Note that Lemma B.1 is written in broadcasting notation, where the scalars in the right hand side have $\mathbf{1} \in \mathbb{R}^d$ implicitly multiplied and the inequality holds coordinatewise. For notational convenience, we will view $\Phi^K_1$, $\Phi^K_2$ as vectors.

**Lemma B.1.** *Under Algorithm 1, $|\Delta^t_i|$ is bounded by*

$$|\Delta^t_i| \leq \Phi^K_1 := \eta_\ell\left(\sqrt{\left\lceil\frac{K}{z}\right\rceil}\cdot\log^{\frac{1}{2}}\left(1+\frac{\lceil\frac{K}{z}\rceil G^2}{\varepsilon^2}\right) + \frac{\eta_\ell(K-\lceil\frac{K}{z}\rceil)G}{\sqrt{v_0}+\varepsilon}\right).$$

---

**Algorithm 2** Delayed preconditioner SM3-I

---

**Require:** Client learning rate $\eta_\ell$, step delay $z \in \mathbb{Z}_{\geq 1}$, and $\varepsilon$-smoothing term $\varepsilon > 0$

**Require:** A full cover $\{S_1, \ldots, S_k\} \subset \mathcal{P}([d])$ where $\bigcup_{\ell=1}^k S_\ell = \{1, \ldots, d\}$

  1: **Initialize:** $x_1 = 0$ and $\mu_0(r) = 0$ for $\forall r \in \{1, \ldots, k\}$

  2: **for** $t = 1, \ldots, K$ **do**

  3:      $g_t \leftarrow \nabla \ell(x_t)$

  4:      **if** $(t-1)/z \in \mathbb{Z}$ **then**

  5:         **for** $r = 1, \ldots, k$ **do**

  6:             $\mu_t(r) \leftarrow \mu_{t-1}(r) + \max_{j \in S_r} g_t^2(j)$

  7:         **end for**

  8:      **end if**

  9:      **for** $j = 1, \ldots, d$ **do**

10:         $\nu_t(j) \leftarrow \min_{r:S_r \ni j} \mu_t(r)$ (minimum taken over all $r$ such that $j \in S_r$)

11:         $x_{t+1}(j) \leftarrow x_t(j) - \frac{\eta_\ell g_t(j)}{\sqrt{\nu_t(j) + \varepsilon}}$

12:      **end for**

13: **end for**

---

**Algorithm 3** Delayed preconditioner SM3-II

---

**Require:** Client learning rate $\eta_\ell$, step delay $z \in \mathbb{Z}_{\geq 1}$, and $\varepsilon$-smoothing term $\varepsilon > 0$

**Require:** A full cover $\{S_1, \ldots, S_k\} \subset \mathcal{P}([d])$ where $\bigcup_{\ell=1}^k S_\ell = \{1, \ldots, d\}$

  1: **Initialize:** $x_1 = 0$ and $\mu_0'(r) = 0$ for $\forall r \in \{1, \ldots, k\}$

  2: **for** $t = 1, \ldots, K$ **do**

  3:      $g_t \leftarrow \nabla \ell(x_t)$

  4:      $\mu_t'(r) \leftarrow 0$ for $\forall r \in [k]$

  5:      **for** $j = 1, \ldots, d$ **do**

  6:         **if** $(t-1)/z \in \mathbb{Z}$ **then**

  7:             $\nu_t'(j) \leftarrow \min_{r:S_r \ni j} \mu_{t-1}'(r) + g_t^2(j)$

  8:             **for all** $r : S_r \ni j$ **do**

  9:                 set $\mu_t'(r) \leftarrow \max\{\mu_t'(r), \nu_t'(j)\}$

10:             **end for**

11:         **else**

12:             $\nu_t'(j) \leftarrow \nu_{t-1}'(j)$

13:         **end if**

14:         $x_{t+1}(j) \leftarrow x_t(j) - \frac{\eta_\ell g_t(j)}{\sqrt{\nu_t'(j) + \varepsilon}}$

15:      **end for**

16: **end for**

---

*Proof.* Forming a bound for the pseudogradients is not trivial due to delayed preconditioner updates. We begin by noting that delayed gradient updates are initiated at local timesteps $k = nz + 1$ for $n \in \mathbb{Z}_{\geq 0}$. We now split cases $k/z \notin \mathbb{Z}$ and $k/z \in \mathbb{Z}$. In the first case, there exists $n \in \mathbb{Z}_{\geq 0}$ such that $nz + 1 \leq k < (n+1)z$, and the latest preconditioner update by client step $k$ is given at timestep $(\lceil k/z \rceil - 1)z + 1 = \lfloor k/z \rfloor z + 1$. In the second case, if $z \neq 1$, then step $k$ is just one step shy of a preconditioner update. The latest update is therefore held at step $(\lceil k/z \rceil - 1)z + 1$ which is no longer identical to $\lfloor k/z \rfloor z + 1$.

With this observation, it is easy to show by induction that

$$v_k(j) \geq v_0(j) + \sum_{\ell=1}^{\lceil \frac{k}{z} \rceil} \left( g_{i,(\ell-1)z+1}^t(j) \right)^2 \quad \text{for} \quad j \in \{1, \ldots, d\} \quad \text{and} \quad k \in \{1, \ldots, K\}.$$

Recall that $\Delta_t = 1/|\mathcal{S}^t| \sum_{i \in \mathcal{S}^t} \Delta_i^t$ and $\Delta_i^t = x_{i,K}^t - x_{i,0}^t$. By telescoping for $K$ local steps and the definition of gradient updates in AdaSquare-SM3, we obtain

$$|\Delta_i^t| = \left| \sum_{p=1}^K \eta_\ell \frac{m_p}{\sqrt{v_p} + \varepsilon} \right| \leq \eta_\ell \sum_{p=1}^K \frac{|g_{i,p}^t|}{\sqrt{v_0 + \sum_{r=1}^{\lceil \frac{p}{z} \rceil} (g_{i,(r-1)z+1}^t)^2} + \varepsilon}$$

For $\mathcal{F} = \{0, 1, \ldots, \lceil K/z \rceil - 1\}z + 1$, we thus have that

$$|\Delta_i^t| \leq \eta_\ell \sum_{p \in \mathcal{F}} \frac{|g_{i,p}^t|}{\sqrt{v_0 + \sum_{r=1}^{\lceil \frac{p}{z} \rceil} (g_{i,(r-1)z+1}^t)^2} + \varepsilon}$$
$$+ \eta_\ell \sum_{p \in [K] \setminus \mathcal{F}} \frac{|g_{i,p}^t|}{\sqrt{v_0 + \sum_{r=1}^{\lceil \frac{p}{z} \rceil} (g_{i,(r-1)z+1}^t)^2} + \varepsilon}.$$

To obtain a deterministic bound, we cannot ignore the worst-case stochastic realization that $g_{i,(r-1)z+1}^t = 0$ for $\forall r \in [\lceil \frac{p}{z} \rceil]$, $p \in [K] \setminus \mathcal{F}$. Therefore, we form the upper bound (where $\sum_1^0 := 0$ by definition)

$$|\Delta_i^t| \leq \eta_\ell \underbrace{\sum_{p \in \mathcal{F}} \frac{|g_{i,p}^t|}{\sqrt{v_0 + |g_{i,p}^t|^2 + \sum_{r=1}^{\lceil \frac{p}{z} \rceil - 1} (g_{i,(r-1)z+1}^t)^2} + \varepsilon}}_{T_1} + \frac{\eta_\ell}{\sqrt{v_0} + \varepsilon} \left( \sum_{p \in [K] \setminus \mathcal{F}} |g_{i,p}^t| \right) \quad (7)$$
$$\leq \eta_\ell T_1 + \frac{\eta_\ell (K - \lceil \frac{K}{z} \rceil) G}{\sqrt{v_0} + \varepsilon}.$$

As $0$ is trivially bounded by any non-negative upper bound, we may without loss of generality assume that $g_{i,(r-1)z+1}^t \neq 0$ for at least one $r \in [\lceil \frac{p}{z} \rceil]$. We further bound $T_1$ as follows:

$$T_1 \leq \sum_{p \in \mathcal{F}} \frac{|g_{i,p}^t|}{\sqrt{|g_{i,p}^t|^2 + \sum_{r=1}^{\lceil \frac{p}{z} \rceil - 1} (g_{i,(r-1)z+1}^t)^2} + \varepsilon} \leq \sum_{p \in \mathcal{F}} \sqrt{\frac{|g_{i,p}^t|^2}{\varepsilon^2 + \sum_{r \in [p] \cap \mathcal{F}} |g_{i,r}^t|^2}}$$
$$\leq \sqrt{|\mathcal{F}|} \sqrt{\left( \sum_{p \in \mathcal{F}} \frac{|g_{i,p}^t|^2}{\varepsilon^2 + \sum_{r \in [p] \cap \mathcal{F}} |g_{i,r}^t|^2} \right)}$$
$$\leq \sqrt{\lceil \frac{K}{z} \rceil} \cdot \log^{\frac{1}{2}} \left( 1 + \sum_{p \in \mathcal{F}} \frac{|g_{i,p}^t|^2}{\varepsilon^2} \right)$$

Note the use of Cauchy Schwartz in the third inequality. A detailed proof of the log inequality used in the third line may be found as part of the proof of Theorem 5.1, equation (12) which uses similar techniques. By Assumption 2, we are done. $\square$

The server-side pseudogradient updates may also be bounded as follows.

**Lemma B.2.** *Under Algorithm 1, each server step size is bounded in absolute value by*

$$\Phi_2^K := \min\left\{\eta\sqrt{(1-\widetilde{\beta}_1)(1-\widetilde{\beta}_1^{2t})}, \frac{\eta}{\tau}\Phi_1^K\right\}.$$

*Proof.* Without loss of generality, we may let $\tau = 0$ when forming the first upper bound for expository purposes.

$$\eta\frac{|\widetilde{m}_t|}{\sqrt{\widetilde{v}_t}+\tau} \leq \frac{\eta(1-\widetilde{\beta}_1)\sum_{\ell=1}^{t}\widetilde{\beta}_1^{t-\ell}|\Delta_\ell|}{\sqrt{\sum_{\ell=1}^{t}\Delta_\ell^2+\tau^2}+\tau}$$

$$\leq \frac{\eta(1-\widetilde{\beta}_1)\left(\sum_{\ell=1}^{t}\widetilde{\beta}_1^{t-\ell}|\Delta_\ell|\right)\sqrt{\sum_{\ell=1}^{t}\widetilde{\beta}_1^{2t-2\ell}}}{\sqrt{\sum_{\ell=1}^{t}\Delta_\ell^2}\sqrt{\sum_{\ell=1}^{t}\widetilde{\beta}_1^{2t-2\ell}}}$$

$$\leq \eta\sqrt{1-\widetilde{\beta}_1}\sqrt{1-\widetilde{\beta}_1^2}\sqrt{\sum_{\ell=1}^{t}\widetilde{\beta}_1^{2t-2\ell}}$$

$$= \eta\sqrt{1-\widetilde{\beta}_1}\sqrt{1-\widetilde{\beta}_1^{2t}}.$$

Note that the final inequality is obtained using Cauchy-Schwartz, while the second bound in the lemma statement follows from the first inequality and Lemma B.1. $\square$

Finally, we form a loose upper bound for the gradient variance.

**Lemma B.3.** *For $k \in \{1, \ldots, K\}$, the uncentered variance estimate $v_k$ as well as $\mu_k$ in Algorithm 1 are bounded by*

$$(B1): \quad 0 \leq \mu_k(b) \leq dkG^2 \quad for \quad and \quad b \in \{1, \ldots, q\},$$
$$(B2): \quad 0 \leq v_k(j) \leq dkG^2 \quad for \quad j \in \{1, \ldots, d\}.$$

*Proof.* Non-negativity of the variance estimates $v_k$ is trivial and implies the non-negativity of $\mu_k$, thus we focus on the upper bound for which we use dual induction. The case $k = 1$ is satisfied by zero initialization. Assuming the inequality holds for $k \leftarrow k - 1$, we have for each $j$

$$v_k(j) = \min_{b:S_b \ni j}\mu_{k-1}(b) + \left(g_{i,k}^t(j)\right)^2 \leq d(k-1)G^2 + G^2 \leq dkG^2.$$

Now, $\mu_k$ is initialized to zero at the start of each step $k$ and its entries are increased while broadcasting over each coordinate $j \in \{1, \ldots, d\}$ by

$$\mu_k(b) \leftarrow \max\{\mu_k(b), v_k(j)\} \quad for \quad \forall b : j \in S_b.$$

For $j = 1$, it is clear that

$$\mu_k(b) \leftarrow v_k(j) \leq dkG^2 \quad for \quad \forall b \in \{1, \ldots, q\}.$$

For $j \geq 2$, inductively, we have

$$\mu_k(b) \leftarrow \max\{\mu_k(b), v_k(j)\} \leq dkG^2$$

as both arguments of the maximum function are upper bounded by $dkG^2$. This completes the proof. $\square$

## B.1. Precompact Convergence Analysis

We aim to analyze the convergence of learning algorithms under the general, non-convex setting. However, extremely popular and well known adaptive optimizers such as Adam whose efficacy is strongly supported by empirical evidence have been shown to fail to converge even for convex settings (Reddi et al., 2018). Therefore, recent works have investigated the asymptotic stabilization of gradients, instead of requiring strict convergence to local or global optima of the objective (Xie et al., 2019; Tong et al., 2020; Chen et al., 2020; Zhang et al., 2020; Reddi et al., 2021; Wang et al., 2022; Sun et al., 2023). Such convergence bounds are of the form $\min_t\|\nabla f(x_t)\| \leq \mathcal{O}(T^{-\alpha})$, and are interpreted via the following lemma:

**Lemma B.4.** *For $x_t$ the $t$-step parameters of any objective $f(x)$ learned by an algorithm, let $\min_{1 \leq t \leq T} \|\nabla f(x_t)\| \leq \mathcal{O}(T^{-\alpha})$ for $\alpha > 0$. Then, there exists a learning algorithm which outputs parameters $\{\widetilde{x}_1, \widetilde{x}_2, \dots\}$ such that $\|\nabla f(\widetilde{x}_t)\| \to 0$ as $t \to \infty$.*

*Proof.* Assuming otherwise gives that $\|\nabla f(x_t)\|$ is $\varepsilon$-bounded away from $0$ for some $\varepsilon > 0$, for any parameter $x_t$ realized by the algorithm. Clearly, $\min_{1 \leq t \leq T} \|\nabla F(x_t)\| \to 0$ as $T \to \infty$ gives a contradiction. More constructively, note that $\forall \varepsilon > 0, \exists \widetilde{T}(\varepsilon) \in \mathbb{N}$ such that $T \geq \widetilde{T}(\varepsilon) \implies \min_{1 \leq t \leq T} \|\nabla f(x_t)\| < \varepsilon$. Letting $\varepsilon = 1/n$ for $n \in \mathbb{N}$ and $T_n := \widetilde{T}(1/n)$, we have that there exists $t_n \in [T_n]$ such that $\|\nabla f(x_{t_n})\| < 1/n$. Letting $\widetilde{x}_i := x_{t_i}$ extracts the desired parameter sequence. $\qquad \square$

This notion of convergence can be formalized as *precompact convergence* which is consistent with sequence properties of precompact normed sets. In this paper, we explicitly formalize the conventions used in prior works, and take the term convergence to mean precompact convergence unless stated otherwise.

**Definition B.5** (Precompact convergence). A sequence $\{y_n\}_{n \in \mathbb{N}}$ in a normed space $\mathcal{Y}$ is said to converge precompactly to $y \in \mathcal{Y}$ if there exists $\varphi : \mathbb{N} \to \mathbb{N}$ such that $y_{\varphi(n)} \to y$.

Our goal is to develop principled federated algorithms whose global gradients are guaranteed to converge precompactly to $0$ regardless of parameter initialization, in the general, non-convex setting. Note that precompact convergence must allow for convergence to each element $y_n$ of the sequence. Now, we are ready to present the following theorem.

**Theorem 5.1.** *In Algorithm 1, we have that*

$$\min_{t \in [T]} \|\nabla f(x_{t-1})\|^2 \leq \frac{\Psi_1 + \Psi_2 + \Psi_3 + \Psi_4 + \Psi_5}{\Psi_6},$$

*where*

$$\Psi_1 = f(x_0) - f(x^*),$$

$$\Psi_2 = \frac{\eta^2 L T d \|\Phi_1^K\|^2}{\tau^2},$$

$$\Psi_3 = \frac{(1 - \widetilde{\beta}_1^T) \eta \eta_\ell K \widetilde{L} T \|\Phi_1^K\|^2}{\widetilde{\alpha}_1 \tau (\sqrt{v_0} + \varepsilon)^2},$$

$$\Psi_4 = \frac{(1 - \widetilde{\beta}_1) \eta \eta_\ell K L T c(\widetilde{\beta}_1) \|\Phi_2^K\|^2}{\widetilde{\alpha}_1 \tau (\sqrt{v_0} + \varepsilon)^2},$$

$$\Psi_5 = \frac{\eta d \|\Phi_1^K\| G \left(1 - \widetilde{\beta}_1 + \log\left(1 + \frac{T \|\Phi_1^K\|^2}{\tau^2}\right)\right)}{\tau},$$

$$\Psi_6 = \frac{3(1 - \widetilde{\beta}_1) \eta \widetilde{\gamma}_1 T}{4\left(\sqrt{T \|\Phi_1^K\|^2 + \widetilde{v}_0} + \tau\right)}.$$

*Here, the constant $c$ is defined with respect to $\widetilde{\beta}_1$ as*

$$c(\widetilde{\beta}_1) := \sum_{u=0}^{\widetilde{u}_0(\widetilde{\beta}_1)} \widetilde{\beta}_1^u u^2 + \int_{\widetilde{u}_0(\widetilde{\beta}_1)}^{\infty} \frac{1}{x^2} \mathrm{d}x \quad \text{for} \quad \widetilde{u}_0(\widetilde{\beta}_1) = \inf\{u \in \mathbb{N} : \widetilde{\beta}_1^v v^2 < \frac{1}{v^2} \text{ for } \forall v \geq u\}$$

*and the intermediary $\widetilde{\gamma}_1, \widetilde{\alpha}_1$ values are defined as*

$$\widetilde{\gamma}_1 := \eta_\ell \frac{K}{\sqrt{v_0 + dKG^2} + \varepsilon}, \quad \widetilde{\alpha}_1 := \frac{1}{2\sqrt{v_0 + dKG^2} + 2\varepsilon}.$$

*Proof.* To enhance readability, we use both coordinatewise and broadcasting notation, where a $[\cdot]_j$ subscript is attached for the $j$-th coordinate. In particular, the arguments are detailed mostly in the latter notation as it significantly clarifies the

intuitions behind the proof. By $L$-smoothness, we have

$$f(x_t) \leq f(x_{t-1}) + \langle \nabla f(x_{t-1}), x_t - x_{t-1} \rangle + \frac{L}{2} \|x_t - x_{t-1}\|^2$$

$$= f(x_{t-1}) + \eta \left\langle \nabla f(x_{t-1}), \frac{\widetilde{\beta}_1^t \widetilde{m}_0 + (1 - \widetilde{\beta}_1) \sum_{r=1}^t \widetilde{\beta}_1^{t-r} \Delta_r}{\sqrt{\widetilde{v}_t} + \tau} \right\rangle + \frac{\eta^2 L}{2} \left\| \frac{\widetilde{\beta}_1^t \widetilde{m}_0 + (1 - \widetilde{\beta}_1) \sum_{r=1}^t \widetilde{\beta}_1^{t-r} \Delta_r}{\sqrt{\widetilde{v}_t} + \tau} \right\|^2$$

$$= f(x_{t-1}) + \eta T_{0,0} + (1 - \widetilde{\beta}_1) \eta \sum_{r=1}^t T_{0,r} + \frac{\eta^2 L}{2} \left\| \frac{\widetilde{\beta}_1^t \widetilde{m}_0 + (1 - \widetilde{\beta}_1) \sum_{r=1}^t \widetilde{\beta}_1^{t-r} \Delta_r}{\sqrt{\widetilde{v}_t} + \tau} \right\|^2 \quad (8)$$

where for $r \in [t]$,

$$T_{0,r} = \widetilde{\beta}_1^{t-r} \left\langle \nabla f(x_{t-1}), \frac{\Delta_r}{\sqrt{\widetilde{v}_t} + \tau} \right\rangle \quad \text{and} \quad T_{0,0} = \left\langle \nabla f(x_{t-1}), \frac{\widetilde{\beta}_1^t \widetilde{m}_0}{\sqrt{\widetilde{v}_t} + \tau} \right\rangle. \quad (9)$$

Note that $T_{0,0}$ can only decay exponentially as training progresses, as $\sqrt{\widetilde{v}_t}$ is monotonically increasing with respect to $t$ and $\nabla f(x_{t-1})$ is coordinatewise bounded by $G$. We decompose $T_{0,r}$ further by

$$T_{0,r} = \underbrace{\widetilde{\beta}_1^{t-r} \left\langle \nabla f(x_{t-1}), \frac{\Delta_r}{\sqrt{\widetilde{v}_t} + \tau} - \frac{\Delta_r}{\sqrt{\widetilde{v}_{t-1}} + \tau} \right\rangle}_{T_{1,r}} + \underbrace{\widetilde{\beta}_1^{t-r} \left\langle \nabla f(x_{t-1}), \frac{\Delta_r}{\sqrt{\widetilde{v}_{t-1}} + \tau} \right\rangle}_{T_{2,r}}.$$

A bound for $T_{1,r}$ can be obtained as:

$$T_{1,r} = \widetilde{\beta}_1^{t-r} \left\langle \nabla f(x_{t-1}), \frac{\Delta_r(\sqrt{\widetilde{v}_{t-1}} - \sqrt{\widetilde{v}_t})}{(\sqrt{\widetilde{v}_t} + \tau)(\sqrt{\widetilde{v}_{t-1}} + \tau)} \right\rangle$$

$$= \widetilde{\beta}_1^{t-r} \left\langle \nabla f(x_{t-1}), \frac{-\Delta_r \Delta_t^2}{(\sqrt{\widetilde{v}_t} + \tau)(\sqrt{\widetilde{v}_{t-1}} + \tau)(\sqrt{\widetilde{v}_{t-1}} + \sqrt{\widetilde{v}_t})} \right\rangle$$

$$\leq \widetilde{\beta}_1^{t-r} \left\langle |\nabla f(x_{t-1})|, \frac{|\Delta_r| \Delta_t^2}{(\widetilde{v}_t + \tau^2)(\sqrt{\widetilde{v}_{t-1}} + \tau)} \right\rangle$$

$$\leq \widetilde{\beta}_1^{t-r} \sum_{j=1}^d G \left[ \frac{|\Delta_r| \Delta_t^2}{(\widetilde{v}_t + \tau^2)(\sqrt{\widetilde{v}_{t-1}} + \tau)} \right]_j$$

$$\leq \frac{\|\Phi_1^K\| G \widetilde{\beta}_1^{t-r}}{\tau} \sum_{j=1}^d \left[ \frac{\Delta_t^2}{\widetilde{v}_t} \right]_j.$$

Lemma E.2 is used to obtain the final inequality. For $T_{2,r}$, we apply a further decomposition for $\gamma_r > 0$ allowed to be arbitrary within a compact interval $\epsilon \eta_\ell$-bounded away from 0,

$$T_{2,r} = \underbrace{\widetilde{\beta}_1^{t-r} \left\langle \frac{\nabla f(x_{t-1})}{\sqrt{\widetilde{v}_{t-1}} + \tau}, \Delta_r + \gamma_r \nabla f(x_{t-1}) \right\rangle}_{T_{2,r}^1} - \gamma_r \widetilde{\beta}_1^{t-r} \left\| \frac{\nabla f(x_{t-1})}{\sqrt{\sqrt{\widetilde{v}_{t-1}} + \tau}} \right\|^2.$$

For expository purposes, we present the case in which local gradient clipping is not triggered. The analysis directly generalizes to the setting where clipping activates. Unraveling the definition of $\Delta_r$ gives

$$\Delta_r = \frac{-\eta_\ell}{|\mathcal{S}^r|} \sum_{i \in \mathcal{S}^r} \sum_{p=1}^K \frac{g_{i,p}^r}{\sqrt{v_{i,p}^r} + \varepsilon},$$

which intuits the following value

$$\gamma_r := \frac{\eta_\ell}{|\mathcal{S}^r|} \sum_{i \in \mathcal{S}^r} \sum_{p=1}^{K} \frac{1}{\sqrt{v_{i,p}^r} + \varepsilon}.$$

We have by Assumption 2 and Lemma B.3 that

$$\gamma_r \in [\widetilde{\gamma}_1, \widetilde{\gamma}_2] := \left[ \eta_\ell \sum_{p=1}^{K} \frac{1}{\sqrt{v_0 + dKG^2} + \varepsilon}, \frac{\eta_\ell K}{\sqrt{v_0} + \varepsilon} \right].$$

Expanding $T_{2,r}^1$ for $\alpha_r > 0$ to be fixed,

$$\widetilde{\beta}_1^{t-r} \left\langle \frac{\nabla f(x_{t-1})}{\sqrt{\widetilde{v}_{t-1}} + \tau}, \Delta_r + \gamma_r \nabla f(x_{t-1}) \right\rangle$$

$$= \frac{\widetilde{\beta}_1^{t-r}}{|\mathcal{S}^r|} \sum_{i \in \mathcal{S}^r} \sum_{p=1}^{K} \left\langle \frac{\nabla f(x_{t-1})}{\sqrt{\widetilde{v}_{t-1}} + \tau}, \frac{\eta_\ell \left( \nabla f(x_{t-1}) - g_{i,p}^r \right)}{\sqrt{v_p} + \varepsilon} \right\rangle$$

$$\leq \frac{\eta_\ell \widetilde{\beta}_1^{t-r} \alpha_r K}{2|\mathcal{S}^r|} \sum_{i \in \mathcal{S}^r} \left\| \frac{\nabla f(x_{t-1})}{\sqrt{\sqrt{\widetilde{v}_{t-1}} + \tau}} \right\|^2$$

$$+ \frac{\eta_\ell \widetilde{\beta}_1^{t-r}}{2|\mathcal{S}^r|\alpha_r} \sum_{i \in \mathcal{S}^r} \sum_{p=1}^{K} \left\| \frac{\left( \nabla f(x_{t-1}) - \nabla F_i(x_{i,p-1}^r) \right)}{\sqrt{\sqrt{\widetilde{v}_{t-1}} + \tau} \left( \sqrt{v_p} + \varepsilon \right)} \right\|^2$$

$$\leq \frac{\eta_\ell \widetilde{\beta}_1^{t-r} \alpha_r K}{2} \left\| \frac{\nabla f(x_{t-1})}{\sqrt{\sqrt{\widetilde{v}_{t-1}} + \tau}} \right\|^2$$

$$+ \frac{\eta_\ell \widetilde{\beta}_1^{t-r}}{2|\mathcal{S}^r|\alpha_r \tau (\sqrt{v_0} + \varepsilon)^2} \sum_{i \in \mathcal{S}^r} \sum_{p=1}^{K} \left\| \nabla f(x_{t-1}) - \nabla F_i(x_{i,p-1}^r) \right\|^2.$$

where in the first inequality we drew the deterministic gradient instead of accessing the stochastic sample via full gradient descent. The first term is controlled by setting

$$\alpha_r = \frac{\gamma_r}{2\eta_\ell K} \in [\widetilde{\alpha}_1, \widetilde{\alpha}_2] := \left[ \frac{1}{2\sqrt{v_0 + dKG^2} + 2\varepsilon}, \frac{1}{2\sqrt{v_0} + 2\varepsilon} \right].$$

We aim to bound the second term via majorization and telescoping arguments. We have by $L$-smoothness, Lemmas B.1, B.2, and Assumption 2 that

$$\left\| \nabla f(x_{t-1}) - \nabla F_i(x_{i,p-1}^r) \right\|^2 \leq \frac{1}{N} \sum_{i' \in [N]} \left\| \left( \nabla F_{i'}(x_{t-1}) - \nabla F_i(x_{i,p-1}^r) \right) \right\|^2$$

$$= \frac{1}{N} \sum_{i' \in [N]} \left\| \left( \nabla F_{i'}(x_{t-1}) - \nabla F_{i'}(x_{r-1}) + \nabla F_{i'}(x_{r-1}) - \nabla F_i(x_{i,p-1}^r) \right) \right\|^2$$

$$\leq \frac{2}{N} \sum_{i' \in [N]} \left( \left\| \nabla F_{i'}(x_{t-1}) - \nabla F_{i'}(x_{r-1}) \right\|^2 + \left\| \nabla F_{i'}(x_{r-1}) - \nabla F_i(x_{i,p-1}^r) \right\|^2 \right)$$

$$\leq \frac{2L}{N} \sum_{i' \in [N]} \left\| x_{t-1} - x_{r-1} \right\|^2 + \frac{2\widetilde{L}}{N} \sum_{i' \in [N]} \left\| x_{i,p-1}^r - x_{i,0}^r \right\|^2$$

$$= 2L \left\| x_{t-1} - x_{r-1} \right\|^2 + 2\widetilde{L} \left\| x_{i,p-1}^r - x_{i,0}^r \right\|^2$$

$$\leq 2L(t-r) \sum_{o=r}^{t-1} \left\| x_o - x_{o-1} \right\|^2 + 2\widetilde{L} \| \Phi_1^p \|^2$$

$$\leq 2L(t-r)^2 \| \Phi_2^K \|^2 + 2\widetilde{L} \| \Phi_1^K \|^2.$$

Note that the first inequality was obtained by Jensen, while the third inequality uses that the client weights $x_{i,0}^r$ are synchronized to the global weights $x_{r-1}$ for $\forall i \in [N]$ at the start of training. Now, we have

$$\frac{\eta_\ell \widetilde{\beta}_1^{t-r}}{2|\mathcal{S}^r|\alpha_r\tau(\sqrt{v_0}+\varepsilon)^2} \sum_{i\in\mathcal{S}^r}\sum_{p=1}^K \left(2L(t-r)^2\|\Phi_2^K\|^2 + 2\widetilde{L}\|\Phi_1^K\|^2\right)$$

$$\leq \frac{\eta_\ell\widetilde{\beta}_1^{t-r}KL(t-r)^2\|\Phi_2^K\|^2}{\alpha_r\tau(\sqrt{v_0}+\varepsilon)^2} + \frac{\eta_\ell\widetilde{\beta}_1^{t-r}\widetilde{L}K\|\Phi_1^K\|^2}{\alpha_r\tau(\sqrt{v_0}+\varepsilon)^2}.$$

Collecting terms gathered thus far gives

$$(1-\widetilde{\beta}_1)\eta\sum_{r=1}^t T_{0,r} \leq (1-\widetilde{\beta}_1)\eta\sum_{r=1}^t \left(\frac{\|\Phi_1^K\|G\widetilde{\beta}_1^{t-r}}{\tau}\sum_{j=1}^d\left[\frac{\Delta_t^2}{\widetilde{v}_t}\right]_j - \frac{3\gamma_r\widetilde{\beta}_1^{t-r}}{4}\left\|\frac{\nabla f(x_{t-1})}{\sqrt{\sqrt{\widetilde{v}_{t-1}}+\tau}}\right\|^2\right)$$

$$+ (1-\widetilde{\beta}_1)\eta\sum_{r=1}^t\left(\frac{\eta_\ell\widetilde{\beta}_1^{t-r}KL(t-r)^2\|\Phi_2^K\|^2}{\alpha_r\tau(\sqrt{v_0}+\varepsilon)^2} + \frac{\eta_\ell\widetilde{\beta}_1^{t-r}\widetilde{L}K\|\Phi_1^K\|^2}{\alpha_r\tau(\sqrt{v_0}+\varepsilon)^2}\right).$$

Now, let us bound the final term in equation (8),

$$\left\|\frac{\widetilde{\beta}_1^t\widetilde{m}_0 + (1-\widetilde{\beta}_1)\sum_{r=1}^t\widetilde{\beta}_1^{t-r}\Delta_r}{\sqrt{\widetilde{v}_t}+\tau}\right\|^2 \leq 2\left\|\frac{\widetilde{\beta}_1^t\widetilde{m}_0}{\sqrt{\widetilde{v}_t}+\tau}\right\|^2 + 2\left\|\frac{(1-\widetilde{\beta}_1)\sum_{r=1}^t\widetilde{\beta}_1^{t-r}\Delta_r}{\sqrt{\widetilde{v}_t}+\tau}\right\|^2$$

$$\leq 2\left\|\frac{\widetilde{\beta}_1^t\widetilde{m}_0}{\sqrt{\widetilde{v}_t}+\tau}\right\|^2 + 2\left\|\frac{(1-\widetilde{\beta}_1)\sum_{r=1}^t\widetilde{\beta}_1^{t-r}\max_{r\in[t]}|\Delta_r|}{\sqrt{\widetilde{v}_t}+\tau}\right\|^2$$

$$\leq 2\left\|\frac{\widetilde{\beta}_1^t\widetilde{m}_0}{\sqrt{\widetilde{v}_t}+\tau}\right\|^2 + 2\left\|\frac{(1-\widetilde{\beta}_1^t)}{\sqrt{\widetilde{v}_t}+\tau}\right\|^2\|\Phi_1^K\|^2$$

$$\leq 2\left\|\frac{\widetilde{\beta}_1^t\widetilde{m}_0}{\sqrt{\widetilde{v}_t}+\tau}\right\|^2 + 2d\frac{\|\Phi_1^K\|^2}{\tau^2}.$$

Substituting into equation (8) gives that

$$f(x_t) \leq f(x_{t-1}) + \eta T_{0,0} + \eta^2 L\left\|\frac{\widetilde{\beta}_1^t\widetilde{m}_0}{\sqrt{\widetilde{v}_t}+\tau}\right\|^2 + \frac{\eta^2 Ld\|\Phi_1^K\|^2}{\tau^2} + (1-\widetilde{\beta}_1)\eta\sum_{r=1}^t\left(\frac{\|\Phi_1^K\|G\widetilde{\beta}_1^{t-r}}{\tau}\sum_{j=1}^d\left[\frac{\Delta_t^2}{\widetilde{v}_t}\right]_j\right)$$

$$+ (1-\widetilde{\beta}_1)\eta\sum_{r=1}^t\left(\frac{\eta_\ell\widetilde{\beta}_1^{t-r}KL(t-r)^2\|\Phi_2^K\|^2}{\alpha_r\tau(\sqrt{v_0}+\varepsilon)^2} + \frac{\eta_\ell\widetilde{\beta}_1^{t-r}\widetilde{L}K\|\Phi_1^K\|^2}{\alpha_r\tau(\sqrt{v_0}+\varepsilon)^2}\right)$$

$$+ (1-\widetilde{\beta}_1)\eta\sum_{r=1}^t\left(-\frac{3\gamma_r\widetilde{\beta}_1^{t-r}}{4}\left\|\frac{\nabla f(x_{t-1})}{\sqrt{\sqrt{\widetilde{v}_{t-1}}+\tau}}\right\|^2\right). \tag{10}$$

Note that the exponential decay caused by $\widetilde{\beta}_1$ in the third term will expectedly dominate the effect of first order moment initialization $\widetilde{m}_0$ as training progresses, and summation over $t \in [T]$ gives $\mathcal{O}(1)$. We initialize $\widetilde{m}_0 \leftarrow 0$ to further simplify the equations. We also further exacerbate the upper bound by substituting $\widetilde{\gamma}_1, \widetilde{\alpha}_1$ into $\gamma_r, \alpha_r$ respectively, which achieves independence from $r$. Telescoping equation (10) then gives

$$\frac{3(1-\widetilde{\beta}_1)\eta\widetilde{\gamma}_1}{4}\sum_{t=1}^T\sum_{r=1}^t\widetilde{\beta}_1^{t-r}\left\|\frac{\nabla f(x_{t-1})}{\sqrt{\sqrt{\widetilde{v}_{t-1}}+\tau}}\right\|^2 \leq f(x_0) - f(x^*) + \frac{(1-\widetilde{\beta}_1)\eta\|\Phi_1^K\|G}{\tau}\sum_{t=1}^T\sum_{r=1}^t\sum_{j=1}^d\widetilde{\beta}_1^{t-r}\left[\frac{\Delta_t^2}{\widetilde{v}_t}\right]_j$$

$$+ \frac{\eta^2 LTd\|\Phi_1^K\|^2}{\tau^2} + \frac{(1-\widetilde{\beta}_1)\eta\eta_\ell K}{\widetilde{\alpha}_1\tau(\sqrt{v_0}+\varepsilon)^2}\sum_{t=1}^T\sum_{r=1}^t\left(L\widetilde{\beta}_1^{t-r}(t-r)^2\|\Phi_2^K\|^2 + \widetilde{L}\widetilde{\beta}_1^{t-r}\|\Phi_1^K\|^2\right). \tag{11}$$

To complete the proof, we aim to ease a logarithm out from the third term on the right hand side. For this purpose, we induce a recursion with a log bound

$$(1 - \widetilde{\beta}_1) \sum_{t=1}^{T} \sum_{r=1}^{t} \widetilde{\beta}_1^{t-r} \frac{\Delta_{t,j}^2}{\sum_{\ell=1}^{t} \Delta_{\ell,j}^2 + \tau^2} \leq \sum_{t=1}^{T} (1 - \widetilde{\beta}_1^t) \frac{\Delta_{t,j}^2}{\sum_{\ell=1}^{t} \Delta_{\ell,j}^2 + \tau^2}$$

$$\leq a_T + c_T \log\left(1 + b_T\right). \tag{12}$$

Setting $T = 1$ gives

$$(1 - \widetilde{\beta}_1) \frac{\Delta_{1,j}^2}{\Delta_{1,j}^2 + \tau^2} \leq a_1 + c_1 \log(1 + b_1),$$

and setting $a_T = 1 - \widetilde{\beta}_1$ satisfies this inequality (among other choices). Assuming formula (12) holds for $T$, let us explore the induction condition for $T + 1$, which is

$$\sum_{t=1}^{T} (1 - \widetilde{\beta}_1^t) \frac{\Delta_{t,j}^2}{\sum_{\ell=1}^{t} \Delta_{\ell,j}^2 + \tau^2} + (1 - \widetilde{\beta}_1^{T+1}) \frac{\Delta_{T+1,j}^2}{\sum_{\ell=1}^{T+1} \Delta_{\ell,j}^2 + \tau^2} \leq a_{T+1} + c_{T+1} \log\left(1 + b_{T+1}\right).$$

For simplicity, we impose that $c_t$ is a monotonically increasing non-negative sequence of $t$. We intend to contain the increase in the left hand side as $T$ grows in the log argument only, in the right hand side. Therefore, we select $a_{T+1} = a_T$. For a suitable choice of $b_{T+1}$ satisfying strong induction, it is enough to resolve

$$(1 - \widetilde{\beta}_1^{T+1}) \frac{\Delta_{T+1,j}^2}{\sum_{\ell=1}^{T+1} \Delta_{\ell,j}^2 + \tau^2} \leq c_{T+1} \log\left(\frac{1 + b_{T+1}}{1 + b_T}\right) = c_{T+1} \log\left(1 + \frac{b_{T+1} - b_T}{1 + b_T}\right).$$

Here, we used monotonicity of $c_t$. Noting that $\log(1 + x) \geq x/(1 + x)$, it is again enough to resolve

$$\frac{\Delta_{T+1,j}^2}{\sum_{\ell=1}^{T+1} \Delta_{\ell,j}^2 + \tau^2} \leq \frac{c_{T+1}(b_{T+1} - b_T)}{b_{T+1} + 1}$$

$$\iff \frac{\Delta_{T+1,j}^2}{\sum_{\ell=1}^{T+1} \Delta_{\ell,j}^2 + \tau^2} + c_{T+1} b_T \leq \left(c_{T+1} - \frac{\Delta_{T+1,j}^2}{\sum_{\ell=1}^{T+1} \Delta_{\ell,j}^2 + \tau^2}\right) b_{T+1}.$$

By positivity of $b_t$ for $t > 1$, a necessary condition is therefore that

$$c_{T+1} \geq \frac{\Delta_{T+1,j}^2}{\sum_{\ell=1}^{T+1} \Delta_{\ell,j}^2 + \tau^2}$$

In order to enhance the tightness of our bound, we choose the minimal permissible value $c_t = 1$ uniformly, which is attained as a suprema. In this setting, we are left with a recursion

$$\frac{\Delta_{T+1,j}^2}{\sum_{\ell=1}^{T+1} \Delta_{\ell,j}^2 + \tau^2} = \frac{b_{T+1} - b_T}{b_{T+1} + 1},$$

and collecting the terms in the form $b_{T+1} = b_T \omega_1(\Delta) + \omega_2(\Delta)$ would provide an optimal recursive bound given our simplifying assumptions, starting with $b_1 = 0$. A less optimal but simpler bound can be formed by selecting $b_{T+1} = b_T + \Delta_{T+1,j}^2/\tau^2$ for $b_1 = \Delta_{1,j}^2/\tau^2$. Therefore, we arrive at

$$(1 - \widetilde{\beta}_1) \sum_{t=1}^{T} \sum_{r=1}^{t} \widetilde{\beta}_1^{t-r} \frac{\Delta_{t,j}^2}{\sum_{\ell=1}^{t} \Delta_{\ell,j}^2 + \tau^2} \leq 1 - \widetilde{\beta}_1 + \log\left(1 + \sum_{\ell=1}^{T} \left(\frac{\Delta_{\ell,j}}{\tau}\right)^2\right)$$

$$\leq 1 - \widetilde{\beta}_1 + \log\left(1 + \frac{T\|\Phi_1^K\|^2}{\tau^2}\right). \tag{13}$$

The remaining term to be bounded in equation (11) is given

$$\frac{(1 - \widetilde{\beta}_1)\eta\eta_\ell KL}{\widetilde{\alpha}_1 \tau(\sqrt{v_0} + \varepsilon)^2} \sum_{t=1}^{T} \sum_{r=1}^{t} \left(\widetilde{\beta}_1^{t-r}(t - r)^2 \|\Phi_2^K\|^2\right).$$

The trick is to notice that the explosion of the series caused by double summation is culled selectively in reverse chronological order by the exponential, rendering the tail end asymptotically vacuous. Note that $(1 - \widetilde{\beta}_1)$ stabilizes the divergence as $\widetilde{\beta}_1 \to 1^-$ in the limit. By a change of variable $u = t - r$,

$$(1 - \widetilde{\beta}_1) \sum_{t=1}^{T} \sum_{r=1}^{t} \widetilde{\beta}_1^{t-r} (t-r)^2 = (1 - \widetilde{\beta}_1) \sum_{u=0}^{T-1} \widetilde{\beta}_1^u u^2 (T-u).$$

Defining

$$\widetilde{u}_0(\widetilde{\beta}_1) = \inf\{u \in \mathbb{N} : \widetilde{\beta}_1^v v^2 < \frac{1}{v^2} \text{ for } \forall v \geq u\},$$

let

$$c(\widetilde{\beta}_1) := \sum_{u=0}^{\widetilde{u}_0(\widetilde{\beta}_1)} \widetilde{\beta}_1^u u^2 + \int_{\widetilde{u}_0(\widetilde{\beta}_1)}^{\infty} \frac{1}{x^2} \mathrm{d}x.$$

Then, I claim that

$$(1 - \widetilde{\beta}_1) \sum_{t=1}^{T} \sum_{r=1}^{t} \widetilde{\beta}_1^{t-r} (t-r)^2 \leq (1 - \widetilde{\beta}_1) c(\widetilde{\beta}_1) T.$$

We prove this by induction. The case $T = 1$ is trivial. Now, assume the desired inequality holds until $T$. For $T + 1$, we want to show

$$(1 - \widetilde{\beta}_1) \sum_{u=0}^{T} \widetilde{\beta}_1^u u^2 (T - u + 1) \leq (1 - \widetilde{\beta}_1) c(\widetilde{\beta}_1)(T + 1)$$

$$\iff (1 - \widetilde{\beta}_1) \sum_{u=0}^{T-1} \widetilde{\beta}_1^u u^2 (T - u) + (1 - \widetilde{\beta}_1) \sum_{u=0}^{T} \widetilde{\beta}_1^u u^2 \leq (1 - \widetilde{\beta}_1) c(\widetilde{\beta}_1)(T + 1)$$

and thus by the inductive hypothesis it is enough to show

$$\sum_{u=0}^{T} \widetilde{\beta}_1^u u^2 \leq c(\widetilde{\beta}_1).$$

However, this is trivial by the definition of $c(\widetilde{\beta}_1)$. Upon substitution into equation (11) and noting that

$$\frac{3(1 - \widetilde{\beta}_1)\eta\widetilde{\gamma}_1}{4} \sum_{t=1}^{T} \sum_{r=1}^{t} \widetilde{\beta}_1^{t-r} \left\| \frac{\nabla f(x_{t-1})}{\sqrt{\sqrt{\widetilde{v}_{t-1}} + \tau}} \right\|^2 \geq \frac{3(1 - \widetilde{\beta}_1)\eta\widetilde{\gamma}_1 T}{4 \left( \sqrt{T\|\Phi_1^K\|^2 + \widetilde{v}_0} + \tau \right)} \min_{t \in [T]} \|\nabla f(x_{t-1})\|^2$$

we simplify as

$$\frac{3(1 - \widetilde{\beta}_1)\eta\widetilde{\gamma}_1 T}{4 \left( \sqrt{T\|\Phi_1^K\|^2 + \widetilde{v}_0} + \tau \right)} \min_{t \in [T]} \|\nabla f(x_{t-1})\|^2 \leq f(x_0) - f(x^*) + \frac{\eta^2 L T d \|\Phi_1^K\|^2}{\tau^2}$$

$$+ \frac{(1 - \widetilde{\beta}_1^T)\eta\eta_\ell K T \widetilde{L} \|\Phi_1^K\|^2}{\widetilde{\alpha}_1 \tau (v_0 + \varepsilon)^2} + \frac{(1 - \widetilde{\beta}_1)\eta\eta_\ell K T L c(\widetilde{\beta}_1) \|\Phi_2^K\|^2}{\widetilde{\alpha}_1 \tau (v_0 + \varepsilon)^2} \tag{14}$$

$$+ \frac{\eta d \|\Phi_1^K\| G \left( 1 - \widetilde{\beta}_1 + \log \left( 1 + \frac{T\|\Phi_1^K\|^2}{\tau^2} \right) \right)}{\tau}$$

Therefore, we immediately conclude that

$$\min_{t \in [T]} \|\nabla f(x_{t-1})\|^2 \leq \frac{\Psi_1 + \Psi_2 + \Psi_3 + \Psi_4 + \Psi_5}{\Psi_6},$$

where

$$\Psi_1 = f(x_0) - f(x^*),$$

$$\Psi_2 = \frac{\eta^2 LTd\|\Phi_1^K\|^2}{\tau^2},$$

$$\Psi_3 = \frac{(1 - \widetilde{\beta}_1^T)\eta\eta_\ell K\widetilde{L}T\|\Phi_1^K\|^2}{\widetilde{\alpha}_1\tau(\sqrt{v_0} + \varepsilon)^2},$$

$$\Psi_4 = \frac{(1 - \widetilde{\beta}_1)\eta\eta_\ell KLTc(\widetilde{\beta}_1)\|\Phi_2^K\|^2}{\widetilde{\alpha}_1\tau(\sqrt{v_0} + \varepsilon)^2},$$

$$\Psi_5 = \frac{\eta d\|\Phi_1^K\|G\left(1 - \widetilde{\beta}_1 + \log\left(1 + \frac{T\|\Phi_1^K\|^2}{\tau^2}\right)\right)}{\tau},$$

$$\Psi_6 = \frac{3(1 - \widetilde{\beta}_1)\eta\widetilde{\gamma}_1 T}{4\left(\sqrt{T\|\Phi_1^K\|^2 + \widetilde{v}_0} + \tau\right)}.$$

Here, the constant $c$ is defined with respect to $\widetilde{\beta}_1$ as

$$c(\widetilde{\beta}_1) := \sum_{u=0}^{\widetilde{u}_0(\widetilde{\beta}_1)} \widetilde{\beta}_1^u u^2 + \int_{\widetilde{u}_0(\widetilde{\beta}_1)}^\infty \frac{1}{x^2}\mathrm{d}x \quad \text{for} \quad \widetilde{u}_0(\widetilde{\beta}_1) = \inf\{u \in \mathbb{N} : \widetilde{\beta}_1^v v^2 < \frac{1}{v^2} \text{ for } \forall v \geq u\}$$

and the intermediary $\widetilde{\gamma}_1, \widetilde{\alpha}_1$ values are defined as

$$\widetilde{\gamma}_1 := \eta_\ell \frac{K}{\sqrt{v_0 + dKG^2} + \varepsilon}, \quad \widetilde{\alpha}_1 := \frac{1}{2\sqrt{v_0 + dKG^2} + 2\varepsilon}.$$

This concludes the proof. $\qquad\square$

Note that we have also shown the following two useful lemmas:

**Lemma B.6.** *For $\widetilde{\beta}_1 \in [0, 1)$ and $T \in \mathbb{Z}_{\geq 0}$, let*

$$\widetilde{u}_0(\widetilde{\beta}_1) = \inf\{u \in \mathbb{N} : \widetilde{\beta}_1^v v^2 < \frac{1}{v^2} \text{ for } \forall v \geq u\},$$

*and*

$$c(\widetilde{\beta}_1) := \sum_{u=0}^{\widetilde{u}_0(\widetilde{\beta}_1)} \widetilde{\beta}_1^u u^2 + \int_{\widetilde{u}_0(\widetilde{\beta}_1)}^\infty \frac{1}{x^2}\mathrm{d}x.$$

*Then, we have that*

$$\sum_{t=1}^T\sum_{r=1}^t \widetilde{\beta}_1^{t-r}(t-r)^2 \leq c(\widetilde{\beta}_1)T.$$

**Lemma B.7.** *Let $\Delta_{\ell,j} \in \mathbb{R}$, $\widetilde{\beta}_1 \in [0, 1)$, and $T \in \mathbb{Z}_{\geq 0}$. Then,*

$$(1 - \widetilde{\beta}_1)\sum_{t=1}^T\sum_{r=1}^t \widetilde{\beta}_1^{t-r} \frac{\Delta_{t,j}^2}{\sum_{\ell=1}^t \Delta_{\ell,j}^2 + \tau^2} \leq 1 - \widetilde{\beta}_1 + \log\left(1 + \frac{T\|\Phi_1^K\|^2}{\tau^2}\right).$$

We present the following corollary.

**Corollary B.8.** *Any of the following conditions are sufficient to ensure convergence of Algorithm 1:*

$$(A): \quad \eta_\ell \leq \mathcal{O}(T^{-1/2}) \quad \text{for} \quad \Omega(T^{-1}) < \eta\eta_\ell < \mathcal{O}(1),$$

$$(B): \quad \eta_\ell = \Theta(T^{-\frac{49}{100}}) \quad \text{for} \quad \Omega(T^{-\frac{1}{2}}) < \eta < \mathcal{O}(T^{\frac{12}{25}}).$$

*Proof.* The proof is formed by comparing orders of $T$. Recall that $\widetilde{\gamma}_1 = \Theta(\eta_\ell)$ and $\widetilde{L} = \Theta(\eta_\ell^{-1})$. As $\Phi_1^K = \Theta(\eta_\ell)$ and $\Phi_2^K = \Theta(\min\{\eta, \eta\eta_\ell\})$, we have for $\eta = \Theta(T^{p_1})$ and $\eta_\ell = \Theta(T^{p_2})$,

$$\psi_1 = \Theta(1)$$
$$\psi_2 = \eta^2\eta_\ell^2 T$$
$$\psi_3 = \eta\eta_\ell^2 T$$
$$\psi_4 = \begin{cases} \eta^3\eta_\ell^3 T & \text{if } \mathcal{O}(\eta_\ell) \leq \mathcal{O}(1) \\ \eta^3\eta_\ell T & \text{if } \Theta(\eta_\ell) > \Omega(1) \end{cases}$$
$$\psi_5 = \eta\eta_\ell \log(1 + T\eta_\ell^2)$$
$$\psi_6 = \begin{cases} \eta\eta_\ell T & \text{if } \mathcal{O}(T\eta_\ell^2) \leq \mathcal{O}(1) \\ \eta\sqrt{T} & \text{if } \Theta(T\eta_\ell^2) > \Omega(1) \end{cases}.$$

If $\mathcal{O}(T\eta_\ell^2) \leq \mathcal{O}(1)$, then $\mathcal{O}(\eta_\ell) \leq \mathcal{O}(1)$ which implies

$$\frac{\psi_1}{\psi_6} : (\eta\eta_\ell T)^{-1} = \Theta\left(T^{-(p_1+p_2+1)}\right)$$

$$\frac{\psi_2}{\psi_6} : \eta\eta_\ell = \Theta\left(T^{p_1+p_2}\right)$$

$$\frac{\psi_3}{\psi_6} : \eta_\ell = \Theta\left(T^{p_2}\right)$$

$$\frac{\psi_4}{\psi_6} : \eta^2\eta_\ell^2 = \Theta\left(T^{2p_1+2p_2}\right)$$

$$\frac{\psi_5}{\psi_6} : \frac{\log(1+T\eta_\ell^2)}{T} = \mathcal{O}(T^{-1})$$

This implies that we must have that $p_2 \leq -1/2$ and $-1 < p_1 + p_2 < 0$ for guaranteed convergence. Thus, $\eta_\ell \leq \mathcal{O}(T^{-1/2})$ such that $\Omega(T^{-1}) < \eta\eta_\ell < \mathcal{O}(1)$ is a sufficient condition. For instance, let $\eta_\ell = \Theta(T^{-1/2})$ and $\Omega(T^{-1/2}) < \eta < \mathcal{O}(T^{1/2})$.

Now, assume $\Theta(T\eta_\ell^2) > \Omega(1)$. If $\Theta(\eta_\ell) > \Omega(1)$, $\Psi_3/\Psi_6$ diverges. Therefore, let $\eta_\ell \leq \mathcal{O}(1)$. We have

$$\frac{\psi_1}{\psi_6} : (\eta\sqrt{T})^{-1} = \Theta(T^{-p_1-\frac{1}{2}})$$

$$\frac{\psi_2}{\psi_6} : \eta\eta_\ell^2\sqrt{T} = \Theta(T^{p_1+2p_2+\frac{1}{2}})$$

$$\frac{\psi_3}{\psi_6} : \eta_\ell^2\sqrt{T} = \Theta(T^{2p_2+\frac{1}{2}})$$

$$\frac{\psi_4}{\psi_6} : \eta^2\eta_\ell^3\sqrt{T} = \Theta(T^{2p_1+3p_2+\frac{1}{2}})$$

$$\frac{\psi_5}{\psi_6} : \frac{\eta_\ell \log(1+T\eta_\ell^2)}{\sqrt{T}} < \mathcal{O}(T^{-\frac{1}{2}+p_2})$$

Therefore, it suffices to satisfy

$$-\frac{1}{2} < p_2 \leq -\frac{1}{4}, \quad -\frac{1}{2} < p_1, \quad p_1 + 2p_2 < -\frac{1}{2}, \quad 2p_1 + 3p_2 < -\frac{1}{2}.$$

An example satisfying these conditions are

$$\eta_\ell = \Theta(T^{-\frac{49}{100}}), \quad \Omega(T^{-\frac{1}{2}}) < \eta < \mathcal{O}(T^{\frac{12}{25}}).$$

$\square$

Note that for all cases, $\eta_\ell$ must decay to establish convergence. However, striking a balance between local and global learning rates provably allows for greater than $\Omega(T^{1/3})$ divergence in the server learning rate without nullifying desirable convergence properties. This theoretically demonstrates the enhanced robustness properties of adaptive client-side federated learning algorithms to mitigate suboptimal choices of server learning rates.

**Corollary B.9.** *Algorithm 1 converges at rate $\mathcal{O}(T^{-1/2})$.*

*Proof.* If $\mathcal{O}(T\eta_\ell^2) \le \mathcal{O}(1)$, then we juxtapose $\psi_1/\psi_6$ and $\psi_2/\psi_6$. It is clear that the minimax value of the respective powers are attained at $p_1 + p_2 = -1/2$, realized by $p_2 = -1/2$ and $p_1 = 0$. In this case, clearly $\Theta(\psi_i/\psi_6) \le \mathcal{O}(T^{-1/2})$ for $1 \le i \le 5$. If $\Theta(T\eta_\ell^2) > \Omega(1)$, then our strategy should be to minimize $p_2$ due to positive coefficients in the powers $\psi_i/\psi_6$. Thus, let $p_2 = -1/2 + \varepsilon$ for $1 \gg \varepsilon > 0$. Then, the order of decay in $\psi_2/\psi_6$ is $p_1 - 1/2 + 2\varepsilon$, which is once again matched against $-p_1 - 1/2$, the power of $\psi_1/\psi_6$. Taking the limit $\varepsilon \to 0^+$, $\mathrm{minimax}\{p_1 - 1/2, -p_1 - 1/2\}$ for the range $-1/2 < p_1$ is attained at $p_1 = 0$. This sets the maximal decay rate to $\mathcal{O}(T^{-1/2})$ for the second case. $\square$

## B.2. Extension to Adam

The extension to the case where Adam is selected as the optimizer for the server, or for both the server and client is straightforward. We present the latter as it generalizes the former analysis. As in Lemma B.1, we have the following bound for the compressed SM3 estimates of the second moment,

$$v_k(j) \ge v_0(j) + \sum_{\ell=1}^{\lceil \frac{k}{z} \rceil} \left( g_{i,(\ell-1)z+1}^t(j) \right)^2 \quad \text{for} \quad j \in \{1, \ldots, d\} \quad \text{and} \quad k \in \{1, \ldots, K\},$$

which allows bounds to be established for the local and global pseudogradients following analogous logic as Lemmas B.2, D.2. As before, we arrive at equation (9) where due to exponential moving averaging on the server-side, we have

$$\widetilde{v}_t = \widetilde{\beta}_2^t \widetilde{v}_0 + (1 - \widetilde{\beta}_2) \sum_{\ell=1}^{t} \widetilde{\beta}_2^{t-r} \Delta_\ell.$$

Now, decompose $T_{0,r}$ as

$$T_{0,r} = \underbrace{\widetilde{\beta}_1^{t-r} \left\langle \nabla f(x_{t-1}), \frac{\Delta_r}{\sqrt{\widetilde{v}_t} + \tau} - \frac{\Delta_r}{\sqrt{\widetilde{\beta}_2 \widetilde{v}_{t-1}} + \tau} \right\rangle}_{T_{1,r}} + \underbrace{\widetilde{\beta}_1^{t-r} \left\langle \nabla f(x_{t-1}), \frac{\Delta_r}{\sqrt{\widetilde{\beta}_2 \widetilde{v}_{t-1}} + \tau} \right\rangle}_{T_{2,r}},$$

where $T_{1,r}$ may be bounded via

$$T_{1,r} = \widetilde{\beta}_1^{t-r} \left\langle \nabla f(x_{t-1}), \frac{\Delta_r(\sqrt{\widetilde{\beta}_2 \widetilde{v}_{t-1}} - \sqrt{\widetilde{v}_t})}{(\sqrt{\widetilde{v}_t} + \tau)(\sqrt{\widetilde{\beta}_2 \widetilde{v}_{t-1}} + \tau)} \right\rangle$$

$$= \widetilde{\beta}_1^{t-r} \left\langle \nabla f(x_{t-1}), \frac{-\Delta_r \Delta_t^2 (1 - \widetilde{\beta}_2)}{(\sqrt{\widetilde{v}_t} + \tau)(\sqrt{\widetilde{\beta}_2 \widetilde{v}_{t-1}} + \tau)(\sqrt{\widetilde{\beta}_2 \widetilde{v}_{t-1}} + \sqrt{\widetilde{v}_t})} \right\rangle$$

$$\le \frac{\|\Phi_1^K\| G \widetilde{\beta}_1^{t-r} (1 - \widetilde{\beta}_2)}{\tau} \sum_{j=1}^{d} \left[ \frac{\Delta_t^2}{\widetilde{v}_t} \right]_j.$$

Due to the exponential decay parameter in the first pseudogradient moment, we have

$$\eta \sum_{t=1}^{T} \sum_{r=1}^{t} \frac{\|\Phi_1^K\| G \widetilde{\beta}_1^{t-r}(1 - \widetilde{\beta}_2)}{\tau} \sum_{j=1}^{d} \left[ \frac{\Delta_t^2}{\widetilde{v}_t} \right]_j \le \eta \sum_{t=1}^{T} \sum_{r=1}^{t} \frac{\|\Phi_1^K\|^3 G \widetilde{\beta}_1^{t-r}(1 - \widetilde{\beta}_2)}{\tau^2}$$

$$\le \frac{\eta \|\Phi_1^K\|^3 G T (1 - \widetilde{\beta}_2)}{\tau^2}.$$

An analogue of the arguments made in the proof of Theorem 5.1 with appropriate modifications, e.g.,

$$\gamma_r := \frac{\eta_\ell}{|\mathcal{S}^r|} \sum_{i \in \mathcal{S}^r} \sum_{p=1}^{K} \frac{(1 - \beta_1) \sum_{\ell=1}^{p} \beta_1^{p-\ell}}{\sqrt{(1 - \beta_2) \sum_{\ell=1}^{\lceil \frac{p}{z} \rceil} \beta_2^{\lceil \frac{p}{z} \rceil - \ell} (g_{i,(\ell-1)z+1}^r)^2} + \varepsilon},$$

gives the main change as the asymptotic behavior of $\Psi_5$, which now satisfies

$$\Psi_5 = \Theta\left(\eta \eta_\ell^3 T\right).$$

The convergence rate is still dominated by $\Psi_1, \Psi_2$ as in Corollary B.9, which gives $\mathcal{O}(T^{-1/2})$.

## C. Federated Blended Optimization

In federated blended optimization, we distribute local optimizer strategies during the subsampling process which may be formalized as functions that take as input the availability of client resources, and outputs the number of local epochs, $K(O_l^i)$, as well as additional hyperparameters such as delay step size $z$ or preconditioner initialization. These may be chosen to streamline model training based on a variety of factors, such as straggler mitigation or dynamically restricted availability of local resources.

---

**Algorithm 4** Server-side ADAGRAD and client-side optimizer mixture (Blended Optimization)

---

**Require:** Local optimizer strategies $O_1, \ldots, O_{Op}$ (e.g. Adam, AdaGrad, SGD...)
**Require:** Initializations $x_0, \widetilde{v}_0 \geq \tau^2$ and $\widetilde{m}_0 \leftarrow 0$
**Require:** Global decay parameter $\widetilde{\beta}_1 \in [0, 1)$
1: **for** $t = 1, \ldots, T$ **do**
2:     Sample participating client multiset $S_l^t$ for each optimizer strategy $l \in [Op]$
3:     **for** each sampled client collection $l \in [Op]$ (in parallel) **do**
4:         **for** each client $i \in S_l^t$ (in parallel) **do**
5:             $x_{i,0}^{t,l} \leftarrow x_{t-1}$
6:             $x_{i,K(O_l^i)}^{t,l} \leftarrow \text{Optimize}(O_l, i, x_{i,0}^{t,l}, Clip = \text{True})$
7:             $\Delta_i^{t,l} = w(O_l) \left( x_{i,K(O_l^i)}^{t,l} - x_{t-1} \right)$
8:         **end for**
9:     **end for**
10:    $S \leftarrow \sum_{l \in [Op]} |S_l^t|$
11:    $\Delta_t = \frac{1}{S} \sum_{l \in [Op]} \sum_{i \in S_l^t} \Delta_i^{t,l}$
12:    $\widetilde{m}_t = \widetilde{\beta}_1 \widetilde{m}_{t-1} + (1 - \widetilde{\beta}_1) \Delta_t$
13:    $\widetilde{v}_t = \widetilde{v}_{t-1} + \Delta_t^2$
14:    $x_t = x_{t-1} + \eta \frac{\widetilde{m}_t}{\sqrt{\widetilde{v}_t} + \tau}$
15: **end for**

---

Federated blended optimization allows the trainer to utilize the unique strengths of each individual optimizer, balancing resource constraints and client noise. Each client has the option to run different optimizer strategies as the training rounds progress, depending on varying individual resource constraints or distribution shift in the local data stream. This faithfully corresponds to real-world settings where the availability of local resources are actively dynamic. Future work will provide empirical results on the performance of blended optimization, including identifying the settings in which mixing optimizer strategies are advantageous for distributed learning. The following theorem shows that under certain non-restrictive conditions, blended optimization still allows for convergence of the global gradient objective.

**Theorem C.1.** *Given client $i \in [N]$, strategy $l \in [Op]$, global timestep $r$, and local timestep $p$, assume that the optimizer strategies satisfy the parameter update rule*

$$x_{i,p}^{r,l} = x_{i,p-1}^{r,l} - \eta_\ell \sum_{\ell=1}^{p} \frac{a_{i,\ell}^{r,l} g_{i,\ell}^{r,l}}{\vartheta_{i,\ell}^{r,l}(g_{i,1}^{r,l}, \ldots, g_{i,\ell}^{r,l})}$$

*where*

$$0 < m_l \leq \vartheta_{i,\ell}^{r,l}(g_{i,1}^{r,l},\ldots,g_{i,\ell}^{r,l}) \leq M_l \quad and \quad 0 < a_l \leq a_{i,\ell}^{r,l} \leq A_l$$

*for all possible values of $i, \ell, r, l$. If $1 \leq K(O_l^i) \leq K$ and $0 < \Xi^- < w(O_l^i) < \Xi^+$, then Algorithm 4 admits an identical convergence bound as Theorem 5.1, with $\Psi_3$, $\Psi_4$ replaced by*

$$\Psi_3 = (1 - \widetilde{\beta}_1^T)\eta\eta_\ell CT\widetilde{L}\|\Phi_1^K\|^2,$$
$$\Psi_4 = (1 - \widetilde{\beta}_1)\eta\eta_\ell CTLc(\widetilde{\beta}_1)\|\Phi_2^K\|^2,$$
$$C = \frac{(\Xi^+)^2 K(K+1)(\max_{l\in[Op]} A_l^2)}{2\widetilde{\alpha}_1\tau \min_{l\in[Op]} m_l^2}.$$

*The intermediary $\widetilde{\gamma}_1, \widetilde{\alpha}_1$ values are defined as*

$$\widetilde{\gamma}_1 := \eta_\ell \frac{\Xi^- \min_{l\in[Op]} a_l}{\max_{l\in[Op]} M_l}, \quad \widetilde{\alpha}_1 := \frac{\Xi^- \min_{l\in[Op]} a_l}{K(K+1) \max_{l\in[Op]} M_l}.$$

We have opted to provide a looser bound for expository purposes, and the proof straightforwardly generalizes to finer bounds that depend on the individual characteristics of the optimizer strategy (e.g. $m_l, M_l, A_l$, etc). The extension to server-side Adam updates follows analogous steps to Section B.2.

It is easy to show that under the bounded gradient assumption (Assumption 2), Adam, AdaGrad, and SGD all satisfy the optimizer condition depicted in Theorem C.1. In Appendix D and E, we materialize two realizations of this framework as an example, using client-side Adam and AdaGrad with delayed preconditioner updates. Note that delayed updates require the debiasing term in Adam to be adjusted accordingly. To prove Theorem C.1, we begin with the following lemma.

**Lemma C.2.** *Under Algorithm 4, $|\Delta_i^{t,l}|$ is bounded by*

$$\Phi_1^K := \eta_\ell \Xi^+ \frac{K(K+1) \max_{l\in[Op]} A_l G}{2 \min_{l\in[Op]} m_l},$$

*and the server-side pseudogradient is bounded in absolute value by*

$$\Phi_2^K := \min\left\{\eta\sqrt{(1-\widetilde{\beta}_1)(1-\widetilde{\beta}_1^{2t})}, \frac{\eta}{\tau}\Phi_1^K\right\}.$$

*Proof.* Unraveling the definition of $\Delta_i^{t,l}$, we have

$$\Delta_i^{t,l} := -\eta_\ell w(O_l)\left(\sum_{p=1}^{K(O_l^i)} \sum_{\ell=1}^{p} \frac{a_{i,\ell}^{r,l} g_{i,\ell}^{r,l}}{\vartheta_{i,\ell}^{r,l}(g_{i,1}^{r,l},\ldots,g_{i,\ell}^{r,l})}\right),$$

which immediately gives

$$|\Delta_i^{t,l}| \leq \eta_\ell \Xi^+ \left(\sum_{p=1}^{K} \sum_{\ell=1}^{p} \frac{A_l G}{m_l}\right) = \eta_\ell \Xi^+ \frac{K(K+1) A_l G}{2 m_l}.$$

For the server bound, the proof is identical to Lemma B.2. $\qquad\square$

We are now ready to prove Theorem C.1.

*Proof.* As the proof follows a similar structure to Theorem 5.1, we provide only an outline for repetitive steps while focusing on differing aspects. As before, $L$-smoothness gives that

$$f(x_t) \leq f(x_{t-1}) + \eta T_{0,0} + (1-\widetilde{\beta}_1)\eta \sum_{r=1}^{t} T_{0,r} + \frac{\eta^2 L}{2}\left\|\frac{\widetilde{\beta}_1^t \widetilde{m}_0 + (1-\widetilde{\beta}_1)\sum_{r=1}^{t} \widetilde{\beta}_1^{t-r}\Delta_r}{\sqrt{\widetilde{v}_t} + \tau}\right\|^2 \tag{15}$$

where for $r \in [t]$,

$$T_{0,r} = \widetilde{\beta}_1^{t-r} \left\langle \nabla f(x_{t-1}), \frac{\Delta_r}{\sqrt{\widetilde{v}_t} + \tau} \right\rangle \quad \text{and} \quad T_{0,0} = \left\langle \nabla f(x_{t-1}), \frac{\widetilde{\beta}_1^t \widetilde{m}_0}{\sqrt{\widetilde{v}_t} + \tau} \right\rangle.$$

Decomposing $T_{0,r}$ as

$$T_{0,r} = \underbrace{\widetilde{\beta}_1^{t-r} \left\langle \nabla f(x_{t-1}), \frac{\Delta_r}{\sqrt{\widetilde{v}_t} + \tau} - \frac{\Delta_r}{\sqrt{\widetilde{v}_{t-1}} + \tau} \right\rangle}_{T_{1,r}} + \underbrace{\widetilde{\beta}_1^{t-r} \left\langle \nabla f(x_{t-1}), \frac{\Delta_r}{\sqrt{\widetilde{v}_{t-1}} + \tau} \right\rangle}_{T_{2,r}},$$

$T_{1,r}$ is bounded by

$$T_{1,r} \leq \frac{\|\Phi_1^K\| \|G \widetilde{\beta}_1^{t-r}}{\tau} \sum_{j=1}^{d} \left[ \frac{\Delta_t^2}{\widetilde{v}_t} \right]_j.$$

For $T_{2,r}$, we aim to apply a further decomposition for $\gamma_r > 0$,

$$T_{2,r} = \underbrace{\widetilde{\beta}_1^{t-r} \left\langle \frac{\nabla f(x_{t-1})}{\sqrt{\widetilde{v}_{t-1}} + \tau}, \Delta_r + \gamma_r \nabla f(x_{t-1}) \right\rangle}_{T_{2,r}^1} - \gamma_r \widetilde{\beta}_1^{t-r} \left\| \frac{\nabla f(x_{t-1})}{\sqrt{\sqrt{\widetilde{v}_{t-1}} + \tau}} \right\|^2.$$

Unraveling the definition of $\Delta_r$ gives

$$\Delta_r = \frac{1}{\sum_{l \in [Op]} |S_l^r|} \sum_{l \in [Op]} \sum_{i \in S_l^r} \Delta_i^{r,l} = \frac{-\eta_\ell}{\sum_{l \in [Op]} |S_l^r|} \sum_{l \in [Op]} \sum_{i \in S_l^r} \sum_{p=1}^{K(O_l^i)} \sum_{\ell=1}^{p} \frac{w(O_l) a_{i,\ell}^{r,l} g_{i,\ell}^{r,l}}{\vartheta_{i,\ell}^{r,l}(g_{i,1}^{r,l}, \ldots, g_{i,\ell}^{r,l})},$$

which induces the following value

$$\gamma_r := \frac{\eta_\ell}{\sum_{l \in [Op]} |S_l^t|} \sum_{l \in [Op]} \sum_{i \in S_l^t} \sum_{p=1}^{K(O_l^i)} \sum_{\ell=1}^{p} \frac{w(O_l) a_{i,\ell}^{r,l}}{\vartheta_{i,\ell}^{r,l}(g_{i,1}^{r,l}, \ldots, g_{i,\ell}^{r,l})} = \sum_{l \in [Op]} \gamma_r^l.$$

For the purposes of the proof, we shall consider a local device to have been dropped and unsampled if any runs less than 1 epoch. Then, we have

$$\gamma_r \in [\widetilde{\gamma}_1, \widetilde{\gamma}_2] := \left[ \eta_\ell \frac{\Xi^- \min_{l \in [Op]} a_l}{\max_{l \in [Op]} M_l}, \eta_\ell \frac{\Xi^+ K(K+1) \max_{l \in [Op]} a_l}{2 \min_{l \in [Op]} M_l} \right].$$

Expanding $T_{2,r}^1$ for $\alpha_r^l > 0$ to be fixed,

$$\widetilde{\beta}_1^{t-r} \left\langle \frac{\nabla f(x_{t-1})}{\sqrt{\widetilde{v}_{t-1}} + \tau}, \Delta_r + \gamma_r \nabla f(x_{t-1}) \right\rangle$$

$$= \frac{\widetilde{\beta}_1^{t-r}}{\sum_{l\in[Op]} |S_l^r|} \sum_{l\in[Op]} \sum_{i\in S_l^r} \sum_{p=1}^{K(O_l^i)} \sum_{\ell=1}^p \left\langle \frac{\nabla f(x_{t-1})}{\sqrt{\widetilde{v}_{t-1}} + \tau}, \frac{\eta_\ell w(O_l) a_{i,\ell}^{r,l}(\nabla f(x_{t-1}) - g_{i,\ell}^{r,l})}{\vartheta_{i,\ell}^{r,l}(g_{i,1}^{r,l}, \ldots, g_{i,\ell}^{r,l})} \right\rangle$$

$$\leq \frac{\eta_\ell \widetilde{\beta}_1^{t-r}}{4\sum_{l\in[Op]} |S_l^r|} \sum_{l\in[Op]} \alpha_r^l \sum_{i\in\mathcal{S}_l^r} K(O_l^i)(K(O_l^i)+1) \left\| \frac{\nabla f(x_{t-1})}{\sqrt{\sqrt{\widetilde{v}_{t-1}} + \tau}} \right\|^2$$

$$+ \frac{\eta_\ell \widetilde{\beta}_1^{t-r}}{2\sum_{l\in[Op]} |S_l^r|} \sum_{l\in[Op]} \frac{1}{\alpha_r^l} \sum_{i\in S_l^r} \sum_{p=1}^{K(O_l^i)} \sum_{\ell=1}^p \left\| \frac{w(O_l) a_{i,\ell}^{r,l}\left(\nabla f(x_{t-1}) - \nabla F_i(x_{i,\ell-1}^{r,l})\right)}{\vartheta_{i,\ell}^{r,l}(g_{i,1}^{r,l}, \ldots, g_{i,\ell}^{r,l})\sqrt{\sqrt{\widetilde{v}_{t-1}} + \tau}} \right\|^2$$

$$\leq \frac{\eta_\ell \widetilde{\beta}_1^{t-r} \max_{l\in[Op]} \alpha_r^l K(K+1)}{4} \left\| \frac{\nabla f(x_{t-1})}{\sqrt{\sqrt{\widetilde{v}_{t-1}} + \tau}} \right\|^2$$

$$+ \frac{\eta_\ell \widetilde{\beta}_1^{t-r}(\Xi^+)^2}{2\tau \sum_{l\in[Op]} |S_l^r|} \sum_{l\in[Op]} \frac{A_l^2}{\alpha_r^l m_l^2} \sum_{i\in S_l^r} \sum_{p=1}^{K(O_l^i)} \sum_{\ell=1}^p \left\| \nabla f(x_{t-1}) - \nabla F_i(x_{i,\ell-1}^{r,l}) \right\|^2$$

We aim to control the first term by setting for all $l \in [Op]$

$$\alpha_r^l = \frac{\gamma_r}{\eta_\ell K(K+1)} \in [\widetilde{\alpha}_1, \widetilde{\alpha}_2] := \left[ \frac{\Xi^- \min_{l\in[Op]} a_l}{K(K+1) \max_{l\in[Op]} M_l}, \frac{\Xi^+ K(K+1) \max_{l\in[Op]} a_l}{2K(K+1) \min_{l\in[Op]} M_l} \right].$$

Via gradient clipping as before, we have

$$\left\| \nabla f(x_{t-1}) - \nabla F_i(x_{i,\ell-1}^{r,l}) \right\|^2 \leq 2L(t-r)^2 \|\Phi_2^K\|^2 + 2\widetilde{L}\|\Phi_1^K\|^2.$$

Noting that

$$\frac{\eta_\ell \widetilde{\beta}_1^{t-r}(\Xi^+)^2}{2\tau \sum_{l\in[Op]} |S_l^r|} \sum_{l\in[Op]} \frac{A_l^2}{\alpha_r^l m_l^2} \sum_{i\in S_l^r} \sum_{p=1}^{K(O_l^i)} \sum_{\ell=1}^p \left\| \nabla f(x_{t-1}) - \nabla F_i(x_{i,\ell-1}^{r,l}) \right\|^2$$

$$\leq \frac{\eta_\ell (\Xi^+)^2 K(K+1)(\max_{l\in[Op]} A_l^2)}{2\widetilde{\alpha}_1 \tau \min_{l\in[Op]} m_l^2} \left( L\widetilde{\beta}_1^{t-r}(t-r)^2 \|\Phi_2^K\|^2 + \widetilde{L}\widetilde{\beta}_1^{t-r}\|\Phi_1^K\|^2 \right),$$

collecting terms into equation (15) gives that

$$f(x_t) \leq f(x_{t-1}) + \eta T_{0,0} + \eta^2 L \left\| \frac{\widetilde{\beta}_1^t \widetilde{m}_0}{\sqrt{\widetilde{v}_t} + \tau} \right\|^2 + \frac{\eta^2 L d \|\Phi_1^K\|^2}{\tau^2} + (1-\widetilde{\beta}_1)\eta \sum_{r=1}^t \left( \frac{\|\Phi_1^K\| |G|\widetilde{\beta}_1^{t-r}}{\tau} \sum_{j=1}^d \left[ \frac{\Delta_t^2}{\widetilde{v}_t} \right]_j \right)$$

$$+ (1-\widetilde{\beta}_1)\eta\eta_\ell \sum_{r=1}^t \underbrace{\frac{(\Xi^+)^2 K(K+1)(\max_{l\in[Op]} A_l^2)}{2\widetilde{\alpha}_1 \tau \min_{l\in[Op]} m_l^2}}_{C} \left( L\widetilde{\beta}_1^{t-r}(t-r)^2 \|\Phi_2^K\|^2 + \widetilde{L}\widetilde{\beta}_1^{t-r}\|\Phi_1^K\|^2 \right)$$

$$+ (1-\widetilde{\beta}_1)\eta \sum_{r=1}^t \left( -\frac{3\gamma_r \widetilde{\beta}_1^{t-r}}{4} \left\| \frac{\nabla f(x_{t-1})}{\sqrt{\sqrt{\widetilde{v}_{t-1}} + \tau}} \right\|^2 \right). \tag{16}$$

By initializing $\widetilde{m}_0 \leftarrow 0$ and enhancing the upper bound by substituting $\widetilde{\gamma}_1$ into $\gamma_r$, telescoping gives

$$
\frac{3(1-\widetilde{\beta}_1)\eta\widetilde{\gamma}_1}{4} \sum_{t=1}^{T}\sum_{r=1}^{t}\widetilde{\beta}_1^{t-r} \left\| \frac{\nabla f(x_{t-1})}{\sqrt{\sqrt{\widetilde{v}_{t-1}}+\tau}} \right\|^2 \leq f(x_0) - f(x^*) + \frac{(1-\widetilde{\beta}_1)\eta\|\Phi_1^K\|G}{\tau}\sum_{t=1}^{T}\sum_{r=1}^{t}\sum_{j=1}^{d}\widetilde{\beta}_1^{t-r}\left[\frac{\Delta_t^2}{\widetilde{v}_t}\right]_j
$$
$$
+ \frac{\eta^2 LTd\|\Phi_1^K\|^2}{\tau^2} + (1-\widetilde{\beta}_1)\eta\eta_\ell C \sum_{t=1}^{T}\sum_{r=1}^{t}\left(L\widetilde{\beta}_1^{t-r}(t-r)^2\|\Phi_2^K\|^2 + \widetilde{L}\widetilde{\beta}_1^{t-r}\|\Phi_1^K\|^2\right). \tag{17}
$$

Again by noting that

$$
\frac{3(1-\widetilde{\beta}_1)\eta\widetilde{\gamma}_1}{4}\sum_{t=1}^{T}\sum_{r=1}^{t}\widetilde{\beta}_1^{t-r}\left\|\frac{\nabla f(x_{t-1})}{\sqrt{\sqrt{\widetilde{v}_{t-1}}+\tau}}\right\|^2 \geq \frac{3(1-\widetilde{\beta}_1)\eta\widetilde{\gamma}_1 T}{4\left(\sqrt{T\|\Phi_1^K\|^2+\widetilde{v}_0}+\tau\right)}\min_{t\in[T]}\|\nabla f(x_{t-1})\|^2,
$$

Lemmas B.6 and B.7 give that

$$
\frac{3(1-\widetilde{\beta}_1)\eta\widetilde{\gamma}_1 T}{4\left(\sqrt{T\|\Phi_1^K\|^2+\widetilde{v}_0}+\tau\right)}\min_{t\in[T]}\|\nabla f(x_{t-1})\|^2 \leq f(x_0) - f(x^*) + \frac{\eta^2 LTd\|\Phi_1^K\|^2}{\tau^2}
$$
$$
+ (1-\widetilde{\beta}_1^T)\eta\eta_\ell CT\widetilde{L}\|\Phi_1^K\|^2 + (1-\widetilde{\beta}_1)\eta\eta_\ell CTLc(\widetilde{\beta}_1)\|\Phi_2^K\|^2
$$
$$
+ \frac{\eta d\|\Phi_1^K\|G\left(1-\widetilde{\beta}_1+\log\left(1+\frac{T\|\Phi_1^K\|^2}{\tau^2}\right)\right)}{\tau}.
$$

This implies that

$$
\min_{t\in[T]}\|\nabla f(x_{t-1})\|^2 \leq \frac{\Psi_1+\Psi_2+\Psi_3+\Psi_4+\Psi_5}{\Psi_6},
$$

where

$$
\Psi_1 = f(x_0) - f(x^*),
$$
$$
\Psi_2 = \frac{\eta^2 LTd\|\Phi_1^K\|^2}{\tau^2},
$$
$$
\Psi_3 = (1-\widetilde{\beta}_1^T)\eta\eta_\ell CT\widetilde{L}\|\Phi_1^K\|^2,
$$
$$
\Psi_4 = (1-\widetilde{\beta}_1)\eta\eta_\ell CTLc(\widetilde{\beta}_1)\|\Phi_2^K\|^2,
$$
$$
\Psi_5 = \frac{\eta d\|\Phi_1^K\|G\left(1-\widetilde{\beta}_1+\log\left(1+\frac{T\|\Phi_1^K\|^2}{\tau^2}\right)\right)}{\tau},
$$
$$
\Psi_6 = \frac{3(1-\widetilde{\beta}_1)\eta\widetilde{\gamma}_1 T}{4\left(\sqrt{T\|\Phi_1^K\|^2+\widetilde{v}_0}+\tau\right)},
$$
$$
C = \frac{(\Xi^+)^2 K(K+1)(\max_{l\in[Op]}A_l^2)}{2\widetilde{\alpha}_1\tau\min_{l\in[Op]}m_l^2}.
$$

The intermediary $\widetilde{\gamma}_1, \widetilde{\alpha}_1$ values are defined as

$$
\widetilde{\gamma}_1 := \eta_\ell\frac{\Xi^-\min_{l\in[Op]}a_l}{\max_{l\in[Op]}M_l}, \quad \widetilde{\alpha}_1 := \frac{\Xi^-\min_{l\in[Op]}a_l}{K(K+1)\max_{l\in[Op]}M_l}.
$$

$\square$

# D. Adam Delayed Moment Updates (ADMU)

We begin with a brief description of ADAM (Kingma & Ba, 2015).

---

**Algorithm 5** Adam Optimization Algorithm

---

**Require:** $\eta_\ell$: Step size
**Require:** $\beta_1, \beta_2 \in [0, 1)$: Exponential decay rates for the moment estimates
**Require:** $f(x)$: Stochastic objective function with parameters $x$
**Require:** $\varepsilon > 0$: Smoothing term
**Require:** $x_0$: Initial parameter vector
1: Initialize $m_0 \leftarrow 0$ (1st moment vector)
2: Initialize $v_0 \leftarrow 0$ (2nd moment vector)
3: Initialize $t \leftarrow 0$ (Timestep)
4: **while** not converged **do**
5:    $t \leftarrow t + 1$
6:    $g_t \leftarrow \nabla_x f_t(x_{t-1})$
7:    $m_t \leftarrow \beta_1 \cdot m_{t-1} + (1 - \beta_1) \cdot g_t$
8:    $v_t \leftarrow \beta_2 \cdot v_{t-1} + (1 - \beta_2) \cdot g_t^2$
9:    $\hat{m}_t \leftarrow m_t/(1 - \beta_1^t)$
10:   $\hat{v}_t \leftarrow v_t/(1 - \beta_2^t)$
11:   $x_t \leftarrow x_{t-1} - \eta_\ell \cdot \hat{m}_t/(\sqrt{\hat{v}_t} + \varepsilon)$
12: **end while**
13: **return** $x_t$

---

**Algorithm 6** Adam with Delayed Moment Updates (ADMU)

---

**Require:** $\eta_\ell$: Step size
**Require:** $z \in \mathbb{Z}_{\geq 1}$: Step delay for second moment estimate updates (where $z = 1$ gives no delay)
**Require:** $\beta_1, \beta_2 \in [0, 1)$: Exponential decay rates for the moment estimates
**Require:** $f(x)$: Stochastic objective function with parameters $x$
**Require:** $x_0$: Initial parameter vector
**Require:** $\varepsilon > 0$: Smoothing term
1: Initialize $m_0 \leftarrow 0$ (1st moment vector)
2: Initialize $v_0 \leftarrow 0$ (2nd moment vector)
3: Initialize $t \leftarrow 0$ (Timestep)
4: **while** not converged **do**
5:    $t \leftarrow t + 1$
6:    $g_t \leftarrow \nabla_x f_t(x_{t-1})$
7:    $m_t \leftarrow \beta_1 \cdot m_{t-1} + (1 - \beta_1) \cdot g_t$
8:    $\hat{m}_t \leftarrow m_t/(1 - \beta_1^t)$
9:    **if** $(t-1)/z \in \mathbb{Z}$ **then**
10:      $v_t \leftarrow \beta_2 \cdot v_{t-1} + (1 - \beta_2) \cdot g_t^2$
11:      $\hat{v}_t \leftarrow v_t/(1 - \beta_2^{\lfloor \frac{t-1}{z} \rfloor + 1})$
12:    **else**
13:      $\hat{v}_t \leftarrow \hat{v}_{t-1}$
14:    **end if**
15:   $x_t \leftarrow x_{t-1} - \eta_\ell \cdot \hat{m}_t/(\sqrt{\hat{v}_t} + \varepsilon)$
16: **end while**
17: **return** $x_t$

---

Considering client-side resource constraints in the federated setting, we propose an adapted version of Adam with delayed precondtioner updates aimed at relieving the cost of moment estimate computation in Algorithm 6 which we call ADMU.

Following (Kingma & Ba, 2015), we provide an intuitive justification for the initialization bias correction employed in ADMU. Recall that the motivation for adaptive step-size in ADAM is updating the parameters via empirical estimates of the pseudo-gradient $\mathbb{E}[g]/\sqrt{\mathbb{E}[g^2]}$, which allows for both momentum and autonomous annealing near steady states. The square root is taken in the denominator to homogenize the degree of the gradient. Bias correction for ADMU adheres to the same principle, while requiring an additional assumption of gradient stabilization during the $z$-step preconditioner update delay. An equivalent formulation of the moment estimates in Algorithm 6 for general $t$ is given

$$m_t = m_0\beta_1^t + (1 - \beta_1)\sum_{r=1}^{t} \beta_1^{t-r} \cdot g_r,$$

$$v_t = v_0\beta_2^{\lfloor \frac{t-1}{z} \rfloor + 1} + (1 - \beta_2)\sum_{r=1}^{t} \beta_2^{\lfloor \frac{t-1}{z} \rfloor + 1 - \lceil \frac{r}{z} \rceil} \cdot g_{\lceil \frac{r}{z} \rceil z - z + 1} \odot g_{\lceil \frac{r}{z} \rceil z - z + 1} \cdot \chi\{\frac{r-1}{z} \in \mathbb{Z}_{\geq 0}\}$$

$$= v_0\beta_2^{\lfloor \frac{t-1}{z} \rfloor + 1} + (1 - \beta_2)\sum_{r=1}^{\lceil \frac{t}{z} \rceil} \beta_2^{\lceil \frac{t}{z} \rceil - r} g_{(r-1)z+1} \odot g_{(r-1)z+1}. \tag{18}$$

We work with $v_t$ as the proof for $m_t$ is analogous with $z = 1$. Assume that the gradients $g_1, \ldots, g_t$ are drawn from a latent gradient distribution $g_i \sim \widetilde{\mathcal{D}}(g_i)$. We aim to extract a relation between the expected delayed exponential moving average of the second moment $\mathbb{E}[v_t]$ and the true gradient expectation $\mathbb{E}[g_t^2]$. Taking expectation of both sides in equation (18),

$$\mathbb{E}[v_t] = v_0\beta_1^{\lfloor \frac{t-1}{z} \rfloor + 1} + (1 - \beta_2)\sum_{r=1}^{\lceil \frac{t}{z} \rceil} \beta_2^{\lceil \frac{t}{z} \rceil - r} \mathbb{E}\left[g_{(r-1)z+1}^2\right]$$

$$\approx \zeta + (1 - \beta_2)\mathbb{E}\left[g_t^2\right]\sum_{r=1}^{\lceil \frac{t}{z} \rceil} \beta_2^{\lceil \frac{t}{z} \rceil - r}$$

$$\approx \mathbb{E}[g_t^2]\left(1 - \beta_1^{\lfloor \frac{t-1}{z} \rfloor + 1}\right).$$

Here, we have used zero initialization for the first moment estimate, while accumulating any error terms in $\zeta$. Several assumptions can lead to small $\zeta$. As in (Kingma & Ba, 2015), we assume that $\beta_1$ is chosen small enough that the exponential moving average decay undermines the influence of non-recent gradients $g_i$ for $i < \lceil \frac{t}{z} \rceil z - z + 1$. A second assumption is that the latent gradient distribution remains stable during the $z$-step delay as training progresses, allowing the approximation $\mathbb{E}[g_t] \approx \mathbb{E}[g_{\lceil \frac{t}{z} \rceil z - z + 1}]$. This leaves the residual scaling of the true gradient second moment of the form $1 - \beta^\varphi$, which is caused by (zero) initialization as setting $v_0 = \mathbb{E}[g_t^2]$ eliminates $\beta^\varphi$. Therefore, bias correction is enforced by scaling the empirical $v_t$ estimate by the inverse. We note that $v_0$ need not be initialized to 0, in which case we should additionally translate $v_t$ by $-v_0\beta_1^{\lfloor \frac{t-1}{z} \rfloor + 1}$ prior to the inverse scaling.

### D.1. Non-convex convergence analysis

A description of FedAdaAdam is given as Algorithm 7. A few remarks are in order. Firstly, to allow for straggler mitigation, we allow the number of client $i$ epochs $\overline{K}_i^t$ at timestep $t$ to vary among the clients $i \in \mathcal{S}_i$. Although Algorithm 7 sets a schedule for client epochs and pseudogradient weights for clarity of exposition, dynamic allocation still allows the convergence proof to go through, as long as the schedule weights are bounded. By default, we set $\overline{K}^t = K$ and $\Xi^t = B = 1$ to avoid tuning a large number of hyperparameters or having to sample from a client epoch count distribution for the client subsampling case.

Secondly, for the purposes of the proof we shall consider a local device to have been dropped and unsampled if any runs less than 1 epoch. We also enforce that pseudogradient weights are bounded positively from below, i.e. $\Xi_i^t > \varepsilon_w > 0$. We now provide a convergence bound for the general, non-convex case which holds for both full and partial client participation.

---

**Algorithm 7** Adaptive server-side ADAGRAD and client-side ADAM (FedAdaAdam)

---

**Require:** Update delay step size $z \in \mathbb{Z}_{\geq 1}$, initializations $x_0, \widetilde{v}_0 \geq \tau^2$ and $\widetilde{m}_0 \leftarrow 0$
**Require:** Global and local decay parameters $\widetilde{\beta}_1, \widetilde{\beta}_2, \beta_1, \beta_2 \in [0, 1)$
**Require:** Pseudogradient weighting schedule $\Xi^1 \times \cdots \times \Xi^T \in \mathbb{R}^{|\mathcal{S}^1|} \times \cdots \times \mathbb{R}^{|\mathcal{S}^T|}$ for $\|\Xi^t\|_\infty \leq B$
**Require:** Client epoch schedule $\overline{K}^1 \times \cdots \times \overline{K}^T \in \mathbb{Z}_{\geq 1}^{|\mathcal{S}^1|} \times \cdots \times \mathbb{Z}_{\geq 1}^{|\mathcal{S}^T|}$ for $\|\overline{K}^t\|_\infty \leq K, \forall t \in [T]$
**Require:** Local epsilon smoothing term $\varepsilon_s > 0$

1: **for** $t = 1, \ldots, T$ **do**
2:     Sample subset $\mathcal{S}^t \subset [N]$ of clients
3:     **for** each client $i \in \mathcal{S}^t$ (in parallel) **do**
4:         $x_{i,0}^t \leftarrow x_{t-1}$
5:         Initialize $m_0, v_0 \geq 0$ with default values $m_0, v_0 \leftarrow 0$
6:         **for** $k = 1, \ldots, \overline{K}_i^t$ **do**
7:             Draw stochastic gradient $g_{i,k}^t \sim \mathcal{D}(x_{i,k-1}^t)$ with mean $\nabla F_i(x_{i,k-1}^t) \in \mathbb{R}^d$
8:             $m_k \leftarrow \beta_1 \cdot m_{k-1} + (1 - \beta_1) \cdot g_{i,k}^t$
9:             $\hat{m}_k \leftarrow m_k / (1 - \beta_1^k)$
10:            **if** $(k-1)/z \in \mathbb{Z}$ **then**
11:                $v_k \leftarrow \beta_2 \cdot v_{k-1} + (1 - \beta_2) \cdot g_{i,k}^t \odot g_{i,k}^t$
12:                $\hat{v}_k \leftarrow v_k / (1 - \beta_2^{\lfloor \frac{k-1}{z} \rfloor + 1})$
13:            **else**
14:                $v_k \leftarrow v_{k-1}$
15:            **end if**
16:            **if** $0 < \|\hat{m}_k / (\sqrt{\hat{v}_k} + \epsilon)\| < \varepsilon_s$ **then**
17:                $m_k \leftarrow 0$
18:            **end if**
19:            $x_{i,k}^t \leftarrow x_{i,k-1}^t - \eta_\ell \cdot \hat{m}_k / (\sqrt{\hat{v}_k} + \epsilon)$
20:         **end for**
21:         $\Delta_i^t = \Xi_i^t \left( x_{i,\overline{K}_i^t}^t - x_{t-1} \right)$
22:     **end for**
23:     $\Delta_t = \frac{1}{|\mathcal{S}^t|} \sum_{i \in \mathcal{S}^t} \Delta_i^t$
24:     $\widetilde{m}_t = \widetilde{\beta}_1 \widetilde{m}_{t-1} + (1 - \widetilde{\beta}_1) \Delta_t$
25:     $\widetilde{v}_t = \widetilde{v}_{t-1} + \Delta_t^2$
26:     $x_t = x_{t-1} + \eta \frac{\widetilde{m}_t}{\sqrt{\widetilde{v}_t} + \tau}$
27: **end for**

---

**Corollary D.1.** *For Algorithm 7, we have an identical bound to Theorem 5.1 with* $\Psi_3, \Psi_4$ *replaced by*

$$\Psi_3 = \frac{(1 - \widetilde{\beta}_1^T)\eta\eta_\ell(1 - \beta_1^{2K})K\widetilde{L}B^2 T\|\Phi_1^K\|^2}{2\widetilde{\alpha}_1\tau\varepsilon^2},$$

$$\Psi_4 = \frac{(1 - \widetilde{\beta}_1)\eta\eta_\ell(1 - \beta_1^{2K})KLTB^2 c(\widetilde{\beta}_1)\|\Phi_2^K\|^2}{2\widetilde{\alpha}_1\tau\varepsilon^2}.$$

*Here, the intermediary* $\widetilde{\gamma}_1, \widetilde{\alpha}_1$ *values are defined for* $K^- := \min_{i,t} \overline{K}_i^t \geq 1$ *as*

$$\widetilde{\gamma}_1 := \eta_\ell\varepsilon_w \sum_{p=1}^{K^-} \frac{1 - \beta_1^p}{G\sqrt{1 - \beta_2^{\lceil \frac{p}{z} \rceil}} + \varepsilon}, \quad \widetilde{\alpha}_1 := \sum_{p=1}^{K^-} \frac{\varepsilon_w(1 - \beta_1^p)}{\left(G\sqrt{1 - \beta_2^{\lceil \frac{p}{z} \rceil}} + \varepsilon\right)(K+1)^2}.$$

The proof is subsumed by or analogous to Theorems 5.1 and C.1, with changes summarized in the following lemma.

**Lemma D.2.** *Under Algorithm 7,* $|\Delta_i^t|$ *is bounded by*

$$|\Delta_i^t| \leq \Phi_1^{\overline{K}_i^t} := |\Xi_i^t| \cdot \left( \eta_\ell\overline{K}_i^t \sqrt{\left( \sum_{r=1}^{\lceil \frac{\overline{K}_i^t}{z} \rceil} \frac{\beta_1^{2\lceil \frac{\overline{K}_i^t}{z} \rceil - 2r}}{\beta_2^{\lceil \frac{\overline{K}_i^t}{z} \rceil - r}} \right)} + \Phi_0^{\overline{K}_i^t} \right)$$

*where*

$$\Phi_0^{\overline{K}_i^t} := \frac{\overline{K}_i^t G\eta_\ell(1 - \beta_1^{\overline{K}_i^t})}{\varepsilon}.$$

*Proof.* Recall that $\Delta_t = 1/|\mathcal{S}^t| \sum_{i \in \mathcal{S}^t} \Delta_i^t$ and $\Delta_i^t = \Xi_i^t\left(x_{i,\overline{K}_i^t}^t - x_{i,0}^t\right)$. By telescoping for $\overline{K}_i^t$ local steps and the definition of gradient updates in ADMU, we obtain

$$\Delta_i^t = \sum_{p=1}^{\overline{K}_i^t} -\eta_\ell\Xi_i^t \frac{\hat{m}_p}{\sqrt{\hat{v}_p} + \varepsilon} = -\eta_\ell\Xi_i^t \sum_{p=1}^{\overline{K}_i^t} \frac{m_0\beta_1^p + (1 - \beta_1)\sum_{r=1}^p \beta_1^{p-r} \cdot g_{i,r}^t}{\sqrt{v_0\beta_2^{\lfloor \frac{p-1}{z} \rfloor + 1} + (1 - \beta_2)\sum_{r=1}^{\lceil \frac{p}{z} \rceil} \beta_2^{\lceil \frac{p}{z} \rceil - r}(g_{i,(r-1)z+1}^t)^2} + \varepsilon}$$

We assume $m_0, v_0 \leftarrow 0$ for expository purposes, although $v_0 > 0$ also suffices for the analysis (ending in a slightly different $\Phi_1^{\overline{K}_i^t}$). This gives that

$$\Delta_i^t = -\eta_\ell\Xi_i^t \sum_{p=1}^{\overline{K}_i^t} \frac{(1 - \beta_1)\sum_{r=1}^p \beta_1^{p-r} \cdot g_{i,r}^t}{\sqrt{(1 - \beta_2)\sum_{r=1}^{\lceil \frac{p}{z} \rceil} \beta_2^{\lceil \frac{p}{z} \rceil - r}(g_{i,(r-1)z+1}^t)^2} + \varepsilon}$$

$$= -\eta_\ell\Xi_i^t \sum_{p=1}^{\overline{K}_i^t} \frac{(1 - \beta_1)\sum_{r=1}^{\lceil \frac{p}{z} \rceil} \beta_1^{\lceil \frac{p}{z} \rceil - r} \cdot g_{i,(r-1)z+1}^t}{\sqrt{(1 - \beta_2)\sum_{r=1}^{\lceil \frac{p}{z} \rceil} \beta_2^{\lceil \frac{p}{z} \rceil - r}(g_{i,(r-1)z+1}^t)^2} + \varepsilon}$$

$$- \eta_\ell\Xi_i^t \sum_{p=1}^{\overline{K}_i^t} \frac{(1 - \beta_1)\sum_{r=1}^p \beta_1^{p-r} \cdot g_{i,r}^t \cdot \chi_{\left\{\frac{p-1}{z} \notin \mathbb{Z}\right\}}}{\sqrt{(1 - \beta_2)\sum_{r=1}^{\lceil \frac{p}{z} \rceil} \beta_2^{\lceil \frac{p}{z} \rceil - r}(g_{i,(r-1)z+1}^t)^2} + \varepsilon}.$$

To obtain a deterministic bound, we cannot ignore the worst-case stochastic realization that $g_{i,(r-1)z+1}^t = 0$ for $\forall r \in [\lceil \frac{p}{z} \rceil]$. Therefore, we form the intermediary upper bound

$$|\Delta_i^t| \leq \eta_\ell|\Xi_i^t| \sum_{p=1}^{\overline{K}_i^t} \frac{(1 - \beta_1)\sum_{r=1}^{\lceil \frac{p}{z} \rceil} \beta_1^{\lceil \frac{p}{z} \rceil - r} \cdot \left|g_{i,(r-1)z+1}^t\right|}{\sqrt{(1 - \beta_2)\sum_{r=1}^{\lceil \frac{p}{z} \rceil} \beta_2^{\lceil \frac{p}{z} \rceil - r}(g_{i,(r-1)z+1}^t)^2} + \varepsilon}$$

$$+ \frac{\eta_\ell|\Xi_i^t|(1 - \beta_1)}{\varepsilon}\left( \sum_{p=1}^{\overline{K}_i^t}\sum_{r=1}^p \beta_1^{p-r} \cdot |g_{i,r}^t| \cdot \chi_{\left\{\frac{p-1}{z} \notin \mathbb{Z}\right\}} \right). \tag{19}$$

Note that the first term is $0$ in the worst-case scenario above, which implies that any non-negative upper bound is trivially satisfied. Therefore, we may assume without loss of generality that at least one sampled gradient $g_{i,(r-1)z+1}^t$ is nontrivial and remove $\varepsilon$ from the denominator to obtain an upper bound. By Cauchy-Schwartz, we have

$$
\left( \sum_{r=1}^{\lceil \frac{p}{z} \rceil} \beta_2^{\lceil \frac{p}{z} \rceil - r} (g_{i,(r-1)z+1}^t)^2 \right) \left( \sum_{r=1}^{\lceil \frac{p}{z} \rceil} \frac{\beta_1^{2\lceil \frac{p}{z} \rceil - 2r}}{\beta_2^{\lceil \frac{p}{z} \rceil - r}} \right) \geq \left( \sum_{r=1}^{\lceil \frac{p}{z} \rceil} \beta_1^{\lceil \frac{p}{z} \rceil - r} \cdot \left| g_{i,(r-1)z+1}^t \right| \right)^2
$$

which implies

$$
\begin{aligned}
\left| \Delta_i^t \right| &\leq \eta_\ell |\Xi_i^t| \sum_{p=1}^{\overline{K}_i^t} \sqrt{ \left( \sum_{r=1}^{\lceil \frac{p}{z} \rceil} \frac{\beta_1^{2\lceil \frac{p}{z} \rceil - 2r}}{\beta_2^{\lceil \frac{p}{z} \rceil - r}} \right) } + \frac{\eta_\ell |\Xi_i^t|(1-\beta_1)}{\varepsilon} \left( \sum_{p=1}^{\overline{K}_i^t} \sum_{r=1}^{p} \beta_1^{p-r} \cdot \left| g_{i,r}^t \right| \cdot \chi_{\left\{ \frac{p-1}{z} \notin \mathbb{Z} \right\}} \right) \\
&\leq \eta_\ell |\Xi_i^t| \sum_{p=1}^{\overline{K}_i^t} \sqrt{ \left( \sum_{r=1}^{\lceil \frac{p}{z} \rceil} \frac{\beta_1^{2\lceil \frac{p}{z} \rceil - 2r}}{\beta_2^{\lceil \frac{p}{z} \rceil - r}} \right) } + \frac{\overline{K}_i^t G \eta_\ell |\Xi_i^t|(1-\beta_1)}{\varepsilon} \cdot \frac{(1-\beta_1^{\overline{K}_i^t})}{(1-\beta_1)} \\
&\leq \eta_\ell |\Xi_i^t| \overline{K}_i^t \sqrt{ \left( \sum_{r=1}^{\lceil \frac{\overline{K}_i^t}{z} \rceil} \frac{\beta_1^{2\lceil \frac{\overline{K}_i^t}{z} \rceil - 2r}}{\beta_2^{\lceil \frac{\overline{K}_i^t}{z} \rceil - r}} \right) } + \frac{\overline{K}_i^t G \eta_\ell |\Xi_i^t|(1-\beta_1^{\overline{K}_i^t})}{\varepsilon}.
\end{aligned}
$$

$\square$

It can be shown that case of no update delay $z = 1$ allows for $\Phi_0^{\overline{K}_i^t} = 0$, following a similar proof to the one given above. Note that $\Phi_0^{\overline{K}_i^t}$ handles the superfluous gradient terms cemented by delaying preconditioner updates for the second moment, while moving averaging is performed for the first moment estimate. It also follows that $\Delta_t$ is also upper bounded by the identical bound scaled by $\max_t \|\Xi^t\|_\infty \leq B$, as the average of the $\Delta_i^t$.

## E. AdaGrad with Delayed Updates (AGDU)

We present AdaGrad with delayed preconditioner as Algorithm 8 for completeness.

---

**Algorithm 8** AdaGrad with Delayed Updates (AGDU)

---

**Require:** $\eta_\ell$: Step size
**Require:** $z \in \mathbb{Z}_{\geq 1}$: Step delay for second moment estimate updates (where $z = 1$ gives no delay)
**Require:** $f(x)$: Stochastic objective function with parameters $x$
**Require:** $x_0$: Initial parameter vector
**Require:** $\varepsilon > 0$: Smoothing term
1: Initialize $v_0 \leftarrow 0$ (2nd moment vector)
2: Initialize $t \leftarrow 0$ (Timestep)
3: **while** not converged **do**
4:     $t \leftarrow t + 1$
5:     $g_t \leftarrow \nabla_x f_t(x_{t-1})$
6:     **if** $(t-1)/z \in \mathbb{Z}$ **then**
7:         $v_t \leftarrow v_{t-1} + g_t^2$
8:     **else**
9:         $v_t \leftarrow v_{t-1}$
10:    **end if**
11:    $x_t \leftarrow x_{t-1} - \eta_\ell \cdot g_t / (\sqrt{v_t} + \varepsilon)$
12: **end while**
13: **return** $x_t$

---

Note that due to delayed updates, local gradient updates are not necessarily elementwise bounded in absolute value by $\eta_\ell$. We may expand the delayed updates for $v_t$ as

$$v_t = v_0 + \sum_{r=1}^{\lceil \frac{t}{z} \rceil} g_{(r-1)z+1} \odot g_{(r-1)z+1}.$$

---

**Algorithm 9** Adaptive server and client-side ADAGRAD (FedAdaAdagrad)

---

**Require:** Update delay step size $z \in \mathbb{Z}_{\geq 1}$, initializations $x_0, \widetilde{v}_0 \geq \tau^2$ and $\widetilde{m}_0 \leftarrow 0$
**Require:** Global decay parameter $\widetilde{\beta}_1 \in [0, 1)$
**Require:** Pseudogradient weighting schedule $\Xi^1 \times \cdots \times \Xi^T \in \mathbb{R}^{|\mathcal{S}^1|} \times \cdots \times \mathbb{R}^{|\mathcal{S}^T|}$ for $\|\Xi^t\|_\infty \leq B$
**Require:** Client epoch schedule $\overline{K}^1 \times \cdots \times \overline{K}^T \in \mathbb{Z}_{\geq 1}^{|\mathcal{S}^1|} \times \cdots \times \mathbb{Z}_{\geq 1}^{|\mathcal{S}^T|}$ for $\|\overline{K}^t\|_\infty \leq K, \forall t \in [T]$
**Require:** Local epsilon smoothing term $\varepsilon_s > 0$, global smoothing term $\tau > 0$
 1: **for** $t = 1, \ldots, T$ **do**
 2:     Sample subset $\mathcal{S}^t \subset [N]$ of clients
 3:     **for** each client $i \in \mathcal{S}^t$ (in parallel) **do**
 4:         $x_{i,0}^t \leftarrow x_{t-1}$
 5:         Initialize $v_0 \geq 0$ with default value $v_0 \leftarrow 0$ (what if use $\tau$ here?)
 6:         **for** $k = 1, \ldots, \overline{K}_i^t$ **do**
 7:             Draw stochastic gradient $g_{i,k}^t \sim \mathcal{D}(x_{i,k-1}^t)$ with mean $\nabla F_i(x_{i,k-1}^t) \in \mathbb{R}^d$
 8:             $m_k \leftarrow g_{i,k}^t$
 9:             **if** $(k-1)/z \in \mathbb{Z}$ **then**
10:                 $v_k \leftarrow v_{k-1} + g_{i,k}^t \odot g_{i,k}^t$
11:             **else**
12:                 $v_k \leftarrow v_{k-1}$
13:             **end if**
14:             **if** $0 < \|m_k/(\sqrt{v_k} + \epsilon)\| < \varepsilon_s$ **then**
15:                 $m_k \leftarrow 0$
16:             **end if**
17:             $x_{i,k}^t \leftarrow x_{i,k-1}^t - \eta_\ell \cdot m_k/(\sqrt{v_k} + \epsilon)$
18:         **end for**
19:         $\Delta_i^t = \Xi_i^t \left( x_{i,\overline{K}_i^t}^t - x_{t-1} \right)$
20:     **end for**
21:     $\Delta_t = \frac{1}{|\mathcal{S}^t|} \sum_{i \in \mathcal{S}^t} \Delta_i^t$
22:     $\widetilde{m}_t = \widetilde{\beta}_1 \widetilde{m}_{t-1} + (1 - \widetilde{\beta}_1)\Delta_t$
23:     $\widetilde{v}_t = \widetilde{v}_{t-1} + \Delta_t^2$
24:     $x_t = x_{t-1} + \eta \frac{\widetilde{m}_t}{\sqrt{\widetilde{v}_t} + \tau}$
25: **end for**

---

We have the following convergence bound.

**Corollary E.1.** *Let* $K^- := \min_{i,t} \overline{K}_i^t \geq 1$ *and*

$$\widetilde{\gamma}_1 := \eta_\ell \varepsilon_w \sum_{p=1}^{K^-} \frac{1}{\sqrt{v_0 + \lceil \frac{K}{z} \rceil G^2} + \varepsilon}, \quad \widetilde{\alpha}_1 := \frac{\varepsilon_w K^-}{2K \left( \sqrt{v_0 + \lceil \frac{K}{z} \rceil G^2} + \varepsilon \right)}.$$

*Then Algorithm 9 has an identical convergence bound to Theorem 5.1.*

Similar to delayed Adam, the proof is analogous to Theorem 5.1 with changes summarized in the following lemma.

**Lemma E.2.** *Under Algorithm 9,* $|\Delta_i^t|$ *is bounded by*

$$|\Delta_i^t| \leq \Phi_1^K := \eta_\ell B \left( \left\lfloor \frac{K-1}{z} \right\rfloor + 1 + \frac{KG}{\sqrt{v_0} + \varepsilon} \right).$$

*Proof.* Recall that $\Delta_t = 1/|\mathcal{S}^t| \sum_{i \in \mathcal{S}^t} \Delta_i^t$ and $\Delta_i^t = \Xi_i^t \left( x_{i,\overline{K}_i^t}^t - x_{i,0}^t \right)$. By telescoping for $\overline{K}_i^t$ local steps and the definition of gradient updates in FedAdaAdagrad, we obtain

$$\Delta_i^t = \sum_{p=1}^{\overline{K}_i^t} -\eta_\ell \Xi_i^t \frac{m_p}{\sqrt{v_p} + \varepsilon} = -\eta_\ell \Xi_i^t \sum_{p=1}^{\overline{K}_i^t} \frac{g_{i,p}^t}{\sqrt{v_0 + \sum_{r=1}^{\lceil \frac{p}{z} \rceil} (g_{i,(r-1)z+1}^t)^2} + \varepsilon}$$

For $\mathcal{F} = \{0, 1, \ldots, \lfloor (\overline{K}_i^t - 1)/z \rfloor\} z + 1$, we thus have that

$$\Delta_i^t = -\eta_\ell \Xi_i^t \sum_{p \in \mathcal{F}} \frac{g_{i,p}^t}{\sqrt{v_0 + \sum_{r=1}^{\lceil \frac{p}{z} \rceil} (g_{i,(r-1)z+1}^t)^2} + \varepsilon}$$

$$- \eta_\ell \Xi_i^t \sum_{p \in [\overline{K}_i^t] \setminus \mathcal{F}} \frac{g_{i,p}^t}{\sqrt{v_0 + \sum_{r=1}^{\lceil \frac{p}{z} \rceil} (g_{i,(r-1)z+1}^t)^2} + \varepsilon}.$$

To obtain a deterministic bound, we cannot ignore the worst-case stochastic realization that $g_{i,(r-1)z+1}^t = 0$ for $\forall r \in [\lceil \frac{p}{z} \rceil]$. Therefore, we form the upper bound

$$\left| \Delta_i^t \right| \leq \eta_\ell |\Xi_i^t| \sum_{p \in \mathcal{F}} \frac{|g_{i,p}^t|}{\sqrt{v_0 + |g_{i,p}^t|^2 + \sum_{r=1}^{\lceil \frac{p}{z} \rceil - 1} (g_{i,(r-1)z+1}^t)^2} + \varepsilon}$$

$$+ \frac{\eta_\ell |\Xi_i^t|}{\sqrt{v_0} + \varepsilon} \left( \sum_{p \in [\overline{K}_i^t] \setminus \mathcal{F}} |g_{i,p}^t| \right) \tag{20}$$

$$\leq \eta_\ell |\Xi_i^t| \left( \left\lfloor \frac{K-1}{z} \right\rfloor + 1 \right) + \frac{\eta_\ell |\Xi_i^t| K G}{\sqrt{v_0} + \varepsilon}$$

where the last line uses that the local epoch schedules are upper bounded by $K$. Noting that $\|\Xi_i^t\|_\infty \leq B$, we are done. $\qquad \square$

# F. Dataset and Models

Below we summarize dataset statistics, the number of clients with each dataset, and models used to train.

Table 2. Summary of datasets

| Datasets | # Devices | Data Partitions | Models | Tasks |
|---|---|---|---|---|
| CIFAR-10 (Krizhevsky, 2009) | 1000 | LDA | ViT-S | Image classification |
| GLD-23K (Weyand et al., 2020) | 233 | natural (each device is a photograher) | ViT-S | 203-class classification |

### F.1. GLD-23K Dataset

The GLD-23k dataset is a subset of the GLD-160k dataset introduced in (Weyand et al., 2020). It contains 23,080 training images, 203 landmark labels, and 233 clients. In Figure 4 we show the convergence of $\texttt{FedAda}^2$ as compared to FedAdam and FedAvg GLD-23K dataset.

### F.2. CIFAR-10 Dataset

The CIFAR-10 dataset (Krizhevsky, 2009) consists of $32 \times 32 \times 3$ images with 10 labels. There are 50,000 training examples and 10,000 test examples. In Figure 5 we show the convergence of $\texttt{FedAda}^2$ as compared to FedAdam and FedAvg using CIFAR-10.

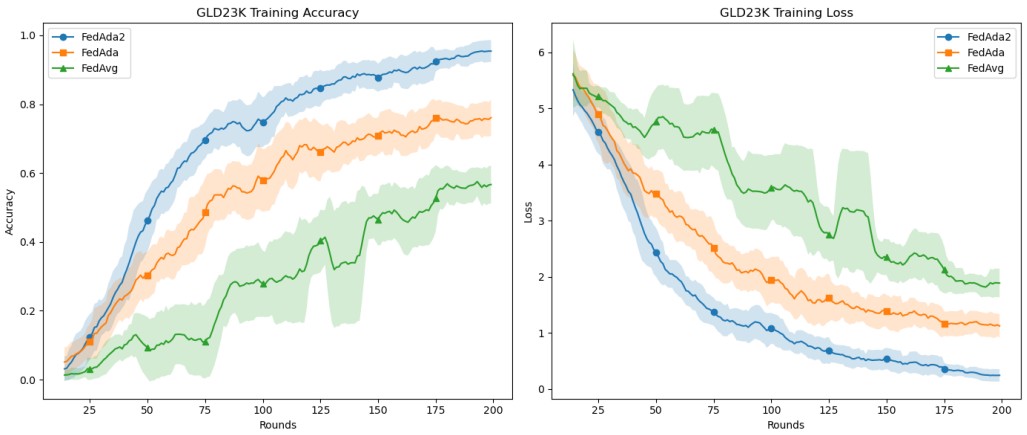

*Figure 4.* GLD23K Dataset Training Accuracy and Loss.

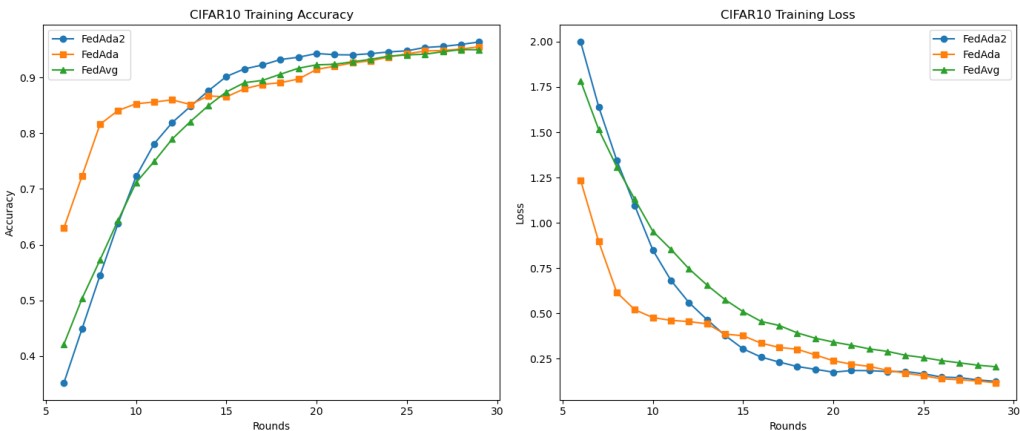

*Figure 5.* CIFAR10 Dataset Training Accuracy and Loss.

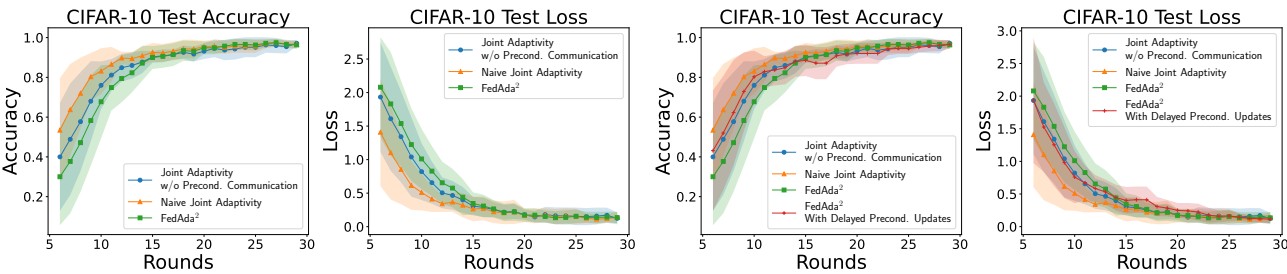

*Figure 6.* An analogue of Figure 3 for the CIFAR-10 dataset, showing performance of low-memory preconditioned federated learning and delayed updates.

## G. Hyperparameters Selection

For both the CIFAR-10 and the GLD-23K dataset, we used Adam optimizer at the client and the server-side. At the client-side $\beta_1$ is set to 0.9 and $\beta_2$ is 0.999 whereas at the server, $\beta_1$ is again 0.9 but $\beta_2$ 0.9 in accordance with Reddi et al. 2021. The hyperparameter grid utilized in the grid search to locate optimal parameters for the optimizers at both the server

and the client-side for all our experiments are given as:

$$\eta_l \in \left\{10^{-6}, 10^{-5}, \ldots, 10^{0}\right\}$$
$$\eta_s \in \left\{10^{-6}, 10^{-5}, \ldots, 10^{0}\right\}$$
$$\tau_l \in \left\{10^{-6}, 10^{-5}, \ldots, 10^{-1}\right\}$$
$$\tau_s \in \left\{10^{-6}, 10^{-5}, \ldots, 10^{-1}\right\}$$

In tables below we summarize the best performing hyperparameters specific to each dataset. The parameters are Log Base-10 that achieve best accuracies.

*Table 3.* Sever Side Learning Rate ($\eta_s$)

|          | FedAvg | FedAdam | FedAda$^2$ |
|----------|--------|---------|------------|
| CIFAR-10 | 0      | -2      | -4         |
| GLD-23K  | 0      | -4      | -4         |

*Table 4.* Client-Side Learning Rate ($\eta_l$)

|          | FedAvg | FedAdam | FedAda$^2$ |
|----------|--------|---------|------------|
| CIFAR-10 | -2     | -2      | -4         |
| GLD-23K  | -1     | -2      | -3         |

*Table 5.* Sever Side Tau ($\tau_s$)

|          | FedAvg | FedAdam | FedAda$^2$ |
|----------|--------|---------|------------|
| CIFAR-10 | 0      | -3      | -5         |
| GLD-23K  | 0      | -5      | -6         |

*Table 6.* Client-Side Tau ($\tau_l$)

|          | FedAvg | FedAdam | FedAda$^2$ |
|----------|--------|---------|------------|
| CIFAR-10 | 0      | 0       | -5         |
| GLD-23K  | 0      | 0       | -2         |

### G.1. Compute Resources

For our experiments, we utilized eight NVIDIA GeForce RTX 2080 Ti GPUs. The entire hyperparameter tuning process along with the data creation process, encompassing both the CIFAR-10 and GLD-23k datasets, required approximately 80 hours of computation time.

