# OpenReview forum: "Efficient Adaptive Federated Optimization"
_ICML.cc/2024/Workshop/WANT — WANT@ICML 2024 Poster_

### Official Review · Reviewer_TQun · 2024-06-12
**Review for paper "Efficient Adaptive Federated Optimization"**

**Confidence:** 3

**Summary:**

This paper proposes $FedAda^2$, an efficient adaptive federated optimiser that leverages constant preconditioning and factorised gradient statistics to achieve low-bandwidth, low-memory adaptive optimisation. Theoretical convergence and empirical evaluation on transformer support the author's claims on negligible degradation of accuracy.

**Strengths:**

* Strong related work section
* Valuable contribution in a central problem in FL, and particularly applicable for attention-based networks
* Inclusion of both theoretical and empirical results to support claims.
* I liked the heterogeneous optimiser setup of client and

**Weaknesses:**

* Missing quantification of memory and bandwidth gains for the experiments run
* Only one modality and type of network has been evaluated
* Analysis is applied on full-gradient descent

**Limitations:**

* There is no mention of how compatible $FedAda^2$ is with local DP-noise.
* How amenable to attacks from malicious actors does $FedAda^2$  make the federated optimisation? Is the only counter-measure applied through gradient-clipping?
* Is the current scheme applicable on asynchronous federated learning aggregation?
* How does the current optimiser behave in low-resource settings where each client only has a budget for very few local steps?
* The authors motivate their method being tailored for cross-device federated learning, but ultimately it is not clear if they have evaluated their claims under partial client participation.

**Suggestions:**

* The evaluation lacks analysis on the memory and bandwidth gains of $FedAda^2$, compared to previous adaptive optimisers. I would sugges that the authors provide these numbers for completeness.
* I would urge the authors to include some discussion on the privacy and robustness of their algorithm.

---

### Official Review · Reviewer_YaJd · 2024-06-13
**Adaptive client optimizer**

**Confidence:** 2

**Summary:**

This paper proposes a class of adaptive distributed learning algorithms to mitigate communication and memory restrictions. It introduces a strategy that allows clients to initialize local preconditioners and adopt a memory-efficient optimizer that factorizes gradient statistics for dimension reduction. The authors prove that their approach achieves similar convergence to other server-side adaptive FL algorithms in non-convex settings.

**Strengths:**

This paper proposes an interesting approach for adaptive client optimization, which is not widely addressed in the FL community. The authors provide extensive discussion on the topic, present the motivation for the problem, and offer technical analysis. Numerical experiments are also provided to support the methods.

**Weaknesses:**

I find the paper very hard to follow and lacking in narrative. The motivation for using adaptive clients is not clearly explained. Even though a client with heavy-tailed gradients can potentially harm the training process, it is unclear if this phenomenon is guaranteed in the domain of Byzantine machine learning. Additionally, I think some existing gradient-based client selection methods can also address this problem, so it is not clear why adaptive clients are the preferred approach.

---

### Decision · Program_Chairs · 2024-06-18

**Decision:**

Accept (Poster)

**Comment:**

We thank the authors for their time and contribution to WANT and we are pleased to share that after the reviewing process the paper has been accepted. Congratulations! We encourage the authors to consider reviewers' feedback for the improvement of the camera-ready version. We hope to see you in person at the workshop and brainstorm on efficient training research together!